**Progress in understanding of Indian Ocean circulation, variability, air-sea exchange and impacts on**
**biogeochemistry**
Helen E. Phillips[1,2], Amit Tandon[3], Ryo Furue[4], Raleigh Hood[5], Caroline C. Ummenhofer[6,7], Jessica A. Benthuysen[8],
Viviane Menezes[6], Shijian Hu[9], Ben Webber[10], Alejandra Sanchez-Franks[11], Deepak Cherian[12], Emily Shroyer[13], Ming
Feng[14,15], Hemantha Wijesekera[16], Abhisek Chatterjee[17], Lisan Yu[6], Juliet Hermes[18], Raghu Murtugudde[19], Tomoki
Tozuka[20,4], Danielle Su[21], Arvind Singh[22], Luca Centurioni[23], Satya Prakash[17], Jerry Wiggert[24]
[1]Institute for Marine and Antarctic Studies, University of Tasmania, Hobart, 7005, Australia
[2]Australian Antarctic Program Partnership, Institute for Marine and Antarctic Studies, University of Tasmania, Hobart,
7005, Australia
[3]Department of Mechanical Engineering, College of Engineering, University of Massachusetts Dartmouth, 02747, USA
[4]APL/JAMSTEC, Yokohama, Japan
[5]University of Maryland Center for Environmental Science, Horn Point Laboratory, Cambridge, 21613, USA
[6]Department of Physical Oceanography, Woods Hole Oceanographic Institution, Woods Hole, 02543, USA
[7]ARC Centre of Excellence for Climate Extremes, University of New South Wales, Sydney, Australia
[8]Australian Institute of Marine Science, Indian Ocean Marine Research Centre, Crawley, Australia
[9]Institute of Oceanology, Chinese Academy of Sciences, Qingdao, China
[10]School of Environmental Sciences, University of East Anglia, Norwich, NR4 7TJ, UK
[11]National Oceanography Centre, Southampton, UK
[12]National Center for Atmospheric Research, Boulder, USA
[13]College of Earth, Ocean and Atmospheric Sciences, Oregon State University, Corvallis, 97331, USA
[14]CSIRO Oceans and Atmosphere, Indian Ocean Marine Research Centre, Crawley, Australia
[15]Centre for Southern Hemisphere Oceans Research, Hobart, Australia
[16]U.S. Naval Research Laboratory, Stennis Space Center, MS, 39529 , USA
[17]Indian National Centre for Ocean Information Services, Ministry of Earth Sciences, Hyderabad, India
[18]South African Environmental Observation Network, Cape Town, South Africa
[19]Department of Atmospheric and Oceanic Science, University of Maryland, College Park, 20742, USA
[20]Department of Earth and Planetary Science, Graduate School of Science, The University of Tokyo, Tokyo, Japan
[21]Sorbonne Universités, UPMC Université Paris 06, CNRS, UMR 7159 LOCEAN-IPSL, Paris, France
[22]Physical Research Laboratory, Ahmedabad, India
[23]Scripps Institution of Oceanography, University of California San Diego, La Jolla, 92093, USA
[24]University of Southern Mississippi, Hattiesburg, 399406, USA

*Correspondence to*: Helen E. Phillips (h.e.phillips@utas.edu.au)

Dedicated to Dr Satya Prakash (1979-2021). This manuscript was written during the COVID-19 pandemic while many juggled family and health issues under lockdown. Satya, our co-author and friend, passed away on 22nd July 2021. Satya was the coordinator of the International Indian Ocean Expedition (IIOE-2) at the Indian National Centre for Ocean Information Services (INCOIS). He played an integral part in Indian Ocean research. He is remembered for his enthusiasm, commitment and smile. His passing is a massive loss to the Indian Ocean community, but there will be much that will still be carried on in his memory thanks to his hard work and passion.

**Abstract.** Over the past decade, our understanding of the Indian Ocean has advanced through concerted efforts toward measuring the ocean circulation and air-sea exchanges, detecting changes in water masses, and linking physical processes to ecologically important variables. New circulation pathways and mechanisms have been discovered, which control atmospheric and oceanic mean state and variability. This review brings together new understanding of the ocean-atmosphere system in the Indian Ocean since the last comprehensive review, describing the Indian Ocean circulation patterns, air-sea interactions and climate variability. Coordinated international focus on the Indian Ocean has motivated the application of new technologies to deliver higher-resolution observations and models of Indian Ocean processes. As a result we are discovering the importance of small-scale processes in setting the large-scale gradients and circulation, interactions between physical and biogeochemical processes, interactions between boundary currents and the interior, and between the surface and the deep ocean. A newly discovered regional climate mode in the southeast Indian Ocean, the Ningaloo Niño, has instigated more regional air-sea coupling and marine heatwave research in the global oceans. In the last decade, we have seen rapid warming of the Indian Ocean overlaid with extremes in the form of marine heatwaves. These events have motivated studies that have delivered new insight into the variability in ocean heat content and exchanges in the Indian Ocean, and have highlighted the critical role of the Indian Ocean as a clearing house for anthropogenic heat. This synthesis paper reviews the advances in these areas in the last decade.

**Contents**

## 1. Introduction

The physical processes taking place in the Indian Ocean and overlying atmosphere underpin the variability evident in monsoons, extreme events, marine biogeochemical cycles, ecosystems, and ultimately human experience. The Indian Ocean rim countries, accounting for one third of the Earth's human population, depend on this ocean for food and resources, and are dramatically impacted by its variability (Hermes et al., 2019). Increasing our understanding of

interactions between geologic, oceanic and atmospheric processes that control the complex physical dynamics of the Indian Ocean region is a priority for many national, bilateral, and international programmes including the Indian Ocean Observing System (IndOOS; Beal et al., 2020), the Climate and Ocean: Variability, Predictability and Change (CLIVAR)/Intergovermental Oceanographic Commission (IOC) - Indian Ocean Region Panel (https://www.clivar.org/sites/default/files/documents/indian/135_IOP5.pdf), and the second International Indian Ocean Expedition (IIOE-2), to name a few. While initiated through IIOE-2, this review draws on the collective results of all of the programmes and individual efforts. We focus, in particular, on questions about the Indian Ocean circulation, climate variability and change such as: 1) how have the atmospheric and oceanic circulation of the Indian Ocean changed in the past and how will they change in the future; 2) how do these changes relate to geography and connectivity with the Pacific, Atlantic and Southern oceans; and 3) what impact does the circulation, variability, and change have on biological productivity and fisheries.

Recent focus on the Indian Ocean has motivated new international efforts in field campaigns and modelling studies, and leveraged advances in global observations that contribute to the Indian Ocean Observing System (IndOOS; Beal et al., 2020). The Argo profiling float array (Roemmich et al., 2012) reached full coverage in the Indian Ocean in 2006, the RAMA moored buoy array (McPhaden et al., 2009) has now delivered multi-year time series of tropical oceanic and atmospheric variability, with some sites dating back to 2000. Satellite systems continue to provide observations vital to interpreting spatial and temporal variability in the in situ observations, and new technology is now enabling high resolution observations of boundary current variability and small scale processes. Thus, since the reviews of Schott and McCreary (2001) and Schott et al. (2009), the spatial coverage of observations and length of time series have increased substantially such that the signals of many previously unresolved processes are now able to be observed.

These new higher-resolution observations and companion improvements in model simulations have highlighted the importance of small scale processes in setting the large-scale gradients and circulation, interactions between physical and biogeochemical processes, interactions between boundary currents and the interior, and between the surface and the deep ocean. Overlaid on these interior Indian Ocean processes, ocean warming due to increasing greenhouse gas concentrations has been shown to be pervasive and relentless (Wijffels et al., 2016), and extending to abyssal depths (Johnson et al., 2008a; Desbruyeres et al., 2017).

The Indian Ocean plays a key role in the global climate system, enabling upwelling of the lower cell of the meridional overturning circulation from abyssal to upper-deep and intermediate waters through diffusive mixing (Schmitz, 1995; Lumpkin and Speer, 2007; McDonagh et al., 2008; Talley, 2013; Hernandez-Guerra and Talley, 2016) and exporting the largest poleward heat flux of all Southern Hemisphere basins (Roxy et al., 2014). In recent decades, the upper 700 m of the entire Indian Ocean has warmed rapidly (Desbruyères et al, 2017). In the southern Indian Ocean, the warming was

directly linked primarily to heat advection from a strengthened ITF and, secondly, to a decrease in mean air-sea flux cooling (Li et al., 2017b; Zhang et al., 2018a). This coupling between the ocean and atmosphere in the Indian and Pacific Oceans shifted the balance of global warming, accelerating ocean warming and causing a hiatus in the warming of Earth's surface atmosphere (Section 6). Marine heatwaves have emerged as an increasing threat to marine ecosystems as ocean temperatures warm (e.g. Oliver et al., 2018). Increasingly vulnerable populations need more reliable monsoon predictions, a task complicated by variability across timescales from intraseasonal to interannual, decadal and beyond in a tightly coupled ocean-atmosphere system (Hazra et al., 2017).

The starting point for this synthesis report are the reviews by Schott and McCreary (2001) and Schott et al. (2009), describing the circulation patterns, air-sea interactions and climate variability on timescales from intraseasonal to interannual, and relatively large spatial scales. We begin with a description of the large scale setting that has been well established since Schott et al. (2009) (Section 2). We then consider the structure and propagation of variability in air-sea interactions at seasonal and intra-seasonal scales, including the contribution of the mesoscale and the ocean's role in air-sea interaction (Section 3). Section 4 discusses new advances in understanding of the upper ocean circulation, organised by region (southern basin, equatorial and northern basin). This section includes an update of the near-surface circulation maps of Talley et al. (2011), including recent work on boundary currents around Australia and Madagascar, and a discussion of the biogeochemical variability observed in each region. The interocean connections with the Pacific, Atlantic and Southern Oceans are discussed in Section 5. Section 6 describes the variability of the Indian Ocean circulation with the recent advances in understanding the warming across the basin, climate modes such as the Indian Ocean Dipole, connection with the El Nino-Southern Oscillation (ENSO), and Indian ocean marine heatwaves. Section 7 focuses on multiscale processes in the Bay of Bengal as an "ocean laboratory", since there have been multiple international programs in this Bay in the last decade. Recent advances from the large scales (>100 km) down to sub-mesoscales (100 m to 10 km) and further down to mixing scales (mm) are discussed. We then link back from mixing to large scales via salinity budgets and coupled phenomena such as the Madden-Julian Oscillation (MJO) to understand the complexity of these processes across multiple scales. We end with a short summary and open questions that will need to be addressed over the next decade.

## 2 Large-scale setting

The oceanic and atmospheric circulation of the Indian Ocean are unlike those in the Pacific and Atlantic oceans, largely due to geography. The Asian landmass limits the northern extent of the Indian Ocean to around 25°N so that there is no high-latitude cooling of the ocean, and consequently no dense water formation such as that seen in the North Atlantic and, to some extent, the North Pacific. The intense seasonal variation in temperature over Asia drives the seasonal monsoons: the southwest monsoon in boreal summer, and northeast monsoon in boreal winter. The timing of the onset of the monsoon,

and associated wet and dry periods in the Indian Ocean rim countries, varies considerably depending on a range of large-scale climate modes and smaller-scale coupled ocean-atmosphere interactions. The seasonally-reversing winds drive seasonally-reversing ocean currents in the northern Indian Ocean (Section 4.4), e.g. the southwest/northeast monsoon current and the Somali Current. Equatorial currents in the Indian Ocean, eastward near the surface above westward undercurrents (Section 4.3), provide rapid connection between the western and eastern basin and are also subject to monsoon dynamics.

In the southern Indian Ocean (Section 4.2), the connection of the Indian and Pacific Oceans through the Indonesian Seas also contributes to the unique circulation patterns. The very warm and fresh ITF water is funneled into the tropical southern Indian Ocean and carried westward by the South Equatorial Current. The warm, fresh waters are much lighter than those further south, creating a north-south density (pressure) gradient that drives near-surface broad, eastward geostrophic currents between 16°S and 32°S and between Madagascar and Australia (Niiler et al., 2003). This pressure gradient also generates the Leeuwin Current, a unique poleward-flowing eastern boundary current (Godfrey and Ridgway, 1985) that is a downwelling region but is also, counter-intuitively, highly productive (Waite et al., 2007). These two features are not found in the southeastern Atlantic and Pacific oceans. There, the eastern basin currents are characterised by a clear subtropical gyre circulation with weak, equatorward flow and upwelling against the coast.

The tropical Indian Ocean (Section 4.3) is home to the largest fraction of sea surface temperature (SST) warmer than 28°C (the tropical warm pool), and is therefore a key region for deep atmospheric convection: the upward part of the Walker Circulation that drives cloud formation and precipitation over the tropical Indo-Pacific. Variation in SST is the primary driver of variation in exchanges between the ocean and atmosphere and is thus a key focus in this paper. Sea surface salinity effects on ocean-atmosphere exchanges have become better understood and are discussed throughout and in particular in Section 7.

The tropical Indian Ocean sea surface temperature (SST) has warmed faster over the period since 1950 than either the tropical Pacific or Atlantic (Han et al., 2014, Fox-Kemper et al., 2021), with implications for primary productivity (Roxy et al., 2014, 2016). The Indian Ocean accounts for 50-70% of the total ocean heat uptake in the global upper (700 m) ocean over the last decade, associated with anthropogenic warming  (Lee et al., 2015). The deeper ocean (700-2000 m) is warming across the globe with a robust signature of anthropogenic warming evident even in the short Argo record since 2005 (Wijffels et al. 2016, Rathore et al. 2020). Warming in the abyss is detectable and widespread, communicated from the surface of the ocean along pathways from Antarctic Bottom Water formation regions (Purkey and Johnson, 2012). Considerable variability in the Indian Ocean climate system exists on the backdrop of this strong, long-term warming trend.

An extensive debate erupted in recent years about whether there was hiatus or a reduced rate of global warming
(Lewandowsky et al. 2018). However, persistent cold anomalies in the eastern Pacific have been argued to have enhanced
oceanic heat uptake, and the strengthened trade winds are consistent with this argument (Kosaka and Xie 2013, England
et al. 2014). It has further been argued that the excess heat taken up by the tropical Pacific has been pumped into the Indian
Ocean via the Indonesian throughflow (Lee et al. 2015). The tropical Indian Ocean is likely affected by the Southern
Ocean trends at a rapid timescale of the order of a decade (Yang et al. 2020), and the Indian Ocean warming may accelerate
the Atlantic meridional overturning circulation (Hu et al. 2019) and the Pacific response to anthropogenic forcing (Zhang
et al. 2019). Based on these oceanic tunnels and atmospheric bridges into and out of the Indian Ocean, one could
hypothesise that the Indian Ocean may be acting as the clearinghouse for oceanic warming under anthropogenic forcing.
Variability in the oceanic and atmospheric circulation of the Indian Ocean is the result of complex interactions that are
both internal and external to the Indian Ocean. The recent review of the IndOOS plan (Beal et al., 2019, 2020) summarises
the major scientific drivers, of which we still have limited understanding (Fig. 1). The over-arching signal is anthropogenic
climate change, causing a background trend of ocean warming and increasing acidity due to uptake of heat and carbon
dioxide and affecting the nature of large and small scale variability mechanisms.

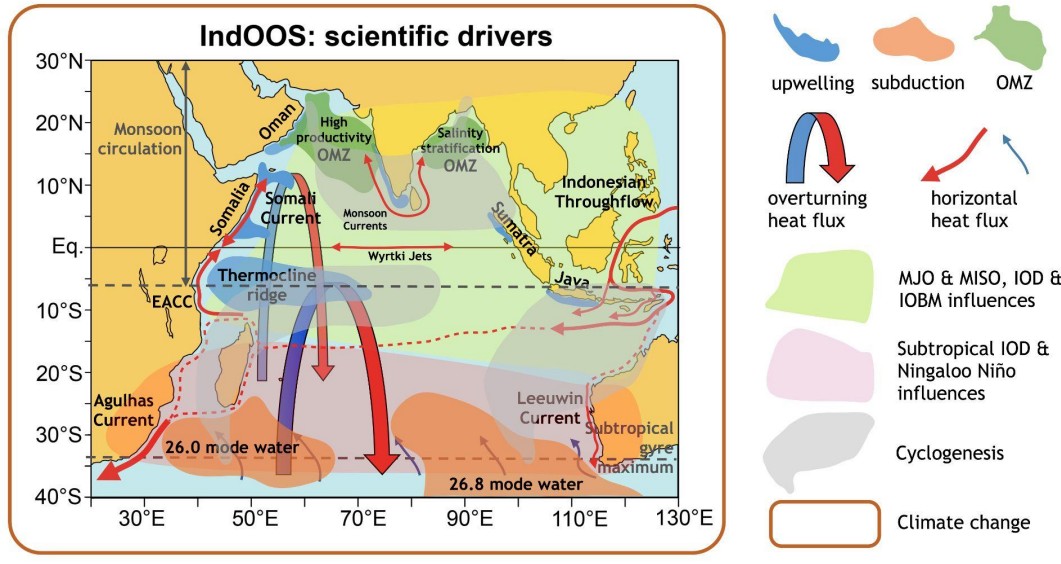


**Figure 1: Schematic view of key phenomena in the Indian Ocean (from Beal et al. 2019). The main scientific drivers**
**of the Indian Ocean Observing System, including the Oxygen Minimum Zones (OMZs), upwelling and subduction**
**zones, major heat flux components, the tropical modes of the Madden-Julian Oscillation (MJO), the Monsoon**

**248**     **Intra-Seasonal Oscillation (MISO), the Indian Ocean Dipole (IOD) and Indian Ocean Basin Mode (IOBM), the**

**249**     **subtropical modes of Ningaloo Niño and subtropical IOD, cyclogenesis, and climate change.**

A net poleward flow of heat out of the Indian Ocean is accomplished by a combination of the horizontal circulation along
the boundaries, coupled with the Indian Ocean's part of the global meridional overturning circulation (MOC) and shallow
overturning cells. The ITF delivers heat from the Pacific into the Indian Ocean. The Agulhas Current moves heat rapidly
southward at surface and intermediate depths (Bryden and Beal, 2001), with 30% of Indian Ocean heat export thought to
be carried across 32°S by this gyre circulation (Talley, 2008). The shallow Leeuwin Current makes a smaller direct
contribution to the poleward flow of heat (Smith et al., 1991; Feng et al., 2003; Furue et al., 2017) but generates a rich
field of mesoscale eddies that carry heat and momentum into the Indian Ocean interior, contributing to heat export across
32°S (Domingues et al. 2006, Feng et al., 2007; Dilmahamod et al. 2018).
In the upper ocean, the shallow overturning consists of the cross-equatorial cell (Miyama et al. 2003; Schott et al. 2004)
and the subtropical cell (Schott et al. 2004). The ascending branches of these cells connect to different upwelling zones in
the southern and northern Indian Ocean and, therefore, play an important role in regulating the climatological mean,
seasonal, and interannual heat balance in the tropical Indian Ocean (Lee 2004; Lee and McPhaden 2008). At intermediate
depths (500-2000 m), mode waters of varying density enter the Indian Ocean from the Southern Ocean. Along their
northward path they mix with lighter waters above, progressively upwelling to the sea surface in a range of locations north
of 10°S to then return south in a widespread southward Ekman transport of near-surface waters (Schott et al., 2009). The
lower part of the mode water layer mixes with denser waters below and joins the southward flowing deep waters (2000-
4000 m). This southward flow also has a contribution from transformed abyssal waters: Antarctic Bottom Water moves
northward at abyssal depths, mixing with lighter waters above, progressively upwelling along its path from the Southern
Ocean to the Indian Ocean to return southward at shallower depths (Talley, 2013). Cross-equatorial flow is accomplished
both at abyssal levels and via the East Africa Coastal Current, seasonally reversing Somali Current (Schott et al., 2009)
and southward Ekman transport (Schott and McCreary, 2001).
The remaining elements of Fig. 1 refer to oxygen minimum zones (OMZ) in the Arabian Sea and Bay of Bengal and the
range of mechanisms that drive strong variations in sea surface temperature leading to shifts in atmospheric convection
and precipitation with major effects on rim countries. These mechanisms include: Madden-Julian oscillation (MJO) and
Monsoon Intraseasonal Oscillation (MISO), Indian Ocean Dipole (IOD), Indian Ocean Basin Mode, Subtropical IOD, and
Ningaloo Niño which are discussed further in Section 6. Cyclogenesis is not discussed in this synthesis. For discussion of
OMZ, the reader is referred to the review papers of McCreary et al. (2013) and Rixen et al. (2020).
Extreme precipitation in the Bay of Bengal and evaporation in the Red Sea and Arabian Sea lead to strong variability in
ocean salinity that in turn impacts ocean circulation and air-sea interaction. The surface salinity gradient in the northern

Indian Ocean decreases from the Arabian Sea in the west to the Bay of Bengal in the east. Strong evaporation over the Arabian Sea results in highly saline surface waters (Antonov et al., 2010; Chaterjee et al., 2012), while surface waters in the Bay of Bengal are comparatively fresh and highly stratified as a result of monsoon precipitation and outflow from river systems such as the Ganges-Brahmaputra (Shetye et al., 1996; Vinayachandran et al., 2002). The surface forcing is balanced by the seasonally reversing monsoon currents to maintain the climatological distribution of salinity.

## 3 Air-sea interactions

The tropical Indian Ocean is highly variable across multiple scales, all of which involve atmosphere-ocean interaction: from the locally intense heat and moisture fluxes that drives tropical cyclones to large-scale convection in the ascending branch of the Hadley circulation, and basin scale ocean heat transport carried by overturning cells that contribute to decadal variability and trends. At intermediate time scales, the intraseasonal oscillations involve strong air-sea coupling (e.g., Demott et al., 2015). The Indian Ocean Dipole (IOD) is an example of an inherently coupled mode of variability (Saji et al., 1999, Webster et al. 1999, Murtugudde et al. 2000). The monsoonal rainfall around the Indian Ocean is largely fuelled by warm SSTs and strong sea-to-air moisture fluxes. These phenomena emphasise the need to understand the mechanisms of air-sea interaction within the Indian Ocean, with a particular focus on how these processes can be better represented in models to aid predictions of variability in the Earth system.

## 3.1 Seasonal cycle and the monsoons

In the open ocean south of 10°S, the wind pattern throughout the year is southeasterly trade winds across the tropics and subtropics and westerlies south of 35°S (Fig. 2). The evaporative cooling of the ocean surface by the trade winds leads to high salinity throughout the subtropics. The curl of the wind stress drives year-round Ekman pumping (downwelling) south of around 15°S (Fig. 2). Downwelling of these denser, high salinity surface waters supplies the downward limb of the shallow Subtropical Cell, STC and Cross-Equatorial Cell, CEC (Schott et al. 2002; Miyama et al. 2003; Schott et al., 2004; Lee 2004; Schott et al., 2009). The subsurface path of the shallow overturning is not well known, and the return to the surface is in any of a number of upwelling zones including the Seychelles-Chagos Thermocline Ridge for the STC and along Somalia, Oman and the west coast of India for the CEC. North of around 10°S, the winds over the Indian Ocean are characterised by seasonal reversals due to the monsoons (Fig. 2), which in turn cause most of the near-surface currents in these regions to seasonally reverse (Schott et al., 2009; Shankar et al., 2002, Section 4.4).

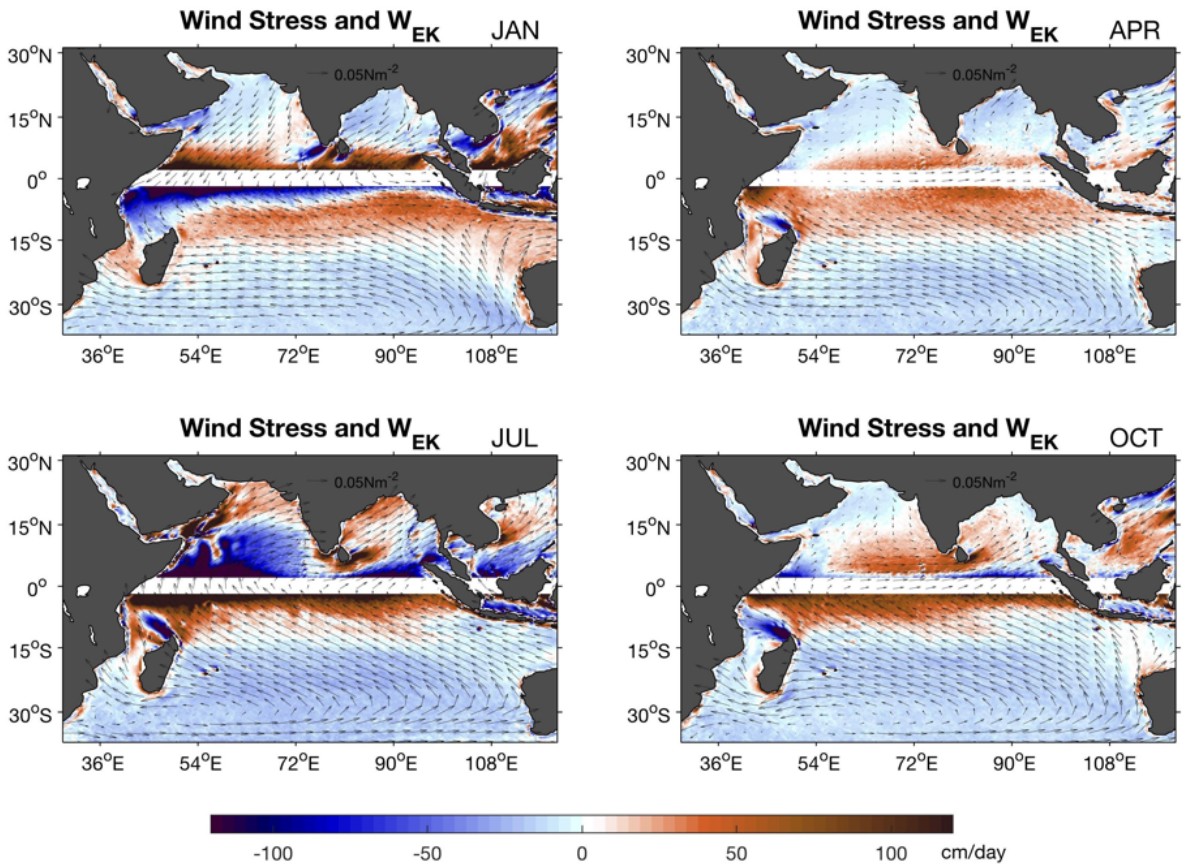

306

**Figure 2: Climatology (2001–2018) of monsoon wind stress (vectors) and Ekman pumping rate (colour shaded) with positive values denoting Ekman suction (upwelling) and negative values Ekman pumping (downwelling) for (a) January - NE monsoon, (b) April, (c) July - SW monsoon, and (d) October. The climatology was constructed by the Objectively Analyzed air-sea Flux High-Resolution (OAFlux-HR) analysis (adapted from Yu 2019).**

A strong positive correlation between seasonal net heat fluxes into the ocean and SST variability (Fig. 3) suggests that the seasonal cycle of SST is largely due to the seasonal cycle of winds and cloud cover (Yu et al., 2007). One prominent exception is the Seychelles-Chagos thermocline ridge (located between 5°S and 10°S and east of 50°E), where upwelling and horizontal advection exhibit substantial seasonal variations that in turn contribute to the seasonal cycle of SST (Hermes and Reason, 2008; Foltz et al., 2010). On the equator and to the north, seasonally reversing winds drive complex patterns of upwelling and downwelling that lead to complex SST variability.

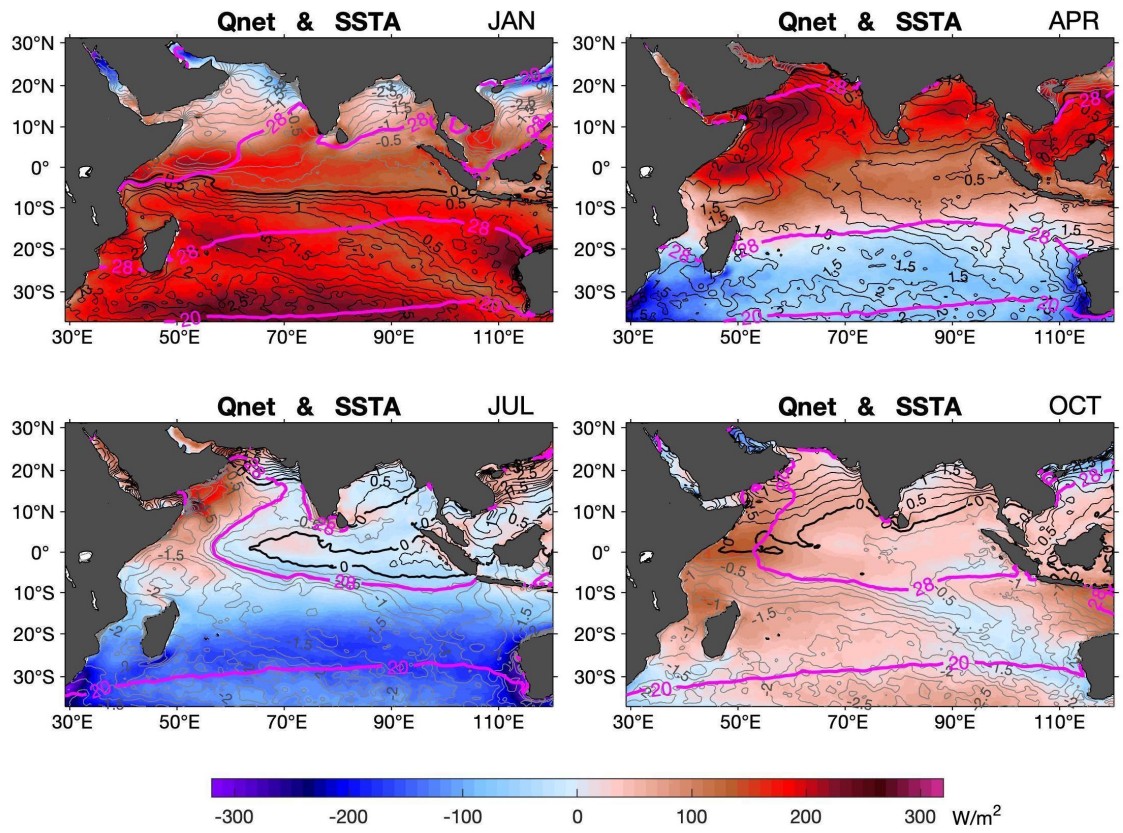

317

**Figure 3: Climatology (2001–2018) of ocean-surface net heat input (colour shaded; positive values denote ocean heat gain and negative values ocean heat loss), SST anomaly (black contours) and 20℃, 28℃ SST contours (pink) for (a) January - NE monsoon, (b) April, (c) July - SW monsoon, and (d) October (adapted from Yu 2019). Net heat flux is the sum of solar radiation, longwave radiation, and turbulent latent and sensible heat fluxes. The turbulent heat flux climatology was constructed by the OAFlux-HR analysis and surface radiation climatology by the NASA CERES EBAF (Kato et al., 2013).**

In the Bay of Bengal and Arabian Sea, surface heat fluxes dominate the seasonal cycle of SST, with the exception of the upwelling zone along the western boundary of the Arabian Sea (Chowdary et al., 2015; Yu et al., 2007). However, salinity effects and subsurface processes (barrier layers, vertical entrainment, variations in the depth of penetration of solar radiation and zonal advection also influence SST variability (Thangaprakash et al., 2016). Rainfall variability driven by the monsoons creates near-surface salinity variability, most notably in the Bay of Bengal where there is a pronounced annual cycle of sea surface salinity (SSS; Fig. 4, Akhil et al., 2014). Freshwater input at the northern end of the Bay forms a shallow mixed layer stratified by low salinity and is advected southwards along the east coast of India, where it is eventually eroded by vertical mixing (Akhil et al., 2014). The variability in freshwater input contributes to the seasonal

cycle of barrier layer thickness in the Bay of Bengal (Howden and Murtugudde, 2001; Thadathil et al., 2007), which in turn modulates how strongly SST responds to surface forcing (Li et al., 2017). The seasonally reversing currents that connect the salty Arabian Sea and fresh Bay of Bengal also strongly influence sea surface salinity patterns (Section 4.1.3).

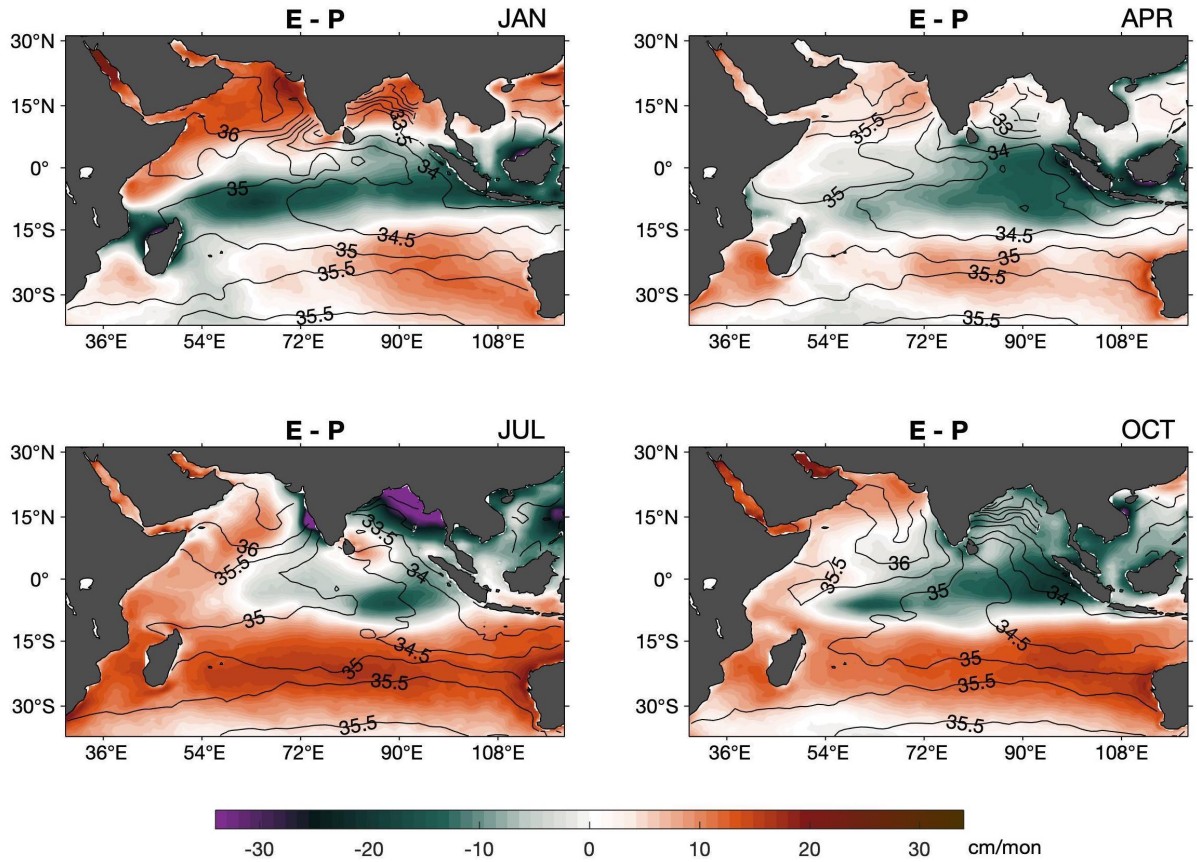

**Figure 4: Climatology (2001–2018) of evaporation minus precipitation (colour shaded; positive values denote freshwater leaving the ocean and negative values addition of fresh water to the ocean) and sea surface salinity (black contours) for (a) January - NE monsoon, (b) April, (c) July - SW monsoon, and (d) October (adapted from Yu 2019).**

The seasonal cycles in the atmosphere and ocean circulation strongly influence the biological productivity of the near-surface Indian Ocean (Wiggert et al. 2006). Fig. 5 shows the seasonal cycle of satellite chlorophyll *a* and surface currents. The dramatically low productivity in the subtropics, where wind stress curl drives large-scale downwelling (Fig. 2), and highly productive coastal boundaries where wind-driven upwelling occurs, highlights the impact of the circulation and atmosphere-ocean interaction on biological productivity. In turn, the chlorophyll *a* distribution has important implications for air-sea interaction, since higher concentrations of phytoplankton lead to increased absorption of solar radiation (e.g.,

Morel and Antoine, 1994; Murtugudde et al. 2002; Giddings et al. 2021). Organisation of chlorophyll *a* at intraseasonal

timescales has also been reported (Section 3.2.1).

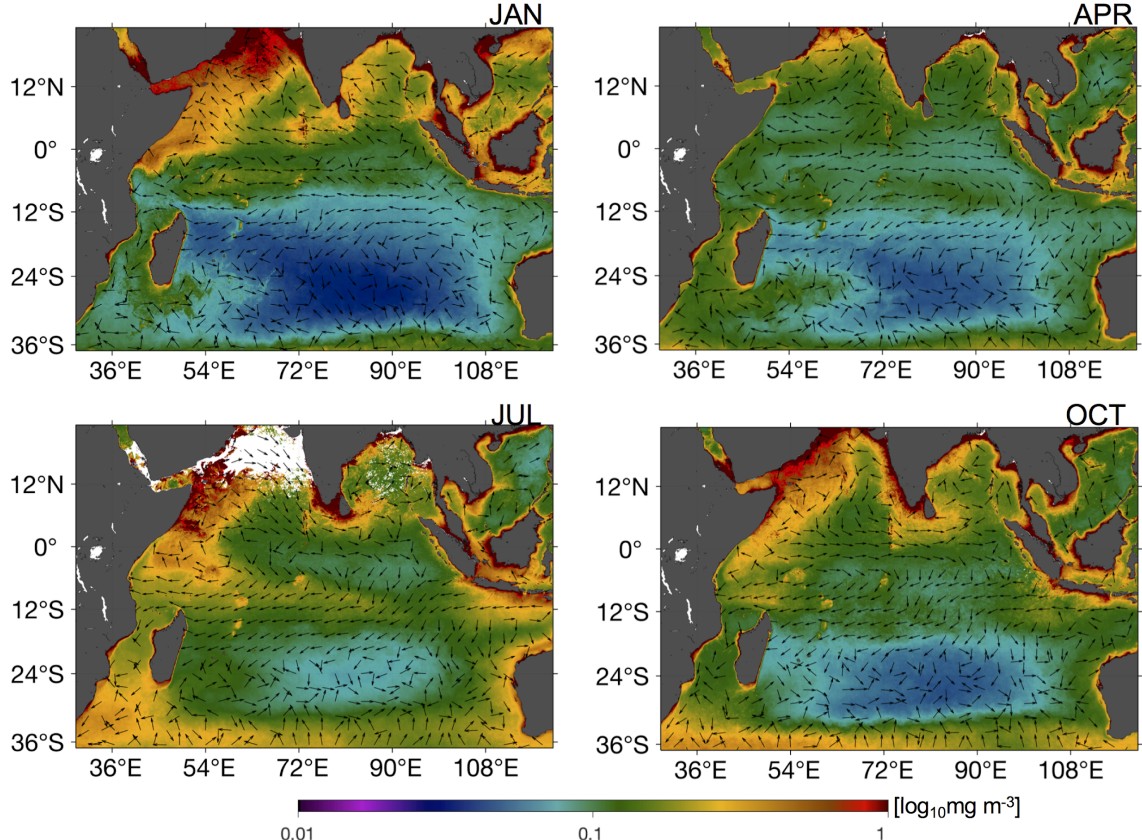

**Figure 5: Climatology (2002-2018) of chlorophyll-a concentrations (colormap) and current velocities (arrows) for (a) January (b) April (c) July (d) October. Chlorophyll a climatology was obtained from the MODIS-Aqua product and current velocities were obtained from the third-degree Ocean Surface Current Analysis Real-time (OSCAR) product.**

## 3.2 Intraseasonal air-sea interaction

### 3.2.1 Madden-Julian Oscillation - MJO

The Madden-Julian Oscillation (MJO; Madden and Julian, 1972, 1971) is the dominant mode of variability in the Indian Ocean at subseasonal time scales. The MJO (Fig. 6) is characterised by eastward-propagating features of enhanced and reduced convection over distances of more than 10,000 km and with a periodicity of around 30–60 days (Zhang, 2005).

The MJO propagates slowly (~5 m s$^{-1}$) through the portion of the Indian and Pacific Oceans where the sea surface is
warm, constantly interacting with the underlying ocean and influencing many weather and climate systems. Within the
large-scale envelopes of enhanced convection, smaller-scale clusters of clouds propagate westward, and can produce local
extremes in rainfall. Air-sea interaction is believed to sustain, and perhaps amplify, the patterns of enhanced and reduced
convection as the MJO propagates eastward (Demott et al., 2015). Indo-Pacific warming trends are warping the life cycle
of the MJO, which is spending less time over the Indian Ocean, more time over the Pacific and altering mean rainfall
trends in parts of the globe (Roxy et al, 2019).

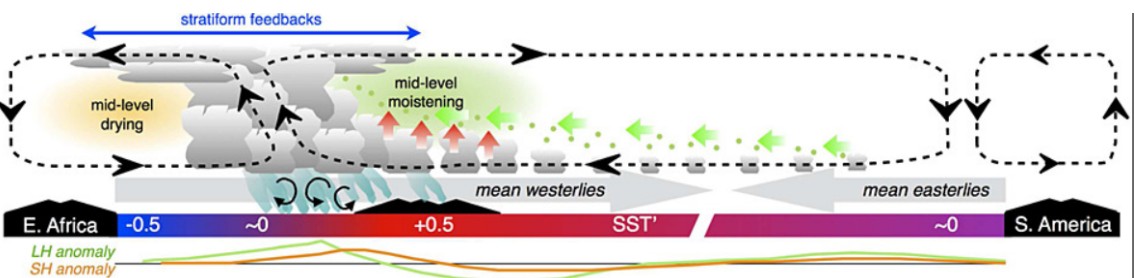


**Figure 6: Schematic depiction of Indian and Pacific Ocean feedbacks to the MJO when convection (gray cloud**
**elements) is maximized in the eastern Indian Ocean. Rainfall (aquamarine), circulation anomalies (black dashed**
**cells), convective downdrafts (black rotor arrows), mean winds (faint gray arrows), and moistening by convective**
**detrainment (small green dots) and horizontal and vertical advection (thick green and red arrows, respectively) are**
**overlaid. Net moistening (drying) is shaded green (orange). Positive (red) and negative (blue) SST anomalies for a**
**strong event are shaded, while latent (sensible) heat flux anomalies are shown with green (orange) curves. Central**
**and East Pacific spatial scale is compressed relative to the Warm Pool. Adapted from DeMott et al. (2015).**
The MJO-related pattern of winds results in anomalous westerly (easterly) winds to the west (east) of the region of
convergence, convection and enhanced rainfall (Fig. 6). These winds generate Kelvin and Rossby waves along the Equator.
The Kelvin waves generated by the MJO have been hypothesised (Kessler et al, 1995; McPhaden 1999; Bergman et al.
2001) to trigger ENSO events in the Pacific. In the Indian Ocean, there is a distinctive sequence of basin-scale ocean waves
generated by the MJO. Eastward-propagating equatorial ocean Kelvin waves strike the coast of Sumatra, where they
generate coastally-trapped Kelvin waves that propagate northward and southward away from the generation site. Kelvin
waves also propagate into the Indonesian seas where they affect the ITF (Pujiana and McPhaden, 2020). Westward-
propagating equatorial ocean Rossby waves are also formed, either due to direct intraseasonal wind forcing or through
reflection of Kelvin waves at the eastern boundary (Oliver and Thompson, 2010; Webber et al., 2010; Nagura and
McPhaden, 2012; Pujiana and McPhaden, 2020). These waves influence local upwelling and currents; they have been
linked to variability in coastal currents around the Bay of Bengal (Vialard et al., 2009), to enhancement of the spring
Wyrtki jets in the eastern equatorial Indian Ocean (Prerna et al., 2019), to changes in subsurface equatorial currents in the

central Indian Ocean (Iskandar and McPhaden, 2011) and to changes in upwelling and chlorophyll a concentration in the off-equatorial central Indian Ocean (Webber et al., 2014). Such waves also propagate energy downwards into the deep ocean (e.g., Pujiana and McPhaden, 2020), contributing to deep ocean variability at multiple time scales (e.g., Matthews et al., 2007). Downwelling Rossby waves in the western Indian Ocean create positive SST anomalies through a combination of reduced entrainment of cooler water from below and zonal advection (Rydbeck et al., 2017; Webber et al., 2012b). These waves therefore act as a triggering mechanism for new MJO events (Rydbeck and Jensen, 2017; Webber et al., 2010, 2012b, 2012a), and may also play a role in amplifying existing MJO events.

MJO-related winds also lead to variability in mixing within and at the bottom of the mixed layer. Westerly wind bursts generate zonal currents that create strong vertical current shear (Moum et al., 2014). These currents and the associated mixing persist after the passage of the atmospheric disturbance. Cooler waters from below the surface are mixed with surface waters, leading to a reduction in available ocean heat content for the next MJO event and thus reducing its potential amplitude (Moum et al., 2016). By examining the causes of SST variability in two separate MJO events, McPhaden and Foltz (2013) showed that the presence or absence of barrier layers may play a crucial role in determining how strongly mixing and vertical entrainment influence SST. They also found that zonal advection plays a relatively stronger role when a barrier layer is present. Chi et al. (2014) confirmed the importance of barrier layers in influencing the turbulent heat flux, but found that thin barrier layers can be eroded by strong current shear that occurs during active phases of the MJO. Wind mixing and surface heat and freshwater fluxes both contribute in roughly equal proportions to intraseasonal variability in mixed layer depth (Keerthi et al., 2016).

Various studies have investigated the relative importance of surface heat fluxes and subsurface ocean processes for the evolution of SST at intraseasonal time scales. The Seychelles-Chagos Thermocline Ridge (SCTR), is a region of high intraseasonal SST variability (Saji et al. 2006, Hermes and Reason, 2008). Several observational studies have concluded that the SST variability here is predominantly generated by variability in surface heat fluxes (Jayakumar et al., 2011; Vialard et al., 2008), while Drushka et al. (2012)  suggest this finding applies across most of the tropical Indian Ocean. Such studies, however, typically exhibit large uncertainty in the subsurface ocean terms. The shallow thermocline and strong high frequency winds in the SCTR region enhance near-inertial waves and lead to strong mixing at the base of the mixed layer as well as in the thermocline (e.g. Cuypers et al. 2013; Sabu et al. 2021). Modelling studies have shown that ocean dynamics play an important role in generating SST variability (Halkides et al., 2015; Han et al., 2007). For example, Fig. 7 from the study of Halkides et al. (2015) shows the relative contribution of modelled ocean dynamical processes and thermodynamical processes (i.e., surface heat fluxes) in forcing intraseasonal SST variability. Fig. 7a shows that ocean dynamical processes (green shading), including horizontal and vertical advection, are the dominant source of intraseasonal SST variability on the equator and in upwelling regions off Indonesia, Sri Lanka and along the western boundary. The ocean dynamical processes are in turn dominated by horizontal advection along the equator and tropical coastlines (Fig. 7b, pink shading), and vertical advection (blue shading) in the off-equatorial ocean interior.

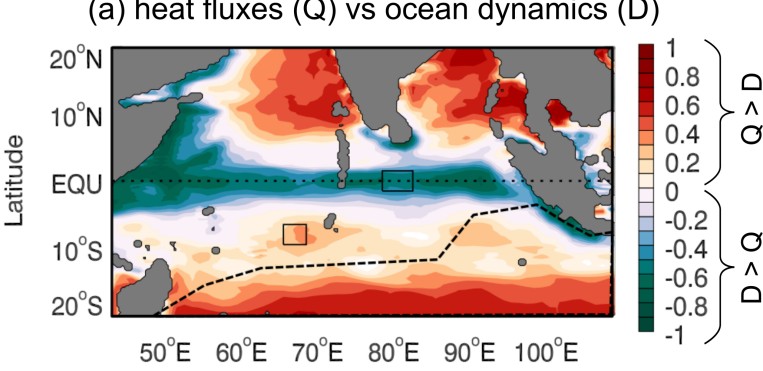

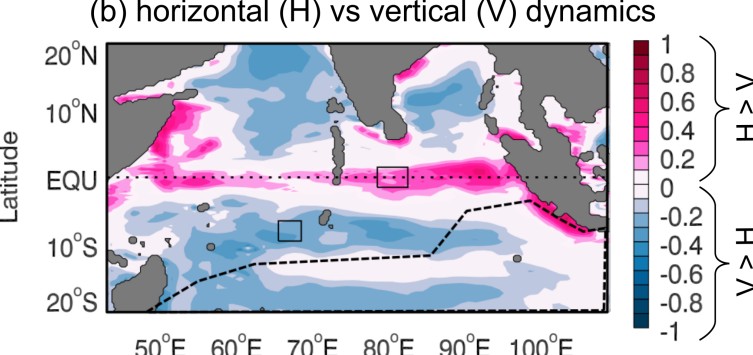


**Figure 7: Modelled balance of processes driving intraseasonal SST variability. (a) Relative role of heat fluxes (Q)**
**and ocean dynamics (D) in driving SST variability, with red (green) colours implying Q (D) dominates forcing, (b)**
**Relative role of horizontal (H) and vertical (V) processes in the dynamical forcing, with pink (blue) colours implying**
**that H (V) processes dominate. All fields are derived from the ECCO-JPL ocean state estimate. The dotted line**
**marks the Equator, dashed line in the southern hemisphere outlines a region in which the model does not fully**
**resolve the ocean heat budget , and the boxes on the Equator and at 10°S mark regions for further analysis not**
**described here. Modified from Halkides et al. (2015).**
Organisation of chlorophyll a at intraseasonal timescales has also been reported, with model studies indicating potential
biophysical feedbacks due to the variability of penetrative radiation into the water column (Waliser et al. 2005, Jin et al.
2013a; Giddings et al.,2021). In the Bay of Bengal, the proportion of incoming solar radiation absorbed within the mixed
layer varies between 60% and 97% due to a combination of variability in chlorophyll a concentration and mixed layer
depth (Lotliker et al., 2016) and an increase in chlorophyll of 0.3 mg/m$^3$ can lead to SST increase of up to 0.35°C on
intraseasonal time scales (Giddings et al., 2021). Representing the seasonal cycle of chlorophyll a concentration in the
Arabian Sea in a coupled model led to substantial changes in the simulated SST and monsoon rainfall over India (Turner
et al., 2012), suggesting that incorporating this process into coupled models may be important to improve simulation of
monsoon rainfall and circulation around the Indian Ocean.
Figure 8 illustrates propagation of surface patterns in an MJO composite constructed by Jin et al. (2013a). In each panel
the peak in outgoing longwave radiation (OLR, a proxy for convection) is indicated by a red diagonal line. The MJO
generates substantial surface heat flux anomalies that create a pattern of surface heat fluxes and SST anomalies such that
warm (cool) SSTs lead enhanced (reduced) convection by a quarter of a phase (e.g., Shinoda et al., 1998). The MJO also
leads to low-frequency rectifications in the mean state of physical and ecosystem responses (Fig. 8, Waliser et al. 2003,
Jin et al., 2013a,b).

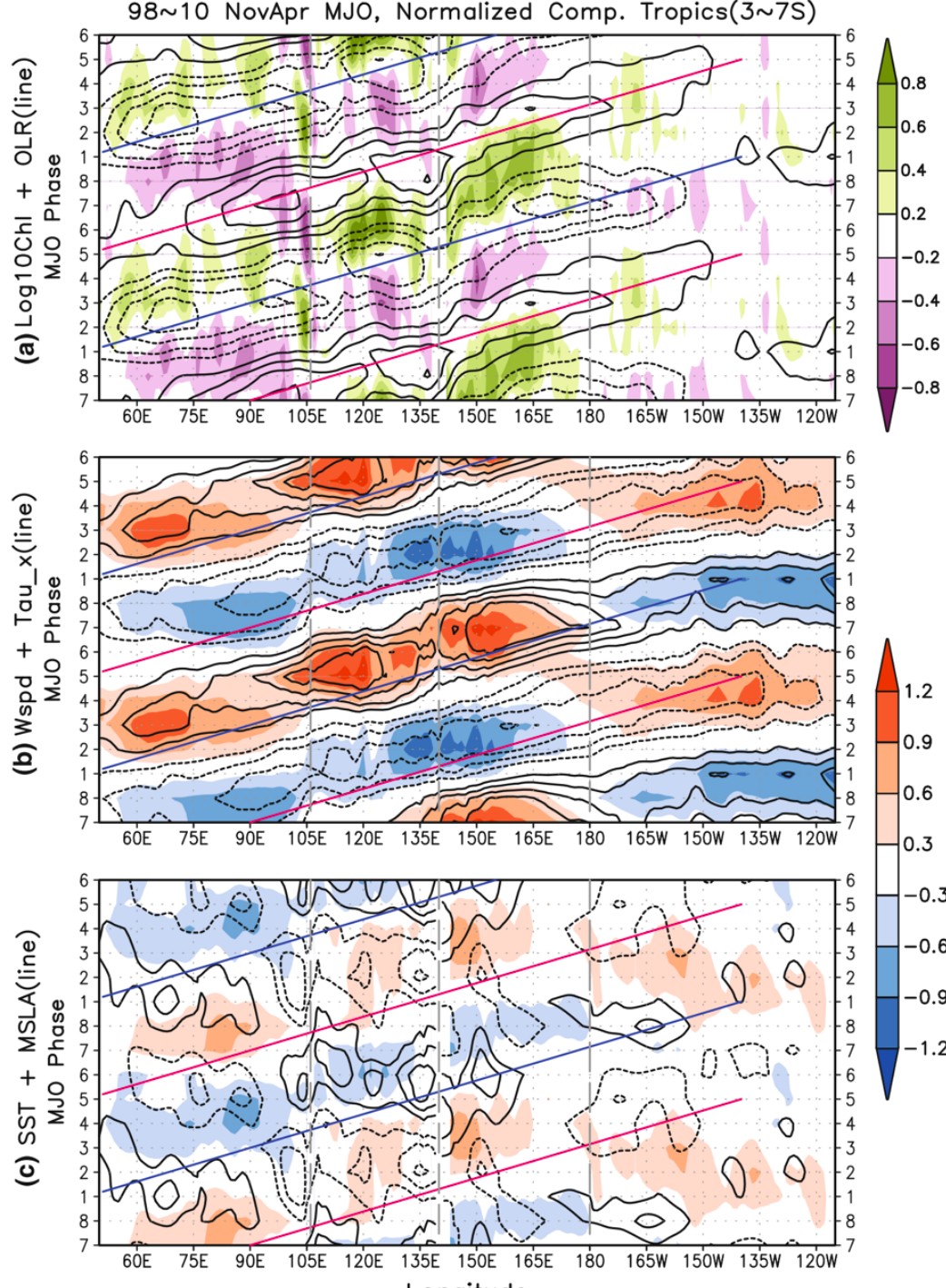

98~10 NovApr MJO, Normalized Comp. Tropics(3~7S)


**Figure 8: MJO composite evolution for the Boreal winter (Nov-Apr) averaged over latitudes 3°–7°S for the period of November 1st, 1997 to October 31st, 2010, of a) Log$_{10}$Chl from SeaWIFS satellite observations (shaded) and satellite-derived outgoing longwave radiation (contour), b) wind speed (shaded) and zonal wind stress (contour), both from the cross-calibrated multi-platform (CCMP) dataset, and c) NOAA-OI satellite SST anomalies (shaded) and AVISO mean sea level anomaly (contour). All contour intervals match shading levels in (c), and solid (dash) line indicates positive (negative) values. All variables are normalised, and the same MJO composite is repeated for two cycles for convenience. There are between 127-227 events in the composite for each MJO phase. Red diagonal lines indicate peak signals of positive OLR, and blue lines indicate negative OLR peak, so these are guides for the MJO propagation. The relative location of each propagation line in all panels is the same. Left and center gray vertical dashed line indicates the western and eastern boundary of the Maritime Continent, and the right gray line is on the Dateline where chlorophyll a propagation stops. From Jin et al. (2013a).**

### 3.2.2 Monsoon Intraseasonal Oscillation - MISO

While the MJO dominates intraseasonal variability during October to April, during May to September (boreal summer, southwest monsoon), the Monsoon Intraseasonal Oscillation (MISO; Goswami, 2012; Suhas et al., 2013) dominates. The dominant timescale for MISOs is 30-60 days but MISOs can also occur on 10-20 day time scales (Goswami et al, 2016) and there are studies that have identified a 3-7 day time scale for MISOs (e.g. Roman-Stork et al, 2020). MISOs can be seen as low pressure systems laden with moisture which deliver rain from atmospheric instabilities (Fig. 9). The MISO is also known as the Boreal Summer Intraseasonal Oscillation (BSISO; Lau and Waliser, 2012; Lee et al., 2013). The MISO oscillations are dynamically linked to the equatorial MJO (e.g., Sperber and Annamalai, 2008), but exhibit northeastward and northwestward propagating features, with the main centre of action being the Bay of Bengal. These northward-propagating bands of enhanced and reduced rainfall exhibit a similar relationship with SST to the MJO: warm SST leading increased rainfall (cool SST leading reduced rainfall) that then determine the wet/dry (or active/break) cycles of the South Asian monsoon (Vecchi and Harrison, 2002; Roxy et al., 2013; Suhas et al., 2013; Zhang et al., 2018). These SST anomalies are primarily forced by variations in surface heat fluxes in the Bay of Bengal (Girishkumar et al., 2017; Vialard et al., 2012), while variations in wind-induced mixing, Ekman pumping and entrainment drive SST variability in the Arabian Sea (Duncan and Han, 2012; Vialard et al., 2012).

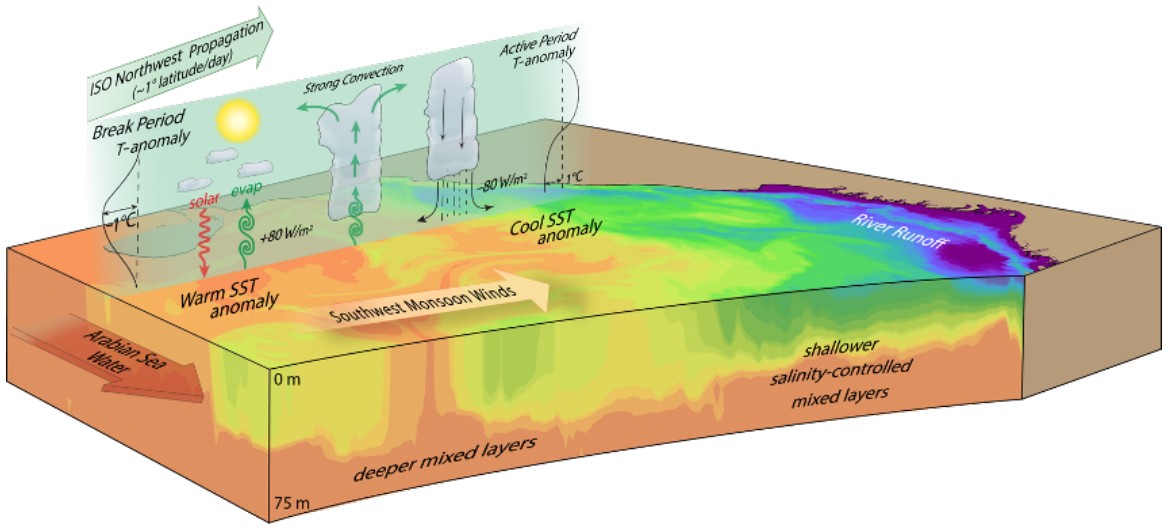

**Figure 9: A schematic of the Monsoon Intraseasonal Oscillation (MISO) in the Bay of Bengal, showing the coupled ocean-atmosphere 30–60 day mode northwestward propagation and associated processes in the atmosphere and the ocean. (From Mahadevan et al., 2016a).**

Simulations of the MISO are still generally poor in state-of-the-art coupled models (e.g., Goswami et al., 2013; Jayakumar et al., 2017; Sabeerali et al., 2013; Sharmila et al., 2013) and re-analysis products (e.g. Sanchez-Franks et al., 2018). Evidence exists from observations of low-level convergence and OLR, as well as from forced atmospheric and coupled ocean-atmosphere model experiments, that both MJOs and MISOs are phenomena that require coupling between the ocean and atmosphere to exist. This is even though the scales of SST anomalies tend to be an order of magnitude smaller than the scales of the propagating atmospheric systems (Waliser et al., 1999; Zhou and Murtugudde, 2009). Including air-sea coupling in simulations of the MISO has been identified as key to improving simulation of this oscillation in some models (e.g., Jayakumar et al., 2017; Li et al., 2018; Roxy et al., 2013; Sharmila et al., 2013), and has been shown to improve aspects of simulation in others (e.g., Bellon et al., 2008; Peatman and Klingaman, 2018).

While new theories continue to be proposed for MJOs (e.g., Wang et al., 2016), MISOs have not received similar attention likely due to their more local nature compared to the global impacts of MJOs (e.g. their impact on ENSO). The mechanism that causes the northward propagation of the MISO is still a topic of research. The most recent theory for MISOs proposed by Zhou et al. (2017a, b) invokes an explicit coupling between the ocean and the atmosphere in a so-called Central Indian Ocean mode. Zonal winds at intraseasonal timescales over the Indian Ocean are argued to be coupled to SSTs to produce a barotropic instability in the meridional gradient of the zonal winds. The horizontal atmospheric eddy fluxes generated by the barotropic instability are invoked to explain the northward propagation and the advection of momentum and moisture as a coupled phenomenon. Key questions remain about the oceanic and air-sea interaction processes that

reorganise the SSTs in the Central Indian Ocean mode as well as the respective roles of the vertical and horizontal shears
in driving northward propagation of MISOs.
Observations and models indicate that MISOs may be slowing down because of the warming in the Indian Ocean
(Sabeerali et al., 2013), which needs to be understood better for providing reliable monsoon predictions and projections in
this climate vulnerable region. This is underscored by the observational evidence that climate variability and change are
increasing the frequency of dry spells and the intensity of wet spells in the Indian summer monsoon, which are directly
related to MISO (Singh et al., 2014).
**3.2.3 Intraseasonal drivers of heavy rainfall**
As the MJO season begins to wind down in April, northward propagating MISOs begin to become dominant in the northern
Indian Ocean, north of around 5°N. While the southwesterlies produce some of the strongest coastal upwelling off Somalia
and cool the Arabian Sea, the Bay of Bengal remains warm and largely above the convective threshold (28°C) owing to
the freshwater input from rainfall as well as rivers discharging into the Bay (Roxy and  Tanimoto, 2007). The freshwater
input creates a shallow density stratification (barrier layer) within the temperature mixed layer and thereby weakens the
upwelling of cold water from the thermocline. MISOs deliver rain from atmospheric instabilities, but what controls the
rainfall at intraseasonal timescales during the summer can be expected to be region specific with moisture supply
determining the rainfall variability over land (Pathak et al., 2017).
Over the ocean, the largely evaporative Arabian Sea is relatively cool but the southwesterlies begin to slow down as they
approach the Western Ghats mountain range on the west coast of India, leading to maximum rainfall there during the
boreal summer monsoon season (Xi et al., 2015). Rather counterintuitively, the warm SST in the Bay of Bengal remains
above the convective threshold (Gadgil et al., 1984; Roxy, 2013) and yet, the ocean is not in direct control of the
intraseasonal rainfall events. Once the SSTs are warm enough to support atmospheric convection, it is baroclinic
instabilities, and not static instabilities induced by warm SSTs, that drive the majority of rainfall over the Bay of Bengal
(Xi et al., 2015).
**3.3 Ocean internal variability impacts on air-sea interaction**
Mesoscale eddies are ubiquitous in the ocean. In the tropical Indian Ocean, however, linear dynamics dominate and the
impacts of eddies are (or seem) small. While the Indian Ocean has the largest SST variability occurring at seasonal
timescales,  strong mesoscale variability is also observed along the Somali coast where the western boundary current
crosses the Equator. The slope of the East African coastline and the equatorial crossing of the low-latitude jet produce
multiple eddies (Nof and Olson, 1993), which are shown to generate strong air-sea coupling at mesoscales (Schott and
McCreary, 2001; Schott et al., 2009; Vecchi et al., 2004;  Seo et al., 2008). Some intraseasonal oscillations in the ocean

were reported in the southwestern tropical Indian Ocean (Kindle and Thompson, 1989) but generally, the impact of ocean internal variability on SSTs in the tropical Indian Ocean has not been widely studied. At the eastern boundary of the subtropical Indian Ocean, instability of the poleward Leeuwin Current generates a rich field of mesoscale eddies that carry heat into the Indian Ocean interior, contributing to air-sea exchange of heat and the oceanic interior poleward heat transport (Domingues et al. 2006, Feng et al., 2007; Dilmahamod et al. 2018). In the subtropical southeast Indian Ocean, mesoscale eddies, and possibly annual and semiannual Rossby waves propagating from the eastern boundary, were found to influence the seasonal variation of the surface layer heat balance through horizontal advection (Cyriac et al. 2019).

Low-frequency internal variability is also possible. Jochum and Murtugudde (2004) performed forced ocean model experiments with climatological forcing alone to demonstrate that significant low-frequency variability at interannual timescales is generated in the Indian Ocean by mesoscale eddies and other types of nonlinearity. The role of internal variability in regional coupled climate variability as well as ecosystem and biogeochemistry remain interesting problems for this already warm ocean, in which even small SST anomalies can be important for generating large-scale ocean atmosphere interactions (Palmer and Mansfield, 1994).

## 4 Upper Ocean Circulation and Biogeochemical Variability

### 4.1 Overview

The near surface circulation in the Indian Ocean consists of the monsoon-dominated, seasonally reversing currents north of around 10°S, and the steady currents to the south, as illustrated in  Fig. 10a for the southwest monsoon (July-August) and Fig. 10b for the northeast monsoon (January-February). This figure has been updated from Talley et al. (2011) to recognise recent advances in understanding of circulation patterns. In the northern Indian Ocean, additions are a revision of the Red Sea circulation (Menezes et al., 2019). In the southern Indian Ocean, moving in an anti-clockwise direction from the Maritime Continent, additions are: 1) seasonally reversing flows in the Java Sea; 2) the Holloway Current along Australia's Northwest Shelf (Holloway and Nye, 1985; Holloway, 1995; Brahmanpour et al., 2016);  3) revised position of the salinity-driven Eastern Gyral Current that flows eastward from around 90°E along approximately 15°S, recirculating Indonesian Throughflow Water from the South Equatorial Current and supplying the poleward-flowing Leeuwin Current (Meyers et al., 1995; Domingues et al., 2007; Menezes et al., 2013, 2014); 4) the near-surface South Indian Countercurrent with 3 distinct branches, northern, central and southern, flowing from the southern tip of Madagascar to Australia where they merge with the poleward-flowing Leeuwin Current  (Menezes et al., 2014 and references therein); and 5) the splitting of the Flinders Current near 110°E, with one branch recirculating back toward Australia, and the other a westward continuation of the Flinders Current, previously not shown  (Duran et al., 2020).

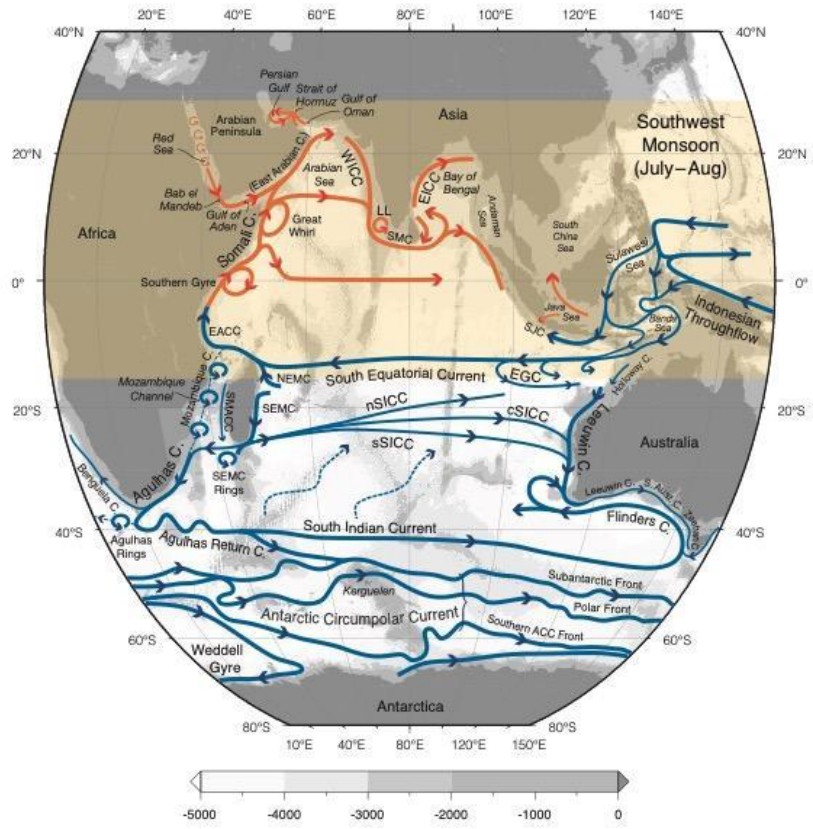

**Figure 10a: Schematic near-surface circulation during the Southwest Monsoon (July-August). Blue: year-round mean flows with no seasonal reversals. Orange: monsoonally reversing circulation (after Schott & McCreary, 2001). The ACC fronts are taken directly from Orsi, Whitworth, and Nowlin (1995). Acronyms: EACC, East African Coastal Current; NEMC, Northeast Madagascar Current; SEMC, Southeast Madagascar Current; SMACC, Southwest MAdagascar Coastal Current; WICC, West Indian Coastal Current; EICC, East Indian Coastal Current; LH and LL, Lakshwadeep high and low; SJC, South Java Current; EGC, Eastern Gyral Current; SICC, South Indian Countercurrent (south, central and southern branches); NEC, Northeast Monsoon Current. Updated from Talley et al. (2011), originally based on Schott and McCreary (2001). The light gray shading shows seafloor bathymetry.**

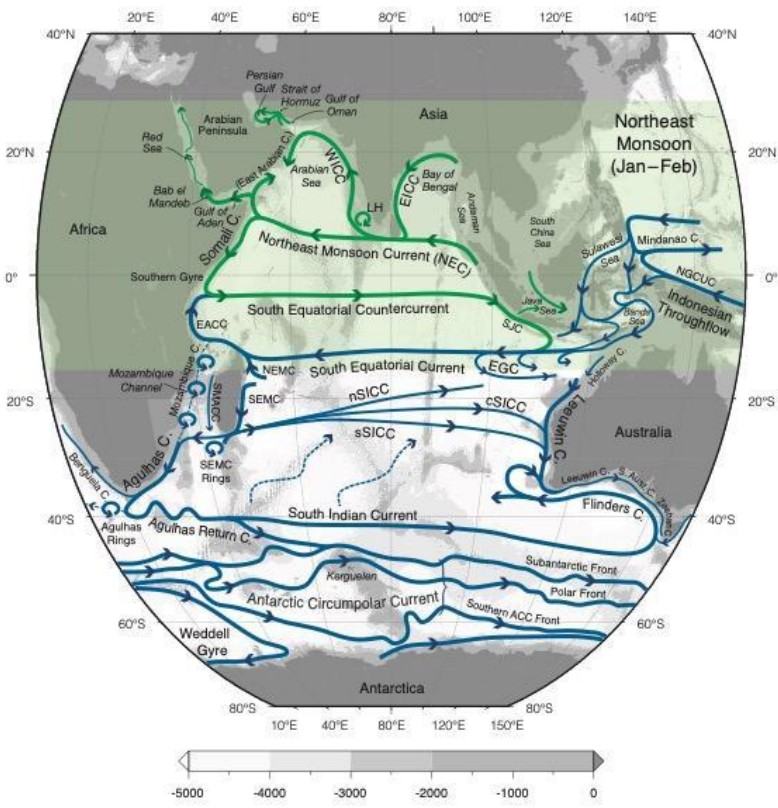

**Figure 10b: Schematic near-surface circulation during the Northeast Monsoon (January-February). Details as for Fig. 10a.**

The intermediate and deep circulation and overturning cells will not be examined in this synthesis. The reader is referred to Talley et al. (2011) and references therein, and in addition, Nagura and McPhaden (2018) who used Argo and CTD data to map out the circulation and water masses in density classes associated with the shallow overturning circulation, with emphasis on the southern hemisphere. There has been some progress on understanding circulation at intermediate and deeper depths in the equatorial band, which is summarised in Huang et al (2018).

### 4.2 Southern Indian Ocean

### 4.2.1 South Equatorial Current

The South Equatorial Current (SEC), the northern limb of the southern Indian Ocean subtropical gyre, carries Indonesian Throughflow (ITF) waters into the interior Indian Ocean, flowing westward between 10–20°S (Fig. 10a and 10b). Upon

reaching the northern tip of eastern Madagascar, it bifurcates and supplies the Northeast Madagascar Current (NEMC;
Schott and McCreary, 2001; Song et al., 2004; Valsala and Ikeda, 2007) and the Southeast Madagascar Current (SEMC)
and contributes to the development of Mozambique Channel eddies. The mean flow through the Mozambique Channel is
weak (Song et al., 2004), although there is an indication from ocean model results that the eddy-dominated flow contributes
on the order of 20 Sv southward (Durgadoo et al., 2013). The Mozambique Channel eddies, eddies from the SEMC and
recirculation combine to feed into the Agulhas Current (Schott and McCreary, 2001).
Between 50 and 80ºE the SEC is coincident with the southern half of the Seychelles-Chagos Thermocline Ridge (SCTR,
Vialard et al., 2009).  The SCTR is characterized by a relatively shallow thermocline and thin mixed layer (~30m) across
the southern tropical Indian Ocean in the latitude band 5-15ºS.  Between 50 and 80ºE the SCTR/SEC is a region of
significant upwelling (Hermes and Reason, 2008; Vialard et al., 2009; Resplandy et al., 2009; Dilmahamod, 2014), which
affects biogeochemistry, and even fisheries (Resplandy et al., 2009; Robinson et al., 2010; Dilmahamod, 2014).
In the eastern IO, the intraseasonal variation of the SEC is mostly attributed to the baroclinic instability of the mean current
(Feng and Wijffels, 2002), which is important for the meridional heat transport in the region and contributes to the demise
of Indian Ocean Dipole events (Ogata and Masumoto, 2011; Yang et al. 2015). Barotropic instability of the SEC has also
been proposed to be a key mechanism for generating intraseasonal variability (Yu and Potemra, 2006). These intraseasonal
signals propagate westward as Rossby Waves, influencing the SEC variability in the western Indian Ocean (Zhou and
Murtugudde, 2008).
Interannual variability in the ITF due to ENSO, IOD and other influences is communicated into the interior Indian Ocean
along the SEC and via Kelvin and Rossby waves (Godfrey, 1989, 1996; Meyers et al., 1995; Meyers, 1996; Wijffels and
Meyers, 2004). Pressure anomalies associated with ENSO and IOD are communicated through the Indonesian seas as
Kelvin and Rossby waves. These anomalies propagate westward into the Indian Ocean as Rossby waves. At the same time
the pressure anomalies drive variations in ITF and SEC transport and induce temperature/salinity variability via advection.
Geostrophic transport variability in the long-time repeat XBT line IX1 shows that the SEC is stronger during La Niña and
positive Indian Ocean Dipole events (Meyers, 1996; Liu et al., 2015). Similarly, the Pacific Decadal Oscillation alters the
SEC and ITF transports and associated water properties (Section 6.1). During the climate change hiatus period of 2000-
2011, the enhanced heat transport of the SEC/ITF was a key mechanism for the fast warming trend in the southern
subtropical Indian Ocean (Section 6.1).

## 4.2.2 Western Boundary

The Agulhas Current (Fig. 10) has long been known as one of the strongest western boundary currents in the global oceans, with an average transport of 75 Sverdrups and current speeds in excess of 2 m s$^{-1}$ (Beal et al., 2015; Beal et al., 2011). The Agulhas Current plays a vital role in the global thermohaline circulation, advecting warm, salty, subtropical water southwards, following the continental shelf of South Africa and meandering less than 150 km offshore (Gründlingh, 1983; Lutjeharms 2006). The strength and warmth of the Agulhas Current influences atmospheric storm tracks and storm development. The large moisture source of the warm Agulhas Current region contributes significantly to the frequency and strength of African precipitation, which significantly impacts rain-fed subsistence farming (Hermes et al. 2019 and references therein).

South of the tip of Africa, the Agulhas Current retroflects eastwards into the South Indian Ocean (Fig. 10). This retroflection area is highly variable, occluding rings that propagate into the South Atlantic Ocean. The Agulhas variability is linked upstream to modes of variability including ENSO (Elipot and Beal, 2018, Trott et al., 2021) and downstream with the Atlantic meridional overturning circulation, providing an essential link between the Pacific, Indian and Atlantic Oceans (Beal et al., 2011). Estimates of the rate of mass and heat exchange carried by Agulhas leakage south of Africa (and the number of rings shed per year) vary and are difficult to verify reliably (Weijer et al., 2014). Daher et al (2020) recently used a combination of drifters and Argo floats to derive an estimate of Agulhas leakage of 20 Sv. van Sebille et al. (2011) and le Bars et al. (2014) suggested upstream variability of the Agulhas Current has an effect on inter-ocean exchange between the South Indian and South Atlantic oceans, primarily by influencing the frequency of ring shedding at the Agulhas retroflection. However, a few recent papers suggest instead that its variability is driven by the Southern Hemisphere Westerlies (Durgadoo et al, 2013; Loveday et al., 2014; Elipot and Beal, 2015).

The Agulhas Current has a seasonal cycle and is strongest in summer (Krug and Tournadre, 2012; Beal and Elipot, 2016) and tied to a baroclinic adjustment of near-field winds (Hutchinson et al, 2018). Seasonal changes in the Agulhas retroflection region (Lutjeharms and van Ballegooyen, 1988; Quartly and Srokosz, 1993) and in the southwest Indian Ocean (Ffield et al., 1997) have been suggested from hydrographic and satellite data (Krug et al., 2012), but with weak statistical significance due to a lack of sufficiently long time series.

Although long term observations in this region are limited there are numerous recent studies that have further elucidated our understanding of the Agulhas Current. Beal and Elipot (2016) used 3 years of in situ data to show that, contrary to expectations, the Agulhas Current has not intensified since the early 1990s. Instead, it has broadened as a result of more eddy activity, driven by intensifying winds. Variability in the path and strength of the Agulhas Current has mostly been attributed to solitary Agulhas meanders within the Current system (also known as Natal pulses) which drive upwelling

and cross-shelf transports, affecting marine productivity, fisheries and recruitment over the Agulhas Bank (Beal and
Bryden, 1999; Roberts et al., 2010, Elipot and Beal, 2015). Recent work has highlighted the importance of submesoscale
eddies in the Agulhas Current frontal region driving an inshore edge flow reversal which can have important consequences
on fisheries (Krug et al., 2017).
The advance in models has also helped improve our understanding of the Agulhas Current, which is generally not well
represented in global ocean models. Hutchinson et al. (2018) used idealized models to expose a link between the
seasonality of the Agulhas Current and propagation of first baroclinic mode Rossby waves communicating the wind stress
signal across the western portion of the Southern Indian Ocean, with the signal from winds further east having little effect.
### 4.2.3 Interior flows
In the central-eastern South Indian Ocean between 20°S and 30°S, the surface geostrophic flow is generally eastward,
opposite to the prediction of both the Ekman and Sverdrup theories (Sharma 1976; Sharma et al., 1978; Godfrey and
Ridgway, 1985; Schott et al., 2009).  This flow is driven by the large-scale, poleward drop in the dynamic height (steric
height) near the sea surface (Godfrey and Ridgway, 1985; Schott et al., 2009) related to the meridional transition from the
very fresh and warm SEC waters to the increasingly cooler, saltier and denser waters to the south. The flow generally
extends from the sea surface to ~200–300 m (Domingues et al., 2007; Palastanga et al., 2007; Divakaran and Brassington,
2011; Menezes et al., 2014). The mechanisms that determine the vertical extent of the interior eastward flow remains
unclear, although this depth coincides with the depth of the shelf break at the eastern boundary and the bottom of the
Leeuwin Current along that boundary. This correspondence may be achieved by the westward propagation of baroclinic
Rossby waves (Weaver and Middleton, 1989; Furue et al., 2013). Below the near-surface eastward flows,  the flow is
weakly westward (Domingues et al., 2007; Schott et al., 2009; Furue et al., 2017).
Embedded in this general eastward flow are narrower eastward jets (Maximenko et al., 2009; Divakaran and Brassington,
2011; Menezes et al., 2014), collectively known as the South Indian (Ocean) Countercurrent (SICC; Palastanga et al. 2007;
Siedler et al. 2006; Menezes et al., 2014).  They start out as a single jet emanating from the southern tip of Madagascar
around 25°S, possibly fed by a partial retroflection of the SEMC (Palastanga et al., 2007; Siedler et al., 2006, 2009) and
divide into separate jets around the Central Indian Ridge (65°E–68°E) (Menezes et al., 2014). Eastward flows exist in
similar latitude bands in the North and South Pacific and North and South Atlantic (Yoshida and Kidokoro, 1967; Merle
et al., 1969; Takeuchi, 1984; Kubokawa, 1999; Qiu and Chen, 2004; Kobashi and Kubokawa, 2012). However, the jets in
these basins are weaker and shallower than the SICCs and do not extend all the way to the eastern boundary (Menezes,
2015).

Three main jets (Fig. 10a) are evident in geostrophic velocity calculated from both altimetric sea surface height and hydrography and are captured in OGCMs (Maximenko et al., 2009; Divakaran and Brassington, 2011; Menezes et al., 2014). The stronger southern jet (3–4 Sv) crosses the basin around 26°S and has an associated thermal front at depths around 100–200 m (Sharma 1976; Siedler et al., 2006; Menezes et al., 2014; Palastanga et al., 2007). This front suggests that the southern SICC has physics similar to the Subtropical Countercurrents (STCCs) of the Pacific Ocean (Kubokawa, 1999; Kobashi and Kubokawa, 2012, Menezes et al., 2014). The location and strength of the SICC vary between studies, from well-defined jets (Siedler et al. 2006, Palastanga et al. 2007, Divakaran and Brassington 2011, Menezes et al. 2014) to a mean velocity structure (Jia et al., 2011a), or even absence of the SICC (Srokosz et al. 2015). Depending on the region and time in which its characteristics were determined, the SICC varies from a weak mean current of 2–3 cm/s (Jia et al., 2011a) to a strong jet of 50 cm/s eastward flow (Siedler et al., 2006).

The eastward flowing Eastern Gyral Current (EGC) is part of an anticyclonic recirculation centred at the Indonesian-Australian basin (5°S–20°S and 100°E–125°E) (Domingues et al., 2007; Menezes et al., 2013, and references therein). Part of the northern SICC merges with the EGC around 15°S, 100°E (Fig. 10a). The EGC supplies ITF-origin water to the Leeuwin Current (LC) and is an essential component of the LC dynamics (Domingues et al., 2007; Benthuysen et al., 2014; Lambert et al., 2016; Furue et al., 2013, 2017; Yit Sen Bull and van Sebille, 2016). The geostrophic flow of the EGC is controlled by the meridional salinity gradient, making its dynamics distinct from the temperature dominated SICC (Menezes et al., 2013). This salinity front is formed by the encounter of the fresh Indonesian Throughflow Water carried westward by the SEC and the salty subtropical underwater formed at the Southern Indian Ocean subtropical salinity maximum. The seasonal cycles of the EGC and the SICC are also distinct: the EGC is stronger in austral winter (3–5 Sv) and weaker (<0.5 Sv) in summer with the cycle in phase with the Leeuwin Current (Feng et al., 2003; Menezes et al., 2013; Furue et al., 2017). The SICC is overall stronger in spring-summer and weaker in winter (Palastanga et al., 2007; Jia et al., 2011a; Menezes et al., 2014) and experiences strong interannual variability, which peaks at biennial timescales and is decadally modulated (Menezes et al., 2016).

The multiple jets of the SICC are embedded in a zone of high eddy kinetic energy, with eddies generated by instabilities of the Leeuwin Current and of the SICC itself (Palastanga et al., 2007; Divakaran and Brassington, 2011; Huhn et al., 2012; Jia et al., 2011a, 2011b; Menezes et al., 2014, 2016; Siedler et al., 2006). By co-locating Argo floats and satellite data, Dilmahamod et al. (2018) described the passage of surface and subsurface South Indian Ocean eddies (SIDDIES). These westward-propagating, long-lived features (>3 months) originate in areas of high evaporation in the eastern Indian Ocean and prevail over a preferential latitude band, forming a permanent structure linking the eastern to the western Indian Ocean (the "SIDDIES Corridor"). This corridor of eddy passage allows the advection of water masses and biogeochemical properties across the basin (Dilmahamod et al., 2018).

### 4.2.4 Eastern Boundary

Unlike any other eastern boundary current, the Leeuwin Current (LC; Figs. 10 and 11a) flows poleward, along the shelf break of the west coast of Australia (Smith et al., 1991). Figure 11a presents the long-term average volume transport of the LC System from an observational climatology with similar structure found in a 1/10° ocean general circulation model (Furue et al., 2017). The primary source waters for the LC are the interior eastward flows (Section 4.2.3) that turn southeastward as they approach the coast and merge with the LC (Fig. 11a; Domingues et al., 2007; D'Adamo et al., 2009; Menezes et al., 2013, 2014; Furue et al., 2017, 2019). On average, the LC carries 0.3 Sv southward at 22°S, gains 4.7 Sv from the Indian Ocean interior, loses 3.5 Sv through downwelling to the layer beneath, and carries 1.5 Sv at its southern limit. The LC is approximately 200–300 m deep, extends from 22°S (North West Cape) to 34°S (Cape Leeuwin) and exists throughout the year despite significant seasonality (Feng et al., 2003; Ridgway and Godfrey, 2015; Furue et al., 2017). The Holloway Current, which flows southwestward on the North West Shelf (D'Adamo et al., 2009; Bahmanpour et al., 2016), is another weaker source to the LC from the north. Inshore of the LC, there exist seasonal equatorward flows that recirculate waters of distinct watermass properties influenced by air-sea interaction over the continental shelf (Woo et al., 2006).

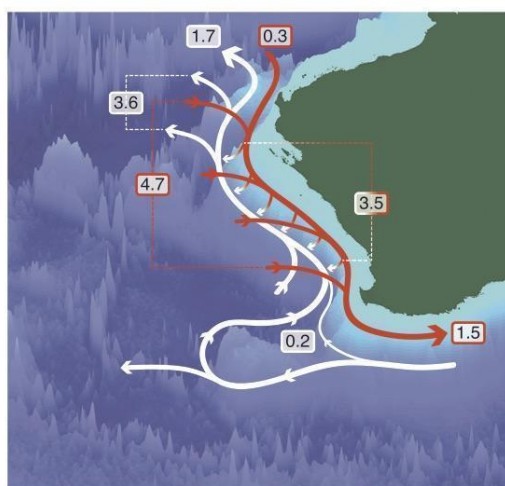

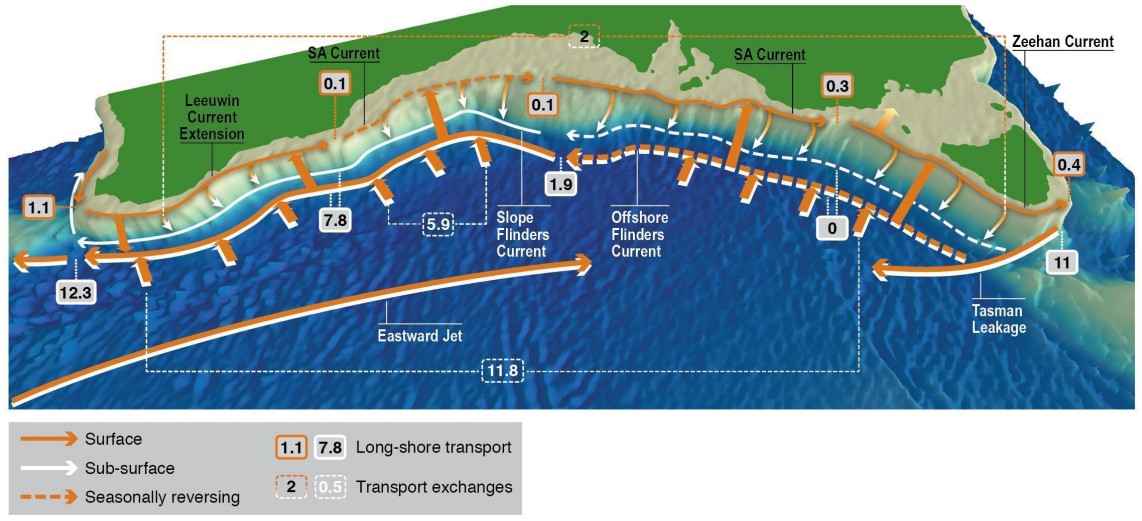

**Figure 11: a) Schematic summary of Australia's Leeuwin Current System three-dimensional transports (Sv). The red arrows and red-outline numbers represent the upper-layer (0–200 m) meridional transport of the poleward Leeuwin Current and meridionally-integrated zonal transport of the shallow eastward flows. The white arrows represent the lower-layer (200–900 m) flows of the Leeuwin Undercurrent. Taken from Furue et al. (2017). © American Meteorological Society. Used with permission. b) Schematic summary of the Southern Australia Current System three-dimensional transports (Sv). Long-shore transport for the Shelf Break Currents and Flinders Current in grey box with orange and white outlines, respectively. Integrated vertical and onshore flow transport in dashed outline box. Reprinted from Duran et al. (2020), with permission from Elsevier, Progress in Oceanography. Both schematics are based on a geostrophic calculation in the CARS ⅛-degree climatology.**

The mean state of the LC is driven by the meridional pressure gradient in the upper ocean (e.g., Godfrey and Ridgway, 1985; Godfrey and Weaver, 1989, 1991), evident as a large poleward decrease in SSH balanced by an eastward surface geostrophic current (Section 4.2.3.1). The eastward flow approaches the eastern boundary, inducing downwelling and a surface poleward current (Fig. 11; Godfrey and Ridgway, 1985; McCreary et al., 1986; Thompson, 1987; Weaver and Middleton, 1989, 1990; Furue et al., 2013; Benthuysen et al., 2014), in opposition to the prevailing southerly winds. As a poleward boundary current, the LC waters are relatively fresh and warm from tropical origins (Rochford, 1969; Andrews, 1977; Legeckis and Cresswell 1981; Domingues et al., 2007; Woo and Pattiaratchi, 2008). Saltier Indian Central Water, the surface water of the Subtropical South Indian Ocean, joins the LC as it flows poleward (Section 4.2.3) increasing the mean density of the LC. Surface cooling along the poleward path also contributes to the increase in density (Woo and Pattiaratchi, 2008; Furue, 2019).

The LC flows around the southwestern corner of Australia and continues to flow eastward along the shelf break of the
south coast of Australia to reach the southern tip of Tasmania near 42°S, 140°E (Fig. 11b, Oliver et al., 2016; Oke et al.,
2018; Duran et al., 2020). This 5500-km long boundary current was first documented as a continuous flow by Ridgway
and Condie (2004). When the longshore current is weak, however, it tends to be somewhat fragmentary (Oke et al., 2018;
Duran et al., 2020) and sometimes even reverses in places (Duran et al., 2020). For this reason, and additionally because
of the scarcity of observational sampling, the current is not traditionally regarded as a single current. Along southern
Australia, the boundary currents can be described following Ridgway and Condie's (2004) naming convention. The
current's western sector is called the Leeuwin Current Extension, the central part, to the south of the Great Australian
Bight, is called the South Australian Current, and the easternmost part along Tasmania is called the Zeehan Current. They
are collectively known as the Shelf-Break Currents (SBCs) of the Southern Australia Current System (Duran et al., 2020).
It is not clear whether the SBCs along the south coast of Australia are, dynamically, an extension of the LC. The SBCs
are at least consistent with the local northward Ekman drift (Ridgway and Condie, 2004; Duran et al., 2020) and hence
would exist without the LC.

On seasonal timescales, the LC transport generally tends to be strongest in austral autumn and weakest in austral summer
(McCreary et al., 1986; Smith et al., 1991; Feng et al., 2003; Furue et al., 2017). There are two theories to explain this
seasonality. In one, the local winds, which generally induce an offshore Ekman drift and therefore tend to weaken the LC,
reach their annual maximum or minimum when the LC transport reaches its minimum or maximum, respectively
(McCreary et al., 1986; Furue et al., 2013). In the other, a seasonal pressure anomaly originates in the Gulf of Carpentaria
and propagates counterclockwise along the shelf break, driving the seasonality of the LC and of the SBCs to the south of
Australia (Ridgway and Godfrey, 2015). Like the LC, the SBCs tend to be strongest in austral autumn and weakest in
austral summer (Ridgway and Condie, 2004; Oke et al., 2018; Duran et al., 2020). In particular, the eastern part of the
South Australian Current is seen to reverse in summer (Duran et al., 2020). This variability is consistent with the
counterclockwise propagation of pressure anomaly shown by Ridgway and Godfrey (2015) and also with the seasonality
of the wind stress along the south coast of Australia, with onshore (offshore) Ekman drift tending to drive eastward
(westward) shelf-break flow (Duran et al. 2020).

On interannual time scales, the LC is modulated by the El Niño Southern Oscillation owing to the steric height anomalies
in the western equatorial Pacific Ocean propagating through the Indonesian Seas and along Western Australia (Feng et al.,
2003). During El Niño and La Niña periods, the LC transport weakens and strengthens, respectively, and is correlated with
Fremantle sea level (Feng et al., 2003). During the strong 2010–2011 La Niña event, the LC reached record strength speeds
(Feng et al., 2013) and the consequences of the unprecedented marine heat wave that resulted are described in Section 6.4.
On multidecadal timescales, the major boundary currents around Australia, including the LC, are reported to have
strengthened during 1979–2014 in an eddy-resolving OGCM, consistent also with observations (Feng et al., 2016; see

Section 6.1 for associated changes). At intraseasonal timescales, winds or heat anomalies on the North West Shelf region due to MJO events lead to intraseasonal variability of the Holloway Current on the North West Shelf and then of the LC (Marshall and Hendon, 2014; Marin and Feng, 2019).

The LC is accompanied by mesoscale eddies that cause the LC to meander energetically (Pearce and Griffiths, 1991; Feng et al., 2005; Waite et al., 2007; Meuleners et al., 2008). Those eddies are, at least partially, generated by barotropic, baroclinic, or mixed instability of the LC itself (Pearce and Griffiths, 1991; Feng et al., 2005; Meuleners et al., 2008). The eddy kinetic energy is greatest when the LC transport is strongest, in May–June ( Fang and Morrow, 2003; Feng et al., 2005). Some of these eddies cause a large meander of the LC: a large anti-cyclonic eddy often forms at 28°–29°S and at 31°–32°S (Feng et al., 2003; Feng et al., 2007) steering the LC offshore to return to the continental shelf further south. This state typically starts during May–June and ends in July–August (Feng et al., 2007). Similarly, it is suggested that the eastern part of the SBCs becomes unstable in boreal autumn and winter, generating eddies, which subsequently propagate westward south of Australia (Oke et al., 2018). Turbulent mixing has been found to be enhanced in anticyclonic eddies near the surface, and in cyclonic eddies at deeper levels (500-1000 m) due to the interaction of the eddies and near-inertial waves, which has implications for watermass modifications and the meridional overturning circulation (Cyriac et al. 2021).

Just below the Leeuwin Current is the equatorward Leeuwin Undercurrent (LUC; Thompson, 1984; Church et al., 1989; Smith et al., 1991; Fig. 11a). The LUC hugs the continental slope and extends from 200 m to 900 m (Furue et al., 2017). The LUC begins at Cape Leeuwin (34°S, 114°E) and is fed by a northward bend of a small fraction of the Flinders Current (FC; Fig. 10, 11; Furue et al., 2017). The remaining part of the FC continues westward but another small fraction of it appears to retroflect eastward and join and augment the LUC (Duran, 2015; Furue et al., 2017). Near 22°S, most of the LUC volume leaves the continental slope and flows offshore (Duran, 2015), apparently following the southern flank of the Exmouth Plateau although its bottom at 900 m is much shallower than the topographic feature (Fig. 11a).

To the south of Australia, an undercurrent has been recently identified below the Zeehan Current in a numerical simulation (Oke et al., 2018) and in a geostrophic calculation based on a gridded T–S climatology (Duran et al., 2020). Traditionally this flow was identified as a branch of the FC (Cirano and Middleton, 2004; Rosell-Fieschi et al., 2013; Feng et al., 2016) because the former flows in the same direction as the latter, but the FC as the northern boundary current of the subtropical gyre cannot exist on an eastern boundary (Anderson and Gill, 1975; Philander and Yoon, 1982; McCreary et al., 1992) and it lacks the vertical structure of an undercurrent (Duran et al., 2020). This northwestward- or westward-flowing undercurrent appears to exist all the way from the west coast of Tasmania to Cape Leeuwin (the southwestern tip of Australia) but its separation from the FC is less clear to the south of the Great Australian Bight and further west, where the FC accelerates and tends to overwhelm the undercurrent (Duran et al., 2020). Below, we call this current "slope FC" following Duran et al. (2020).

The mechanisms responsible for the LUC and undercurrent off southern Australia remain an open question, although
models have been developed to investigate potential processes. The linear, continuously stratified models of McCreary et
al. (1986) and Kundu and McCreary (1986) produce a surface poleward and a subsurface equatorward current, resembling
the LC and LUC, along the eastern boundary. This class of model, however, requires large vertical diffusivity to produce
a realistic LC and LUC (McCreary, 2013, personal communication). Along a continental slope, alongshore and cross-shelf
buoyancy advection cause a shelf break front, forming a surface intensified poleward current, like the LC, and an
equatorward undercurrent by thermal wind shear (Benthuysen et al., 2014). Analytical shelf models have been extended
to include cross-shelf buoyancy gradients to derive a poleward undercurrent like the LUC (Schloesser, 2014). These
process-based analytical theories have not been tested in an eddy-resolving model.
The LUC and the slope FC are connected to the LC and the SBCs, respectively, by downwelling (Fig. 11; Furue et al.,
2017; Duran et al., 2020), suggesting a common, but as yet unexplained, dynamics. Note, however, that for the LC–LUC
pair, the mean downwelling appears to occur along isopycnal surfaces, and hence the LC water mass is not found in the
LUC (Furue, 2019). For the SBCs and the slope FC, the nature of the downwelling is not known. The seasonality of these
undercurrents are not well known. No systematic seasonal variability of the LUC was evident in a hydrographic
climatology and ocean general circulation model (Furue et al., 2017).

### 4.2.5 Biogeochemical Variability

The ITF impacts both ocean currents and basin-scale biogeochemistry (Talley and Sprintall, 2005; George et al., 2013;
van Sebille et al., 2014). Talley and Sprintall (2005) mapped silicate on the 31.96 potential density surface, revealing a
striking silicate maximum associated with the SEC that extends westward to at least 60ºE, highlighting the broad reach of
ITF nutrient influence into the Indian Ocean.  Ayers et al. (2014) estimated the depth- and time-resolved nitrate, phosphate,
and silicate fluxes at the three main exit passages of the ITF that feed into the SEC: Lombok Strait, Ombai Strait, and
Timor Passage. They found that the nutrient flux is significant relative to basin wide new production, and that the majority
of ITF nutrient supply to the Indian Ocean via the SEC is to thermocline waters, where it is likely to support primary
production and significantly impact biogeochemical cycling.
Satellite chlorophyll and primary production estimates suggest that values in the SEC are considerably higher than those
found in the southern hemisphere subtropical gyre to the south, with Chla from ~0.10 to 1.0 mg/m$^3$ and primary production
from ~400 to 1000 mgC m$^{-2}$ d$^{-1}$ (Fig. 5;  Figs. 5 and 6 in Hood et al., 2017)  The highest concentrations and rates in the
SEC are observed in the Eastern Indian Ocean in July and August during austral winter, associated with the ITF nutrient
sources and upwelling off Java.  The lowest chlorophyll concentrations and rates are observed in January (austral summer).
Model results and satellite observations show that the SEC/SCTR region exhibits an annual cycle in surface chla
concentration and primary production, with the highest values in austral winter (June-August; > 0.20 mg/m$^3$ and >600
mgC m$^{-2}$ d$^{-1}$, respectively) due to the strong southeasterly winds that increase wind stirring and induce upwelling
(Resplandy et al., 2009; Dilmahamod, 2014; Fig. 5; Figs. 5 and 6 in Hood et al., 2017). Vertical sections of the SEC/SCTR
region also reveal a deep chla maximum (George et al., 2013). Along 65ºE this maximum shoals from > 100 m at 16ºS to
~50 m at 10ºS due to upwelling. The increases in surface Chl-a concentrations in austral winter are associated with
decreases in the subsurface chla maximum (Resplandy et al., 2009; Dilmahamod, 2014). Surface freshening associated
with the core of the SEC also influences the chla distribution in the SCTR region by modulating the static stability and
mixed layer depth (George et al. (2013).
The SEC provides relatively oligotrophic (low nutrient, low chlorophyll and low primary production) tropical source
waters that feed into the EACC, NEMC, SEMC and the Mozambique channel. Chlorophyll *a* concentrations and
production rates in Mozambique Channel surface waters are generally low (< 0.4 mg/m$^{-3}$ and < 700 gC m$^{-2}$ d$^{-1}$, Fig. 5),
and not significantly different in cyclonic and anticyclonic eddies (Lamont et al., 2014; Barlow et al., 2014; Figs. 5, 6 and
20 in Hood et al., 2017). Deep chlorophyll maxima are observed between 25 and 125 m depth depending on the proximity
to the shelf and the influence of mesoscale eddies (Barlow et al., 2014; Lamont et al., 2014). Eddies in the Mozambique
Channel also have a strong influence on the lateral transport of nutrients and chlorophyll from the coasts of Madagascar
and Africa. Indeed, enhanced phytoplankton production within both cyclonic and anticyclonic eddies in the Mozambique
Channel often occurs in response to lateral nutrient inputs into the euphotic zone by horizontal advection from the coasts
of Madagascar and Africa rather than through eddy induced upwelling and downwelling (José et al., 2014; Lamont et al.,
2014; Roberts et al., 2014). In contrast, in the Southeast Madagascar Current, topographically-induced coastal upwelling
brings cold, nutrient-rich water up to the surface, which supports high rates of primary production (Lutjeharms and Machu,
2000; Ho et al., 2004; Quartly and Srokosz, 2004). This upwelling and its impacts are observed in both the austral summer
and winter (Ho et al., 2004).
The Agulhas Current itself is warm and oligotrophic with sources derived from low nutrient and low chlorophyll surface
waters from the Mozambique Channel, Southeast Madagascar Current and the southwestern tropical Indian Ocean (Fig.
5; Lutjeharms, 2006). Chlorophyll *a* concentrations and production rates in Agulhas Current surface waters are particularly
low during austral summer (< 0.2 mg/m$^{-3}$ and < 500 mgC m$^{-2}$ d$^{-1}$) with higher concentrations and rates in the austral winter
(Machu and Garcon, 2001; Figs. 5, 6 and 20 in Hood et al., 2017). The Agulhas Current can drive upwelling and elevate
primary production in the coastal zone through meandering and topographic interactions, but it can also dramatically
suppress primary production when it impinges onto the shelf (Schumann et al., 2005).
In general, chlorophyll concentrations and primary production are elevated in the coastal zone of southeast Africa along
the inshore side of the Agulhas Current (Fig. 5; Machu and Garcon, 2001; Goschen et al., 2012; Figs. 5, 6 and 20 in Hood
et al., 2017). This enhancement is most pronounced in austral summer and further southward downstream, and it is
associated with upwelling-favourable (easterly) winds and the aforementioned topographically-induced upwelling.
The near-surface eastward flows are generally associated with very low (oligotrophic) nutrient and chlorophyll-$a$ (Chl-$a$)
concentrations (< 0.1 mg/m$^3$) and also very low primary production (< 500 mgC m$^{-2}$ d$^{-1}$; Fig. 5  Figs. 5 and 6 in Hood et
al., 2017). A well-defined deep Chl-$a$ maximum is observed  between 50 and 150 m during the austral fall along 55ºE
between 20 and 30ºS (Coles et al., unpublished data).  An exception to this, however, is the South-East Madagascar bloom
(SMB). The SMB occurs in near-surface waters off the southeastern coast of Madagascar in the late austral summer/fall
(Jan-April). It was first described as a dendroid bloom by Longhurst (2001), owing to its branching shape that projects
eastward (Fig. 12). The bloom can extend over a 2,500 km$^2$ area with Chl-$a$ concentrations reaching 2-3 mg/m$^3$ (Longhurst,
2001), making it a 'hot spot' for primary production in an otherwise oligotrophic region. Fig. 12 illustrates the bloom's
large spatial variability, with high Chl-$a$ filaments apparently co-occurring and being transported with mesoscale and
submesoscale eddies and jets.

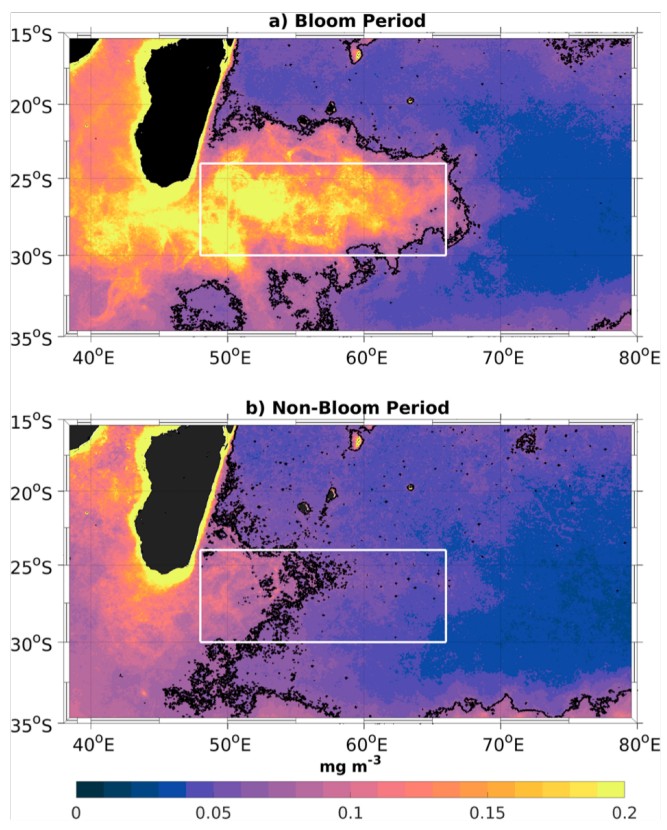


**Figure 12: (a) Spatial maps of mean Chl-a concentration (mg/m3) during months of maximum austral summer bloom. (b) Same as (a) but during January of minimum Chl-a concentration in austral summer. The black contour denotes the 0.07 mg/m3 threshold used to distinguish between bloom and non-bloom years. From Dilmahamod et al. (2019).**

Why the SMB flourishes in late austral summer is unclear. Longhurst (2001) attributed SMB development to mixed layer deepening and entrainment of nutrients by the vigorous mesoscale eddy field. These nutrients could stimulate phytoplankton growth in the photic zone, with the eddies shaping the eastward propagation of the enhanced surface Chl-*a* concentrations. However, Uz (2007), Srokosz and Quartly (2013), and Dilmahamod et al. (2019) subsequently showed that the bloom occurs within a warm (> 26.5°C), shallow mixed layer (~30 m) overlying a strong pycnocline. Furthermore, they suggested that diazotrophs known to inhabit the region (Poulton et al., 2009) might introduce new nitrogen (N) from $N_2$ fixation that could support the enhanced Chl-*a* concentration as observed elsewhere (Mulholland et al., 2014; Hood et al. 2004; Coles et al., 2004). Subsequent studies also highlight the role of mesoscale eddies (Fig. 12), that could advect,

disperse and co-mingle nutrients and/or phytoplankton biomass (Dilmahamod et al. 2019; Huhn et al., 2012; Raj et al., 2010; Srokosz and Quartly, 2013; Srokosz et al., 2004, 2015; Uz, 2007).

A different explanation of the SMB and its eastward projection was proposed by Srokosz et al. (2004). In their proposed mechanism, the bloom initiates off Madagascar due to coastal processes that bring limiting nutrients to the photic zone and phytoplankton are transported horizontally by mesoscale eddies, resulting in an eastward propagation of the bloom. Dilmahamod et al. (2020) extend this further using a model to suggest that, from a nutrient flux analysis, horizontal advection of low-salinity nutrient-rich Madagascan coastal waters can indeed trigger a phytoplankton bloom. Alternatively, the apparent eastward propagation of the SMB has recently been attributed to advection by the SICC (Fig. 10; Dilmahamod et al., 2019; Huhn et al., 2012; Wilson and Qiu, 2008). Indeed, Huhn et al. (2012) further suggested that the bloom is shaped by a meridional barrier of jet-like Lagrangian coherent structures associated with the SICC.

At the eastern boundary, the tropical source waters and downwelling tendency of the Leeuwin Current combine to create a warm, oligotrophic current with low productivity. Chl-$a$ concentrations are usually $< 30$ mgChla m$^{-2}$ and rates of primary production rates generally do not exceed 500 mgC m$^{-2}$ d$^{-1}$ (Koslow et al., 2008; Lourey et al., 2006; Lourey et al., 2013). Productivity in the Leeuwin Current is lowest during austral summer, when the water column is stratified. During summer, subsurface chlorophyll maxima are found between 50 and 120 m depth (Hanson et al., 2007) as observed in open ocean subtropical oligotrophic waters (e.g., Venrick, 1991). However, rates of primary production in near shore upwelling regions (e.g., off of the North West Cape during summer) can sometimes attain very high levels (3000–8000 mgC m$^{-2}$ day$^{-1}$) as observed in other eastern boundary upwelling zones (Furnas, 2007).

In all seasons, meanders in the Leeuwin Current give rise to warm core, anticyclonic eddies that carry moderately high chlorophyll coastal water offshore. The elevated chlorophyll concentrations in these eddies is due to the presence of coastal diatom communities. These diatoms are transported offshore into cooler oligotrophic waters that are dominated by much smaller open ocean phytoplankton species (Waite et al., 2007a; Paterson et al., 2008; Waite et al., 2016). These eddies, which can extend to more than 2000 m depth, are unusual because they are downwelling (anticyclonic) circulations that should inhibit the input of new nutrients from depth. Nonetheless, these eddies, and the elevated chlorophyll concentrations that are associated with them, persist for months (Feng et al., 2007; Moore et al., 2007; du Fois et al., 2014). It has been hypothesized that the diatom communities in these eddies are supported by internal nutrient recycling and/or lateral supply (Waite et al., 2007a; Paterson et al., 2013; Thompson et al., 2007, 2011).

Generation of these warm (and cold) core eddies by the Leeuwin Current is prolific between 20$^\circ$ and 35$^\circ$ S (Gaube et al., 2013). Most of these eddies move directly westward and some may be very long-lived (Feng et al., 2005; Feng et al., 2007; Moore et al., 2007; Gaube et al., 2013; du Fois et al., 2014). The persistence and potential biogeochemical/ecological impacts of these eddies in the open ocean have not been investigated fully.


### 4.3 Equatorial regime

### 4.3.1 Wyrtki Jets

Owing to the seasonally reversing monsoon winds, the equatorial Indian Ocean (EIO) exhibits unique characteristics and
is in contrast with the equatorial Atlantic and Pacific Oceans. Unlike the other basins, the annual winds along the EIO are
very weak and mostly meridionally oriented except during the two intermonsoon seasons between boreal winter (April-
May) and summer (Oct-Nov) when strong westerly wind bursts prevail along the EIO (see Schott and McCreary, 2001
and references therein). The semi-annual cycle in the zonal wind is well known observationally and was shown to be due
to the meridional advection of easterly momentum by the cross-equatorial monsoon winds (Ogata and Xie, 2011). The
westerly winds force strong eastward jets in the top 100 m along the equator that are known as spring and fall Wyrtki Jets,
respectively (Wyrtki, 1973). These surface jets are usually confined within the top 100 m of the water column (Han et al.,
1999; Iskander et al., 2011) and deepen (shoal) the thermocline and elevate (lower) the sea level in the east (west) (Rao et
al., 1989; Schott and McCreary, 2001; Nagura and McPhaden, 2010a). These jets play a major role in zonal redistribution
of mass, heat, salt and other water properties at the Equator and in off-equatorial basins (Reppin et al., 1999; Murtugudde
and Busalacchi, 1999; Han et al., 1999; McPhaden et al., 2015; Chatterjee et al., 2017). Long term ADCP observations
from the RAMA equatorial mooring suggest that the fall jet in the central EIO is usually stronger with a maximum transport
of ~19.7 Sv compared to the spring jet which shows maximum transport of ~14.9 Sv with comparable standard deviations
(McPhaden et al., 2015).
These eastward surface zonal currents tend to propagate westward during spring and eastward during fall (Nagura and
McPhaden, 2016). The westward phase propagation speed during spring is estimated to be on average between 0.7-1.5 m
s$^{-1}$ (Qiu et al., 2009; Nagura and McPhaden, 2010a) and driven primarily by the westward propagating surface zonal winds
associated with atmospheric deep convection that moves from the Maritime Continent to the northern Bay of Bengal
during spring (Nagura and McPhaden, 2010b; Nagura and McPhaden, 2016). Equatorial Rossby waves may also contribute
to this westward propagation (Nagura and McPhaden, 2010a). In contrast, during fall, as the deep convection moves
southeastward, the surface equatorial zonal winds, and thus surface currents, propagate eastward.
The spring and fall Wyrtki Jets also show considerable intraseasonal and interannual variability. While the intraseasonal
variability of the Wyrtki Jets has been shown to be influenced by their own instability (Sengupta et al., 2001, 2007; Han
et al., 2004) and local winds (Masumoto et al., 2005, Sengupta et al., 2007, Iskander et al., 2009; Prerna et al., 2019), the
interannual variability of the Wyrtki Jets is mainly caused by the anomalous wind forcing along the EIO associated with
ENSO (Murtugudde et al., 2000; Gnanaseelan et al., 2012; Joseph et al., 2012) and IOD (Nagura and McPhaden, 2010b;
Nyadjro and McPhaden (2014); Prerna et al., 2019): IOD weakens (strengthens) the equatorial zonal winds during its

positive (negative) phase. While IOD modulates the zonal winds along the entire equator, the influence of ENSO is primarily limited to the eastern part of the EIO (Gnanaseelan et al., 2012). Moreover, it has been shown that these climate modes affect the boreal fall jet more significantly than the boreal spring jet. Recent modelling studies suggest that MJO convection can lead to a stronger spring Wyrtki jet particularly in the eastern EIO. The interannual variability of MJO can, therefore, contribute to the observed interannual variability of this equatorial jet as well (Deshpande et al., 2017; Prerna et al., 2019).

### 4.3.2 5-30 Day Ocean Waves and Instabilities

Meridional velocity along the equator shows prominent high frequency variability at all depths, in the periodic band of 10-20 days with a peak at ~15 days (referred to as biweekly variability) and in the 20-30 days band with a peak at ~25 days (Masumoto et al., 2005; David et al., 2011; Chatterjee et al., 2013; Smyth et al., 2014). This variability is attributed to Yanai waves, first discovered in the atmosphere (also referred to as mixed Rossby-Gravity waves; Yanai and Maruyama 1966; Arzeno et al., 2020; Pujiana and McPhaden, 2021). Unlike Kelvin and Rossby waves, Yanai wave phases can propagate westward or eastward depending upon their frequency, but their group velocity is always eastward (Miyama et al., 2006). These waves lead to convergent meridional heat flux into the equatorial regime (Shinoda, 2009; Smyth et al., 2014). While these waves were first observed in the ocean in the late 1990s, the establishment of the equatorial RAMA moorings (McPhaden et al., 2009) over the last two decades has provided more insight into these processes. Bi-weekly (10-20 day) is shown to be forced by the direct meridional wind stress (Sengupta et al., 2004) and to some extent by the meridional gradient of the zonal wind stress (Miyama et al., 2006). The 20–30-day band can be excited by off-equatorial barotropic/baroclinic instabilities in addition to direct wind forcing. A detailed review of the biweekly variability is provided in Schott et al. (2009) and hence, we focus on the 20-30 day variability in this review.

While the 20-30-day oscillation in meridional velocity is reported near the surface in the central EIO (David et al., 2011), in the eastern EIO these variabilities are seen only in subsurface layers (100-200 m depth) of the water column (Masumoto et al., 2005). This indicates a possible downward energy propagation of a vertical beam that carries energy to deeper depths. In the central EIO, these 20-30-day Yanai waves are excited by horizontal shear between the westward-flowing South Equatorial Current and the eastward-flowing Southwest Monsoon Current during IOD events (Fig. 10a; David et al., 2011). In the western EIO, these waves are primarily driven by cross equatorial meridional winds (Chatterjee et al., 2013). During early boreal summer (June/July), when the Somali current begins to cross the Equator along the western boundary of the basin, it bends offshore to conserve potential vorticity (Schott and McCreary, 2001) and forms a gyral circulation known as the Southern Gyre (Fig. 10a). Subsequently, these swift currents turn barotropically unstable and generate eddy flow that is advected southward to the Equator near the western boundary i.e. at ~50-55°E. They generate a westward propagating cross-equatorial flow with a wavelength set by the eddy field which is similar to the wavelength of 20-30 day Yanai waves and thus excite these frequencies efficiently (Chatterjee et al., 2013).

The ocean response to convectively coupled Kelvin waves (CCKW) in the atmosphere was investigated using ocean glider
measurements from the CINDY/DYNAMO field experiment (Webber et al., 2014; Matthews et al., 2014). CCKW are
atmospheric weather systems that propagate eastward along the Equator and are an important constituent of the MJO
convection (Baronowski et al., 2016). CCKW enhance surface wind speed and latent heat flux during their passage
suppressing the diurnal cycle of SST and leading to sustained decrease in bulk SST of around 0.1°C, one third of the SST
anomaly due to a single, average MJO event, suggesting the oceanographic impact could have a strong feedback on the
MJO cycle (Baronowski et al., 2016). Using RAMA moored measurements of upper ocean and surface atmosphere
variability, Pujiana and McPhaden (2018) demonstrated that CCKW force oceanic Kelvin waves, affect surface heat fluxes
and generate upper ocean turbulence.
### 4.3.3 Equatorial Upwelling and Downwelling
In the Pacific and Atlantic Oceans, permanent easterlies drive permanent equatorial upwelling due to Ekman
divergence, but in the Indian Ocean where the mean winds are weak and westerly, permanent upwelling does not
exist (Schott and McCreary, 2001). Mean westerly winds along the Equator are downwelling favorable, driving
surface convergence and thermocline divergence, which has been observed and described with Argo and RAMA data
(Wang and McPhaden, 2017). Instead of upwelling along the equator, coastal upwelling along the coasts of Sumatra
and Java is prominent. During June-October, south-easterly trade winds blow close to the Equator and drive the
offshore Ekman transport away from the Sumatra-Java coast (Quadfasel and Cresswell, 1992; Sprintall et al., 1999;
Susanto et al., 2001). The associated wind-driven upwelling intensifies as the monsoon progresses, reaching its peak
by August and finally weakening by October as the monsoon winds wane. Recent studies suggest that when the
easterly winds prevail during summer, upwelling favourable Kelvin waves also contribute to intensifying the
equatorial upwelling (Iskander et al., 2009; Chen et al., 2016). During boreal winter-early spring (December-March),
an intermittent/weaker subsurface thermocline shoaling is evident (Chen et al., 2016). Subsequently, the prevalence
of westerly winds, which drive downwelling Kelvin waves, depress the thermocline in the east (Susanto et al., 2001;
Prerna et al., 2019).  Apart from this seasonal cycle, interannual climatic variability associated with ENSO and IOD
events (Saji et al., 1999; Vinayachandran et al., 1999, Nyadjiro and McPhaden, 2014) also influences the upwelling
intensity in this region (Section 6.2).

### 4.3.4 Equatorial Undercurrents
In the Pacific and Atlantic, easterly winds produce an eastward mean undercurrent in the thermocline but in the
Indian Ocean westerly winds do not produce a mean westward undercurrent. The reason is that nonlinear momentum
advection drives mean eastward currents in the thermocline that flow up the zonal pressure gradient (Nagura and
McPhaden, 2014). The  Indian Ocean Equatorial Undercurrent (EUC) is, therefore, a much weaker and seasonally
varying transient feature driven by seasonally reversing monsoon winds (Reppin et al., 1999; Schott and McCreary,
2001). The equatorial RAMA moorings have recorded an eastward EUC with a core within the thermocline during
boreal winter and spring (Chen et al., 2015, 2019) and occasionally in summer and fall at a depth of 90-170 m
(Iskandar and McPhaden, 2011). During winter, the eastward EUC is forced by the upwelling Kelvin and Rossby
waves that are in turn forced by easterly winds along the equator in that season. During summer, the westward EUC
is primarily forced by the eastward pressure gradient generated by the downwelling reflected Rossby waves off the
eastern boundary of the basin. On intraseasonal timescales of 30-70-days, the EUC variability is dominated by that of
Kelvin and Rossby waves of lower order baroclinic modes (Iskander and McPhaden, 2011). The undercurrents also
undergo significant interannual variations related to the IOD. These variations are important in the mass and heat
balance on IOD time scales, with significant impacts on upwelling and SST (Zhang et al., 2014; Nyadjro and
McPhaden, 2014)

**4.3.5 Cross-Equatorial Circulation**
The cross-equatorial circulation in the upper ocean is achieved by the Cross-Equatorial Cell (CEC), driven by
southern hemisphere southeasterly winds and the seasonally-reversing monsoon winds in the northern hemisphere
(Miyama et al. 2003; Schott et al. 2002, 2004, 2009). Thermocline waters subducted in the subtropical southeast
Indian Ocean move equatorward and enter the northern hemisphere via the western boundary to upwell off Somalia
and Oman. The return across the Equator, the surface branch of the CEC, is via the near-surface meridional flow in
the interior Indian Ocean that is southward in the mean at nearly all longitudes (Miyama et al., 2003; Lee, 2004). This
cell is unique to the Indian Ocean and is consistent with Sverdrup dynamics, being driven by the predominantly
negative wind stress curl (Godfrey et al., 2001; Miyama et al., 2003; Wang and McPhaden, 2017). It carries most of
the cross-equatorial transport of mass and heat (Schott and McCreary, 2001) and helps to moderate the seasonal
climate of the region. The seasonal cross-equatorial mass flux is oppositely directed along the western boundary and
in the interior (Beal et al., 2013). Flow in the interior is directed from the summer to the winter hemisphere (Horii et
al., 2013; Wang and McPhaden, 2017) consistent with monsoon wind forced Ekman and Sverdrup dynamics as
proposed in the model study of Miyama et al. (2003).

In OGCMs, the southward flow of the CEC was found to occur just below the surface, beneath a northward surface
current (Wacogne and Pacanowski, 1996; Miyama et al. 2003). This "equatorial roll", also unique to the Indian
Ocean, is only of order 100 m depth and so has little impact on the cross-equatorial heat transport of the CEC. Horii
et al. (2013) and Wang and McPhaden (2017) presented the first observational evidence for the equatorial roll.

The spatial structure and time evolution of the cross-equatorial circulation is difficult to depict due to its dependence
on the fluctuating monsoon winds. Consequently, the flow patterns obtained from an Eulerian average as in Fig. 10
cannot capture the monsoon-dependent streamlines that a flow will follow at a given moment. Lagrangian methods
based on ocean drifter velocities (Laurindo et al. 2017) and real and simulated surface drifter trajectories identify
pathlines that connect the monsoonal Indian Ocean, revealing three cross-equatorial gyre pathways that connect the
Somali Current with the interior flow north and south of the Equator (Fig. 7 in l'Hegaret et al., 2018).

### 4.3.6 Biogeochemical Variability

Much of the current understanding of biogeochemical variability in the equatorial zone of the Indian Ocean is based on
satellite ocean color observations and models, augmented by some additional, relatively sparse, in situ measurements.
Seasonal climatologies of near-surface chlorophyll concentrations and primary production show a significant seasonality
in equatorial waters that is clearly associated with monsoon forcing (Fig. 5, Wiggert et al., 2006; Strutton et al., 2015;
Figs. 5 and 6 in Hood et al., 2017). In general, Chl-*a* concentrations and primary production increase northward from the
equator with the lowest concentrations ($< 0.1$ mg m$^{-3}$) and rates ($<800$ mg C m$^{-2}$ d$^{-1}$) occurring during the boreal spring
intermonsoon period. During the southwest monsoon, Chl-*a* concentrations and rates of primary production increase in
western equatorial waters in response to monsoon-forced mixing and upwelling. However, concentrations and rates in the
central and eastern equatorial waters stay relatively low ($< 0.5$ mg m$^{-3}$, $<800$ mgC m$^{-2}$ d$^{-1}$, respectively). Island wake
effects can be seen advecting high chlorophyll water ($> 0.5$ mg m$^{-3}$) along the equator from the Chagos-Laccadive ridge
at 73°E eastward during the autumn intermonsoon period and westward during spring (see Fig. 1 in Strutton et al., 2015).
Well-developed deep Chl-*a* maxima have been observed in the equatorial Indian Ocean along 65°E centered at about 50
m depth in November-December (George et al., 2013) and along 80°E centered at about 75 m in August-September
(Sorokin et al., 1985). It is unknown whether or not this subsurface Chl-*a* maximum exists along the equator throughout
the year, but it is probably present whenever the water column is stratified. Models predict the presence of a subsurface
(60 m) Chl-*a* maximum in eastern Indian Ocean equatorial waters along 87°E (Wiggert et al., 2006) that is present
throughout the year except during the southwest monsoon when high chlorophyll surface water is advected into the region.
Physical processes at time scales from intraseasonal to interannual (i.e., Wyrtki Jets, MJO and IOD) have been shown to
influence biogeochemistry. For example, IOD events can significantly increase chlorophyll concentrations and primary
production in eastern Indian Ocean equatorial waters (Wiggert *et al*., 2009). In addition, relaxation of an IOD can deplete
upper ocean nutrients, decreasing biological productivity (Kumar *et al*., 2012). Biogeochemical responses to the IOD also
have significant higher trophic level impacts (Marsac and Le Blanc, 1999).
Satellite observations and biophysical model simulations show how chlorophyll concentrations and primary production
near the Seychelles-Chagos thermocline ridge, can be increased by MJO-induced wind mixing and nutrient entrainment
(Resplandy et al., 2009). They also concluded that IOD-driven interannual variability of thermocline depth influences the
biogeochemical response to MJO: the deepened nutricline following IOD events inhibits nutrient input into the mixed
layer and thus decreases the biogeochemical response to MJO.
In model simulations, Wyrtki jets depress the thermocline and nitracline along the equator on the eastern side of the basin
and, as a result, lower equatorial primary production when they arrive in the spring and autumn (Wiggert et al. 2006). This
pattern was observed in a 25 day time series study on the equator at 80.5°E in late 2006 that showed a deepening of the
surface layer, nutracline and subsurface Chl-*a* maximum during the autumn Wyrtki jet period (Kumar et al., 2012).
Finally, Strutton et al. (2015) examined time-series measurements of near-surface chlorophyll concentration from a
mooring deployed in 2010 at 80.5 E in the equatorial Indian Ocean. These data revealed at least six spikes in chlorophyll
from October through December, separated by approximately 2-week intervals and coinciding with the development of
the fall Wyrtki jets. The chlorophyll pulses were associated with increases in eastward surface winds and eastward currents
in the mixed layer and inconsistent with upwelling dynamics because eastward winds that cause intensification of the
Wyrtki jet should drive downwelling. Strutton et al. (2015) concluded that the chlorophyll spikes could be explained by
two alternative mechanisms: (1) turbulent entrainment of nutrients and/or chlorophyll from across the base of the mixed
layer by wind stirring or Wyrtki jet-induced shear instability or (2) enhanced southward advection of high chlorophyll
concentrations into the equatorial zone associated with wind-forced biweekly Yanai waves.
**4.4 Northern Indian Ocean**
The two main basins of the northern Indian Ocean, the Bay of Bengal (BoB) and the Arabian Sea (AS), are characterized
at the surface by remarkably contrasting sea surface salinity with differences of the order of 3 psu (e.g. Chatterjee et al.
2012, Gordon et al. 2016, Hormann et al. 2019) decreasing from west to east (Fig. 4). The fresh surface layer of the BoB
is maintained by large freshwater input deriving from direct rainfall over the ocean and river runoff, especially during the
South Asian monsoon. The salt balance of the BoB is maintained by the subsurface supply of salt water via the Southwest
Monsoon Current (Fig. 10, Vinayachandran et al., 2013). The saltier SSS of the AS is the consequence of an evaporative
regime (e.g., Rao & Sivakumar, 2003; Sengupta et al., 2006). A reversing monsoonal near-surface circulation (Fig. 10 a,b)
plays a central role in the exchanges of freshwater and heat between the BoB and the AS (McCreary et al. 1993, Hormann
et al. 2019).
Recent multi-year deployments of satellite tracked surface drifters drogued at 15 m depth (Wijesekera et. al, 2016,
Centurioni et al. 2018) have helped to better constrain the amplitude and structure of the circulation and the exchange
processes between the two basins, and to refine the findings reported by other authors (e.g. Schott and McCreary, 2001).
Additionally, implementation of a moored buoy network along the slope and shelf of the Indian coast has helped
significantly in enhancing our understanding of the east India Coastal Current (EICC) and west India Coastal Current
(WICC) (Fig. 10, Mukherjee et al., 2014; Amol et al., 2014; Mukhopadhyay et al., 2020; Chaudhuri et al. 2020).

### 4.4.1 Bay of Bengal

**4.4.1 Bay of Bengal**
In a climatological sense, the main features of the near-surface circulation of the western BoB (Figs. 10a and 10b) are the
reversing EICC, the Southwest/Northeast Monsoon Current (SMC/NMC) and the seasonally variable Sri Lanka Dome.
The eastern side of the BoB, extending into the Andaman Sea, is characterised by a sluggish circulation.
**4.4.1.1 Southwest/Northeast Monsoon Currents**
During the boreal summer SW monsoon, the Southwest Monsoon Current (SMC, Fig. 10a) flows eastward around the
Indian subcontinent supplying salty water from the Arabian Sea to the fresher Bay of Bengal (e.g., Jensen, 2001; Jensen
et al., 2016; Vinayachandran et al., 2013; Wijesekera et al., 2015, 2016). During the winter monsoon, the Northeast
Monsoon Current (NMC, Fig. 10b) reverses the flow carrying fresher water into the Arabian Sea. Figure 13 provides a
snapshot from an operational forecast system, the Coupled Ocean-Atmosphere Mesoscale Prediction System (COAMPS),
of the NMC flow and route for freshwater to enter the Arabian Sea (Wijeskera et al., 2015).

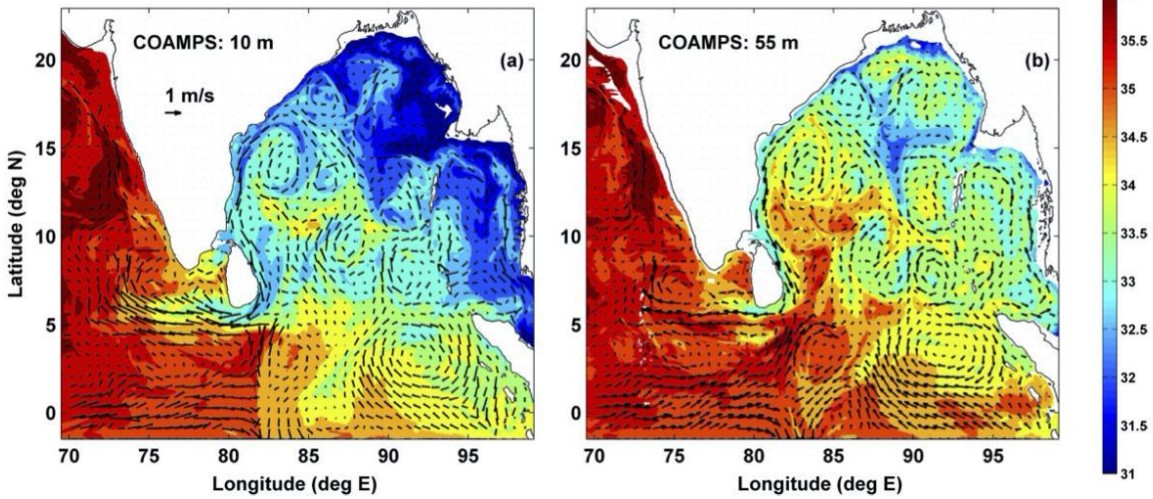


**Figure 13: COAMPS velocity vectors (arrows) and salinity (psu, color shading) at (a) 10 m and (b) 55 m on 18**
**December 2013. Modified from Wijeskera et al. (2015).**
A more recent study has found that the origins of the Arabian Sea high salinity water are specifically from the western
Arabian Sea and western Equatorial Indian Ocean, and they reach the Bay of Bengal via a combination of the Indian Ocean
EUC and the SMC (Sanchez-Franks et al., 2019; Section 8.2). Changes in the supply of salty water to the Bay of Bengal
varies interannually due to the strength in the equatorial currents, forced by the local wind field and ENSO (Sanchez-
Franks et al., 2019), and is expected to influence the salinity budget of the Bay of Bengal (Vinayachandran et al., 2013)
and thus modulate SST variability (Fig. 10a, Jensen, 2001;Jensen et al., 2016; Li et al., 2017; Vinayachandran et al., 2013,
2018; Webber et al., 2018).

## 4.4.1.2 East Indian Coastal Currents (EICC)

The EICC forms the western boundary current of the Bay of Bengal and plays an important role in the basin-scale heat
and salt budget of the Indian Ocean, and hence in determining the local climate (Shenoi et al, 2002), biological processes
(Madhupratap et al, 2003; Vinayachandran et al, 2005; Naqvi et al, 2006; Dileepkumar, 2006; McCreary et al, 2009) and
marine fisheries (Vivekananda and Krishnakumar, 2010) of this region. It reverses its direction seasonally north of 10°N
in response to a combination of local alongshore winds, remote alongshore winds in the eastern BoB, remote forcing from
the equatorial Indian Ocean and the interior Ekman pumping of the basin (Shankar et al., 1996; McCreary et al., 1996;
Vinayachandran et al., 1996; Mukherjee et al., 2018). The EICC is generally equatorward south of 10°N throughout the
year. While local winds dominate the EICC forcing during summer and winter, remote forcing dominates during the inter-
monsoon periods (Shankar et al., 1996; McCreary et al., 1996; Suresh et al., 2013).
Climatological ship-drift and hydrographic data suggest the EICC flows poleward during February-September (Shetye et
al., 1993) and turns equatorward during November-January (Shetye et al., 1996; Fig. 10). While the annual cycle is driven
by local alongshore winds and interior Ekman pumping, the semiannual cycle is the result of asymmetry in the monsoon
and equatorial forcing (Mukherjee et al., 2018). During boreal spring (March-May), the EICC is strongest with a magnitude
exceeding 1m/s with unidirectional currents to about 150 m, forming the western boundary current of a cyclonic basin-
wide gyre of the BoB. The local alongshore winds are weakest and the stronger EICC is primarily forced by the interior
anticyclonic Ekman pumping over the basin (McCreary et al., 1996; Shankar et al., 1996; Vinayachandran et al., 1996;
Mukherjee et al., 2018). During boreal summer, the EICC is weaker and is restricted to within the top 70 m of the water
column. The poleward flow is generally limited to the central part of the coast between 10-18°N and often switches to
short pulses of poleward currents along the coast (Mukherjee et al., 2018; Francis et al., 2020). The poleward flow is
driven by local winds, but the response of the interior cyclonic Ekman pumping and equatorial winds driving an opposite
flow along the coast causes a weaker poleward EICC in summer than in spring (McCreary et al., 1996; Vinayachandran
et al., 1996; Shankar et al., 2002). The basin-scale gyre also disappears in summer and the EICC then consists of several
eddies along the coast. The EICC turns equatorward during November-January (Shetye et al., 1996).
Near-surface alongshore currents also display significant 120 day and intraseasonal variability. The magnitude of the 120
day variability is generally weaker than the semiannual period, particularly in the southern part of the coast. As for the
annual period, upward phase propagation along the coast is also evident for the semiannual and 120 day period, except at
Cuddalore where downward phase propagation is common during summer and winter months (Mukherjee et al., 2014;
Mukhopadhyay et al., 2020). Further, unlike annual and semiannual periods, the 120 day and intraseasonal variability
decorrelate along the coast indicating that these high frequencies are dominated by local responses rather than remote
forcing (Mukherjee et al., 2018; Mukhopadhyay et al., 2020).
**4.4.1.3 Undercurrents**
ADCP observations suggest that during summer and winter, when the near-surface current is shallow, the EICC often
exhibits undercurrents along the continental slope. As the EICC is deeper in the north, the undercurrent is observed at a
depth of 100-150 m and can extend up to 700 m. However, in the south undercurrents are seen at relatively shallow depths
of about 70-75 m (Francis et al., 2020). While these undercurrents are observed throughout the coast, they are much more
prominent and more frequent at Cuddalore, the southernmost station of the coast (Fig. 14, Mukherjee et al., 2014;
Mukhopadhyay et al., 2020).

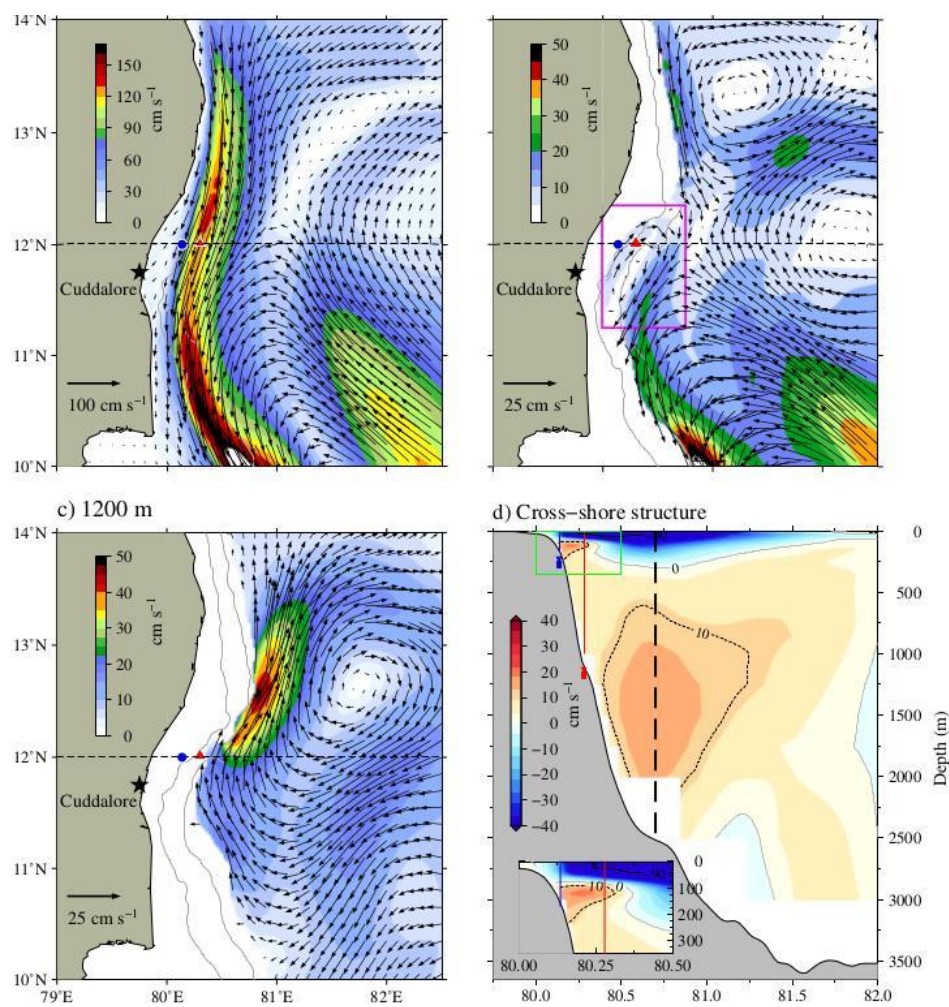

**Figure 14: Circulation pattern in the southwestern Bay of Bengal at (a) surface (b) 200 m and (c) 1200 m on 15 November 2014. Vectors show the current direction, and overlaid is the current magnitude (cm s$^{-1}$). Note that the scales of current vectors and color bars are different at each subplot. Blue circle (red triangle) represents the location of ADCP deployed on the shelf (slope) off Cuddalore. Dashed black line represents the 12°N latitude. Continuous gray lines represent the 100 m and 1000 m bathymetric contours. Rectangular box (magenta) indicates the subsurface eddy near the shelf break. (d) Cross-shore structure of alongshore currents across 12°N. Dashed black vertical line shows the core of the undercurrent, and red (blue) vertical lines show the location of ADCP mooring on the slope (shelf). Inset plot is the zoomed view of shelf break region indicated by green box (Reproduced from Francis et al., 2020).**

The prominent upward phase propagation of the annual signal in the subsurface layers, particularly in the southern stations, suggests downward propagation of energy and is thereby attributed as one of the main causes of the undercurrents

(Mukherjee et al., 2014). A recent modelling study suggests that the wintertime undercurrent off Cuddalore consists of
two separate subsurface anticyclonic eddy circulations: a shallow small scale circulation at a depth range of 100-200 m
and a broader and deep flow below 500 m depth off the continental slope (Francis et al., 2020). The shallow subsurface
anticyclonic eddy was found to spin off from the zonal shear of the mean near-surface EICC along the shelf break (Fig.
14). These eddies exhibit high frequency fluctuations and have 20-30 km length scales. Since the zonal share of the EICC
is primarily linked to the strength of the EICC itself, the variability and strength of this undercurrent is also linked with
the EICC.
**4.4.1.4 Sri Lanka Dome**
The Sri Lanka Dome (Vinayachandran et al. 1999; Schott and McCreary, 2001; Wijesekera et. al, 2016, Cullen and
Shroyer, 2019) is mainly visible as a closed anticyclonic (clockwise) eddy in the near-surface geostrophic current velocity
field starting in May and lasting through October (Fig. 15c). It is a recurring upwelling dome that forms east of Sri Lanka
between 5-10°N, 83–87°E. The SLD is embedded within the Southwest Monsoon Current (SMC) system (Gadgil, 2003)
and enhances the SMC exchange from the Arabian Sea to the Bay of Bengal (Anutaliya et al., 2017). Upwelling associated
with the SLD influences the vertical exchange of water properties, enhances biological productivity, and cools sea surface
temperature (SST) which affects local atmospheric convection (Vinayachandran et al., 2004; de Vos et al., 2014).

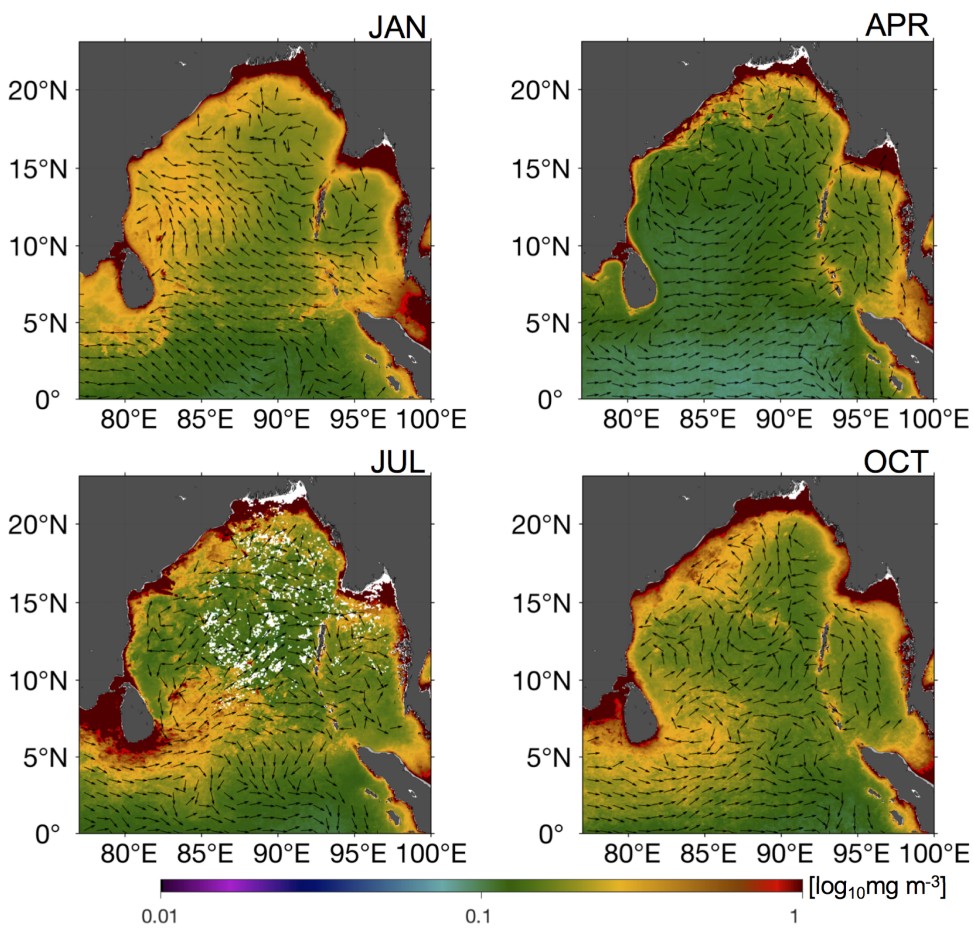


**Figure 15: Climatology (2002-2018) of chlorophyll-a concentrations (colormap) and current velocities (arrows) in the Bay of Bengal for (a) January (b) April (c) July (d) October. Chlorophyll climatology was obtained from the MODIS-Aqua product and current velocities were obtained from the third-degree Ocean Surface Current Analysis Real-time (OSCAR) product.**



### 4.4.2 Arabian Sea

Like the Bay of Bengal (BoB), the Arabian Sea (AS) near surface circulation is also driven primarily by the seasonally reversing monsoon winds. The AS is connected to the BoB through the passage between the southern tip of India and the equatorial wave guide, and to the southern hemisphere by the cross equatorial flow via the Somali current system. The Somali current (Fig. 10) forms one of the western boundary currents of the AS. Another major boundary current system

is along the west coast of India, the WICC (Fig. 10), which transports heat and salt from the northern Arabian Sea to the BoB, and vice-versa. Recent observations (Chatterjee et al., 2012) and modelling studies (Shankar et al., 2016; Vijith et al., 2016) indicate that the northern extent of the WICC reaches up to 20ºN during the winter monsoon, carrying fresher BoB water to the northern latitudes and modulating the wintertime convection there. In the last couple of decades, the strengthening of WICC and NMC has decreased the SSS in the eastern Arabian Sea (Varna et al., 2021).

### 4.4.2.1 Somali current System

The Somali Current is a seasonally reversing western boundary current and is often composed of discontinuous non-linear eddy driven flows. During summer it flows poleward and the upwelling here is nearly as large as for the eastern boundary upwelling regimes of the Pacific and Atlantic Oceans (See Schott and McCreary, 2001 for a detailed review). Unfortunately, owing to piracy, direct in-situ observations are very rare in this region and mostly date back to the early 1960s and 1970s. Hence, the scientific community has mostly relied on numerical model simulations to enhance understanding of this region over the last few decades.

Recent modelling studies suggest that during the summer monsoon, unlike other western boundary currents, the Somali Current system can be divided into three dynamically distinctive regions (Wang et al., 2018; Chatterjee et al., 2019): northern (north of 8ºN), central (3-8ºN) and the southern (south of 3ºN) part. The northern and southern parts are driven by the large anticyclonic gyres called the Great Whirl (GW) and the southern Gyre (SG), respectively (Fig. 10a). Local southwesterly alongshore winds known as the Findlater Jet (Findlater, 1969) drives Ekman transport all along the Somali coast (Schott and McCreary, 2001) with varied magnitude which is strongest in the southern part, and significantly weakens northward (Chatterjee et al., 2019). The wind stress forcing leads to Ekman Pumping in the central Arabian Sea, setting up a bowl-shaped mixed layer and warming at the 100 m level. Ekman downwelling velocities are strongest in the northern part and likely contribute to the formation of the Great Whirl front which upwells cold subsurface water in this part of the coast. The central part, in contrast, is mainly driven by the local winds and remotely forced Rossby waves. In fact, the annual Rossby waves radiated out of the southwestern coast of Sri Lanka seem to play a major role in the reversal of currents to poleward flow in the northern part of the Somali coast as early as mid April.  This reversal likely initiates the generation of the Great Whirl (Beal and Donohue, 2013; Vic et al., 2014), a month before the strong northeastward Findlater Jet commences along the Somali coast. As the monsoon progresses, these downwelling favourable Rossby waves oppose the coastal Ekman upwelling and thereby start to weaken the upwelling all along the coast. Moreover, as the alongshore winds peak, this favours enhanced mixing at the bottom of the mixed layer, which deepens the thermocline further. This process is more conspicuous in the central part of the coast, where the depth of the 22ºC isotherm deepens by about 30-40 m from June to August (Chatterjee et al., 2019). By this time, the upwelling becomes limited to the northern part of the coast along the Great Whirl front of the Somali region.

Climatological characteristics of the Somali Undercurrent (SUC) have been revealed by a new multi-decadal time series of temperature, salinity and geostrophic velocity constructed from repeat XBT transects and Argo observations (Zang et al. 2021). They find that the SUC flows southeastward during the monsoon transition periods in boreal spring (April-June) and fall (September-November), against the northeastward flow of the Somali Current. The depth of the SUC core is shallower during the spring transition (~500 dbar in April) and has a maximum depth of ~1200 dbar in September. Core speeds are 2.5-4 cm s$^{-1}$ in spring; in fall the core speed strengthens from 0.2 to 10.6 cm s$^{-1}$ from September to November and then disappears in December. Volume transport of the Somali Current and SUC (0-2000 dbar) has a maximum of 29.6 Sv northeastward in the summer monsoon and minimum of 13 Sv southwestward during the fall transition when the SUC is strong.

4.4.2.2 West India Coastal Current (WICC)

The WICC reverses its direction annually: flowing equatorward (upwelling favourable) during the summer monsoon (May to September; Fig. 10a) and poleward (downwelling favourable) during the winter monsoon (November to February; Fig. 10b). The equatorward flow during the summer characterises the WICC as a classical eastern boundary current (Shetye and Shenoi, 1988). Interestingly, as the monthly mean alongshore winds off the west coast of India are always equatorward throughout the year, the surface currents flow against the winds during the winter, driven by coastally-trapped Kelvin waves forced remotely in the BoB and along the east coast of India (McCreary et al., 1993; Shankar and Shetye, 1997; Shankar et al., 2002; Suresh et al., 2016). Recent observations based on satellite data and alongshore ADCP moorings reveal strong interannual variability of this seasonal cycle. Vialard et al. (2009b), based on a short ADCP record during 2006-2008, reported an absence of seasonal cycle off Goa and they attributed this absence to the radiation of Rossby waves south of the critical latitude. As the longer record of ADCP data became available, a clear seasonal cycle in the WICC became evident with weaker amplitudes in the south, stronger poleward (Amol et al., 2014).

The WICC also shows significant intraseasonal variability at times, particularly during boreal winter, exhibiting much stronger energy in the intraseasonal band than in the seasonal band (Vialard et al., 2009b; Amol et al., 2014). Unlike the seasonal cycle, intraseasonal variability is stronger in the south and weakens poleward. Vialard et al. (2009b) attributed this intraseasonal variability to the atmospheric MJO forcing. Recently, a modelling study suggested that interception of the intraseasonal equatorial Rossby waves by the southern tip of India and Sri Lanka excites coastal Kelvin waves which contribute significantly (~60–70%) to the intraseasonal variability along the west coast (Suresh et al., 2013). A satellite sea level study by Dhage and Sturb (2016) confirmed the model-based findings of Suresh et al. (2013) and revealed that large-scale winds from the south of India and Sri Lanka also contribute to the coastal signals along the west coast of India.

Another striking feature observed in these ADCP data is the clear signature of upward phase propagation in all timescales during both monsoon seasons. This upward phase propagation is more conspicuous for the seasonal period than for the

intraseasonal. As a result, the phase of the surface currents often tends to be opposite that in the subsurface layers (Amol
et al., 2014). Moreover, it is found that the strength of this undercurrent intensifies northward along the west coast with
strongest undercurrent off Mumbai and the weakest off Kanyakumari (southernmost point of Indian mainland), indicating
a possible downward propagation of energy along the ray path as suggested earlier by Nethery and Shankar (2007). Since
the ray angle ($\theta$) depends on the frequency ($\sigma$) and stratification ($N_b$) according to $\theta = \sigma / N_b$ (McCreary, 1984; Nethery and
Shankar, 2007) the angle the beam makes from the horizontal is deeper for the intraseasonal band than for the seasonal.
As a result, intraseasonal beams propagate energy deeper into the water column. Therefore, while the WICC shows some
coherence along the coast in the seasonal time scale, it completely decorrelates horizontally for the intraseasonal period.

### 4.4.3 Biogeochemical Variability

In the Bay of Bengal, the large freshwater input gives rise to enhanced stratification that inhibits upwelling and wind-
mixing and therefore nutrient supply to surface waters (Kumar et al., 2002; Vinayachandran et al., 2002; Madhupratap et
al., 2003; Vinayachandran, 2009). Nonetheless, increased productivity is observed along the coast primarily in association
with riverine nutrient inputs (Vinayachandran, 2009). These nutrients stimulate diatom blooms (Sasamal et al., 2005)
leading to significant increases in Chl-*a* concentration ($\sim$ 30–100 mgChla m$^{-2}$) and production ($\sim$ 0.55–1 gC m$^{-2}$ d$^{-1}$) near
the coast (Gomes et al., 2000; Fig. 15). This high Chl-*a* river water flows either along the coast or offshore, up to several
hundred kilometers, depending on the coastal current pattern (Vinayachandran, 2009). Along the Indian coast, the flow
of Chl-*a*–rich water is determined by the EICC, which flows northward during the spring intermonsoon period and
Southwest Monsoon and southward during the autumn intermonsoon and Northeast Monsoon (Fig. 15). When the EICC
meanders seaward from the Indian coast, it leads to offshore increases in high chlorophyll water. During the spring
intermonsoon and Southwest Monsoon the northward-flowing EICC is upwelling favorable, which may contribute to
increases in Chl-*a* concentration and primary production along the coast (Hood et al., 2017)

Elevated productivity is observed further offshore in the southwestern Bay of Bengal during the Northeast Monsoon
(Vinayachandran and Mathew, 2003; Vinayachandran, 2009). Modeling studies suggest that this is caused by wind-driven
entrainment, not only of subsurface nutrients, but also of phytoplankton from the subsurface chla maximum that is present
during the autumn intermonsoon period (Vinayachandran et al., 2005). In contrast, productivity near the coast is
suppressed during the Northeast Monsoon when the EICC flows southward (Fig. 14). Presumably, this is due to a
combination of the downwelling-favorable currents and winds. However, primary production over the shelf in the northern
part of the Bay increases during the Northeast Monsoon (Gomes et al., 2000; Fig. 15 ), possibly due to river nutrient inputs
(Vinayachandran, 2009) and / or wind-stress and buoyancy-driven nutrient entrainment as is observed in the northern
Arabian Sea during the Northeast Monsoon (Wiggert et al., 2000; 2005; Hood et al., 2017).

Subsurface Chl-*a* maxima are observed in the Bay of Bengal during all seasons whenever and wherever wind forcing
and/or currents are insufficiently strong to upwell or entrain them into the surface layer (Sarma and Aswanikumar, 1991;
Murty et al., 2000; Sarjini and Sarma, 2001; Kumar et al., 2007).  During the intermonsoon periods the Bay of Bengal
transitions to more oligotrophic conditions with relatively low surface chlorophyll concentrations ($< 0.6$ mg/m$^3$; Fig. 15)
and production rates ($< 700$ mgC m$^{-2}$ d$^{-1}$; see Fig. 6 in Hood et al., 2017). *Trichodesmium erythraeum* blooms have been
observed during the intermonsoon periods along with high abundances of *Synechococcus* and heterotrophic dinoflagellates
(Sarjini and Sarma, 2001; Jyothibabu et al., 2008). In offshore waters subsurface chlorophyll maxima are generally located
between 40 and 70m in autumn and 60 and 90m in spring (Kumar et al., 2007).  These deep Chl-*a* maxima tend to shoal
near the coast (Sarma and Aswanikumar, 1991; Murty et al., 2000) and their depth and chlorophyll concentrations are
strongly influenced by eddies (Kumar et al., 2007).

Strong upwelling also occurs along the southern coast of Sri Lanka during the Southwest Monsoon  (Vinayachandran,
2004; 2009; de Vos et al., 2014).  Satellite SST and chlorophyll images reveal dramatic eastward advection of cool ($< 28°$
C) chlorophyll rich upwelled water by the SMC (Vinayachandran, 2004; 2009; de Vos et al., 2014).  Chlorophyll-rich
waters from the southwestern coast of India are also advected by the SMC towards Sri Lanka during the Southwest
Monsoon (Vinayachandran, 2004; 2009; Strutton et al., 2015). Surface chlorophyll concentrations and rates of primary
production along the southern coast of Sri Lanka during the Southwest Monsoon can exceed 10 mgChla m$^{-3}$ (de Vos et
al., 2014) and 1000 mgC m$^{-2}$ d$^{-1}$ (Fig. 6 in Hood et al., 2017), respectively, compared to much lower concentrations and
rates during the Northeast Monsoon when the NMC flows westward (de Vos et al., 2014; Hood et al., 2017).
Vinayachandran (2004; 2009) attribute the productivity response during the Southwest Monsoon to nutrient enrichment
from coastal upwelling driven by monsoon winds. Presumably, these high chlorophyll concentrations and production rates
are associated with diatom blooms. This elevated productivity extends to the east of Sri Lanka during the peak of the
Southwest Monsoon (Vinayachandran et al., 1999; Vinayachandran, 2004; 2009).  This eastward extension into the
southern Bay of Bengal occurs along the path of the SMC (Vinayachandran et al., 1999) and is associated with upward
Ekman pumping east of Sri Lanka.  This Ekman pumping also leads to the formation of the aforementioned Sri Lanka
Dome (Vinayachandran and Yamagata, 1998).

The western side of the northern Indian Ocean transitions during the southwest monsoon to a eutrophic coastal upwelling
system in response to the upwelling favorable winds and currents (Wiggert et al., 2005; Hood et al., 2017 and references
cited therein; Fig. 5 ; Figs. 5 and 6 in Hood et al., 2017; Lakshmi et al., 2020). These changes can be seen in ocean color
data as substantial increases in chla concentrations along the coasts of Somalia, Yemen and Oman (e.g., Brock and
McClain, 1992; Banse and English, 2000; Kumar et al., 2000; Lierheimer and Banse, 2002; Wiggert et al., 2005; George
et al., 2013; Hood et al., 2017). Chlorophyll-*a* concentrations in the western Arabian Sea can exceed 40 mgChla m$^{-2}$ during

the southwest monsoon with production rates > 2.5 gC m$^{-2}$d$^{-1}$ (Marra et al. 1998; Fig. 6 in Hood et al., 2017). However, the environmental conditions vary significantly between the eutrophic coastal zones to the west and the oligotrophic open ocean waters offshore that are influenced by wind-curl induced downwelling to the southwest of the Findlater Jet (Lee et al., 2000; Lakshmi et al., 2020). The surface nitrate and Chl-*a* concentrations decline dramatically from > 10 to < 0.02 μM and from > 1.0 to < 0.2 mgChla m$^{-3}$, respectively, from the west coast to open ocean in the Arabian Sea (Brown et al., 1999; Wiggert et al., 2005; Hood et al., 2017). In general, the phytoplankton community structure transitions to larger cells (diatoms) during the southwest monsoon in the western Arabian Sea (Brown et al., 1999; Tarran et al., 1999; Shalapyonok et al., 2001; Lakshmi et al., 2020). However, small primary producers remain important, even in areas strongly influenced by coastal upwelling (Brown et al., 1999; Lakshmi et al., 2020). In contrast, during the oligotrophic spring and fall intermonsoon periods, surface waters in the western Arabian Sea are dominated by picoplankton (Garrison et al., 2000). Subsurface Chl-*a* maxima are observed between 40 and 140 meters in the central southeastern Arabian Sea during all seasons (Gunderson et al., 1998; Goericke et al., 2000; Ravichandran et al., 2012), at times occurring in layers below the oxyclines of the oxygen minimum zone (Georicke et al., 2000). These features are strongly influenced by mesoscale features (Gundersen et al., 1998).

During the southwest monsoon off Oman and Somalia, the presence of the topographically-locked eddies generate strong offshore flows that advect high nutrient, high Chl-*a* concentrations and coastal phytoplankton communities hundreds of kilometers offshore (Keen et al., 1997; Latasa and Bidigare, 1998; Manghnani et al., 1998; Gundersen et al., 1998; Hitchcock et al., 2000; Lee et al., 2000; Kim et al., 2001). These advective effects can be seen, for example, in association with the Great Whirl off the coast of northern Somalia (Hitchcock et al., 2000) and in the filaments that develop off the Arabian Peninsula during the southwest monsoon (Wiggert et al. 2005; Hood et al., 2017). In contrast, during the northeast monsoon, the circulation and winds transition to downwelling favourable. During the northeast monsoon, cold dry northeasterly winds from southern China and the Tibetan Plateau flow across the northern Arabian Sea. The sheer from these winds, combined with surface cooling and buoyancy-driven convection, drive mixing and entrainment of nutrients that, in turn, promote modest increases in chlorophyll and primary production over the northern Arabian Sea (Wiggert et al., 2000; Wiggert et al., 2005; Fig. 5; Figs. 5 and 6 in Hood et al., 2017). These increases in Chl-*a* have been associated with increased diatom abundance (Banse and McClain, 1986; Sawant and Madhupratap, 1996). In the last decade, however, there appears to have been a shift in the composition of winter phytoplankton blooms in the northern and central Arabian Sea from diatom dominance to blooms of a large, green mixotrophic dinoflagellate, *Noctiluca scintillans* (Gomes et al., 2014; Goes et al., 2020).

During the southwest monsoon, the upwelling-favorable WICC induces upwelling along the west coast of India, which increases Chl-*a* concentrations by more than 70% compared to the central Arabian Sea (Kumar et al., 2000; Naqvi et al., 2000; Luis and Kawamura, 2004; Hood et al., 2017). The increased Chl-*a* concentrations near the coast are associated

with increases in diatom abundance (Sawant and Madhupratap, 1996). However, these increases in Chl-*a* and their offshore extent are modest compared to the western Arabian Sea (Fig. 5; Fig. 5 in Hood et al., 2017). In contrast, during the northeast monsoon the WICC is downwelling-favorable and tends to suppress primary production off the southwestern coast of India. The depletion of nutrients in this region during the northeast monsoon coincides with blooms of *Trichodesmium* and dinoflagellate species (Parab et al., 2006; Matondkar et al., 2007) resulting in the extremely high rates of nitrogen fixation (Gandhi et al., 2011, Kumar et al., 2017). However, as discussed above, further north and offshore, nutrient entrainment enhances phytoplankton biomass and primary production during the northeast monsoon (Wiggert et al., 2000; McCreary et al., 2001; Luis and Kawamura, 2004; Gomes et al., 2014; Goes et al., 2020; Fig. 5). Near-surface Chl-*a* and primary production off the west coast of India (estimated from satellite ocean color measurements) increases from ~9 to 24 mgChla m$^{-2}$ and from ~1 to 2.25 g C m$^{-2}$ d$^{-1}$, respectively, from winter to the summer monsoon (Luis and Kawamura, 2004; Fig. 5; Figs. 5 and 6 in Hood et al., 2017). The elevated productivity during the southwest monsoon is modulated by the coastal Kelvin waves that originate from the Bay of Bengal and propagate along the West Indian Shelf, modifying circulation patterns and upwelling (Luis and Kawamura, 2004).

## 5 Inter-ocean exchange

### 5.1 Indonesian Throughflow

#### 5.1.1 General features

The Indonesian Throughflow (ITF) transfers low-salinity tropical waters from the Pacific to the Indian Ocean via the Indonesian seas (Fig. 10). The ITF is the only tropical oceanic pathway that links ocean basins and plays an important role in the global ocean circulation and climate system (Sprintall et al., 2014; 2019). The simultaneous measurements in the exit channels of the ITF from the International Nusantara Stratification and Transport (INSTANT) program during 2004-2006 (Gordon et al., 2008; Sprintall et al., 2009) suggested that the ITF has a mean transport of 15 Sv into the Indian Ocean. The ITF pathway is composed of many narrow channels within the Indonesian seas, among which about 80% of the total ITF is through the Makassar Strait (Fig. 10, Gordon et al., 2008, 2010). The remaining passages include the Maluku Sea, Lifamatola Passage, Karimata Strait and Sibutu Passage (Fang et al., 2010; Gordon et al., 2012; Susanto et al., 2013).

#### 5.1.2 Variability, dynamics and influence

The interannual variability of the ITF is mainly dictated by the ENSO-related wind forcing through the Pacific waveguide with stronger transport during La Niña years (Meyers, 1996; England and Huang, 2005; Hu and Sprintall, 2016), but the IOD occasionally offsets the Pacific ENSO influences through the Indian Ocean wind variability and Indian Ocean

waveguide (Sprintall and Révelard, 2014; Liu et al. 2015; Feng et al., 2018). For the strong negative IOD event in 2016, the Indian Ocean influence overwhelmed that of the Pacific leading to record low ITF volume transports because of the reduction in the interbasin pressure gradient (Pujiana et al., 2019). Strong wind forcing over the equatorial Indian Ocean triggers equatorial Kelvin waves and influences the ITF variability on intraseasonal, semi-annual and interannual time scales (Drushka et al., 2010; Pujiana et al., 2013; Shinoda et al., 2012). Kelvin waves through the Indian Ocean waveguide are suggested to influence the interannual variability in the tropical Pacific Ocean (Yuan et al., 2013; Pujiana and McPhaden, 2020).

The ENSO cycle also influences the outflowing ITF transport through the salinity effect in the downstream buoyant pool, contributing about 36% of the total ITF interannual transport variation (Hu and Sprintall, 2016; Section 6.1). Fresh anomalies in the buoyant pool during La Nina years can be as large as 0.2 in practical salinity averaged over the upper 180 m of the water column (Phillips et al. 2005). Such salinity anomalies can strengthen the volume transport of the LC through an increase in the zonal density gradient driving stronger southward flow (Feng et al., 2015a). The Inter-decadal Pacific Oscillation/Pacific Decadal Oscillation (IPO/PDO), through modulations of decadal wind stress in the tropical Pacific, has also directly influenced the strength of the ITF (Feng et al., 2011; Hu et al., 2015; Mayer et al., 2018). This has, in turn influenced heat and freshwater transports, causing upper ocean heat content to increase in the southern Indian Ocean (Feng et al., 2010; Schwarzkopf and Böning, 2011; Nidheesh et al., 2013; Sprintall, 2014; Lee et al., 2015; Nieves et al., 2015; Du et al., 2015; Ummenhofer et al., 2017) and produced interhemispheric contrasts in sea surface temperature (Dong and McPhaden, 2016). During the negative IPO phase, such as during the hiatus in warming of the globally averaged surface atmosphere (1998-2012), enhanced trade winds in the Pacific strengthened the ITF volume and heat transport into the Indian Ocean, driving a rapid warming trend in the Southern Indian Ocean (England et al., 2014; Nieves et al., 2015; Lee et al., 2015; Liu et al., 2015, Zhang et al., 2018). Contributions from air-sea exchanges (Jin et al. 2018a,b) have also been suggested to be important, as has a reduction in the oceanic heat exported from the Indian Ocean at its southern boundary (Lisa Beal, personal communication).

Using a combination of theory, ocean reanalyses, OGCM simulations, and coupled climate model simulations, Jin et al. (2018a,b) found eastern and western Indian Ocean heat content to be affected by remote Pacific forcing through two distinct mechanisms: oceanic influences transmitted through the ITF and the atmospheric bridge. The intensified freshwater input within the Maritime Continent during the past decade was found to strengthen the ITF and its heat and freshwater transports into the Indian Ocean, causing significant warming and freshening trends and accelerated sea-level rise in the eastern Indian Ocean (Hu and Sprintall, 2017a, 2017b; Zhang et al., 2018; Jyoti et al., 2019). The decadal enhancement of the ITF transport has increased upper ocean heat content anomalies in the southeast Indian Ocean and increased the likelihood of marine heatwaves off the west coast of Australia (Feng et al., 2015b; Section 6.4).

## 5.2 Agulhas Leakage

### 5.2.1 General features

At the tip of Africa, the southward-flowing Agulhas Current retroflects with most of the flow heading eastwards along the northern edge of the ACC, recirculating back into the Indian Ocean (Fig. 10, Section 4.2.2). Around 20-30% of the Agulhas Current enters the Atlantic Ocean as Agulhas leakage in the form of Agulhas rings and cyclones (van Sebille, 2010a). Agulhas leakage estimates are sensitive to the definition used to calculate the leakage, ranging roughly between 10 and 20 Sv (van Sebille et al., 2010b; Beron-Vera et al., 2013; Cheng et al., 2016; Holton et al., 2017). Bars et al. (2014) proposed an algorithm to measure Agulhas leakage anomalies using absolute dynamic topography data from satellites.

The division of flow between Agulhas Leakage and Agulhas retroflection can be influenced by the upstream Agulhas Current. In a Lagrangian particle tracking experiment, van Sebille et al. (2009) found that a weaker Agulhas Current, detaching farther downstream and generating anti-cyclonic vorticity, potentially leads to more Agulhas leakage and larger Indian-Atlantic inter-ocean exchange. However, eddy-resolving model results suggest that as model resolution increases, the sensitivity of the leakage to Agulhas Current transport anomalies is reduced (Loveday et al., 2014). In addition, the ITF potentially influences the Agulhas leakage (Le Bars et al., 2013) as model outputs suggest that the Indian Ocean contributes 12.6 Sv to the Agulhas leakage, half of which is from the ITF (Durgadoo et al., 2017).

### 5.2.2 Variability, dynamics and influence on climate

The magnitude of the Agulhas leakage is controlled by wind forcing including the trade winds and the Southern Hemisphere Westerlies (e.g., Durgadoo et al., 2013). The poleward shift in the Southern Hemisphere westerlies associated with anthropogenic forcing induced a clear increase in the Agulhas leakage during 1995-2004 as shown in numerical simulations (Biastoch et al., 2009; Biastoch and Böning, 2013). Increased wind stress curl in the South Indian Ocean associated with the southward shift of westerlies led to significant warming in the Agulhas Current system since the 1980's (Rouault et al., 2009); however further work showed that this is due to an increase in eddies leading to a broadening of the current as opposed to intensification (Beal and Elipot, 2016). Given the non-linear nature of Agulhas leakage, the difficulty of observing it and ocean model biases in the region, quantifying Agulhas leakage is very challenging (Holton et al., 2017). At seasonal time scales, the Agulhas leakage variability is controlled by eddies, however recent studies have shown that eddies might not contribute as significantly to leakage as was thought and the non-eddy leakage transport is likely to be constrained by large-scale forcing at longer time scales (e.g., Cheng et al., 2018). A recent study shows that the subsurface signal from the ENSO cycle influences the Agulhas leakage through Rossby waves with a time lag of 2 years (Paris et al., 2018).

The Agulhas leakage carries warm and saline water from the Indo-Pacific Ocean into the Atlantic Ocean. The Agulhas leakage has been suggested to influence the Atlantic Meridional Overturning Circulation strength (AMOC; Beal et al. 2011; Weijer and van Sebille, 2014; Biastoch et al. 2015) and modify the AMOC convective stability (e.g., Haarsma et al., 2011; Caley et al., 2012; Castellanos et al., 2017). It is suggested that the increases in the Agulhas leakage due to anthropogenic warming during the past decades would act to strengthen the Atlantic overturning circulation (e.g., Beal et al., 2011).

The Agulhas leakage is an important source of decadal variability in the AMOC through Rossby waves (Biastoch et al., 2008; 2015). Source waters from the Agulhas Current take more than four years and mostly one to four decades to arrive in the North Atlantic Ocean (van Sebille et al., 2011; Rühs et al., 2013). The increased Agulhas leakage during 1995-2004 has contributed to the salinification of the South Atlantic thermocline waters (Biastoch et al., 2009). Hindcast experiments suggest that the Agulhas leakage increased by about 45% during the 1960s-2000s, leading to the observed warming trend in the upper tropical Atlantic Ocean (Lübbecke et al., 2015).

## 5.3 Supergyre connection to the South Pacific

The extreme strong westerly wind stress in the Southern Hemisphere gives rise to a wide and energetic subtropical supergyre (Figure 16), the Southern Hemisphere supergyre, that connects three ocean basins (e.g., Ridgway and Dunn, 2007; Speich et al., 2007; Lambert et al., 2016; Maes et al., 2018; Cessi, 2019). Although the near-surface circulation is eastward across the southern Indian Ocean, there are subsurface westward flows beneath (Section 4.2.3; Schott and McCreary 2001; Domingues et al. 2007; Furue et al. 2017), and the depth-integrated circulation reveals the westward return flow of the equatorward side of the Indian Ocean's anti-clockwise subtropical gyre. In Figure 16, the southern side of the Indian Ocean subtropical gyre extends eastward south of Australia to connect with the western Pacific subtropical gyre. The return flow is accomplished via a pathway that includes the East Australian Current, the South Pacific's western boundary flow; the Tasman Leakage, a westward flow south of Tasmania that carries Pacific Ocean water back to the Indian Ocean (distinct from the Flinders Current that hugs the continental slope, Duran et al 2020; Section 4.2.4); and northwestward flow in the eastern Indian Ocean to close the circulation. The ITF and Leeuwin Current are also part of the supergyre, connecting the Indian and Pacific Oceans through the Indonesian seas (e.g. Ridgway and Dunn, 2007).

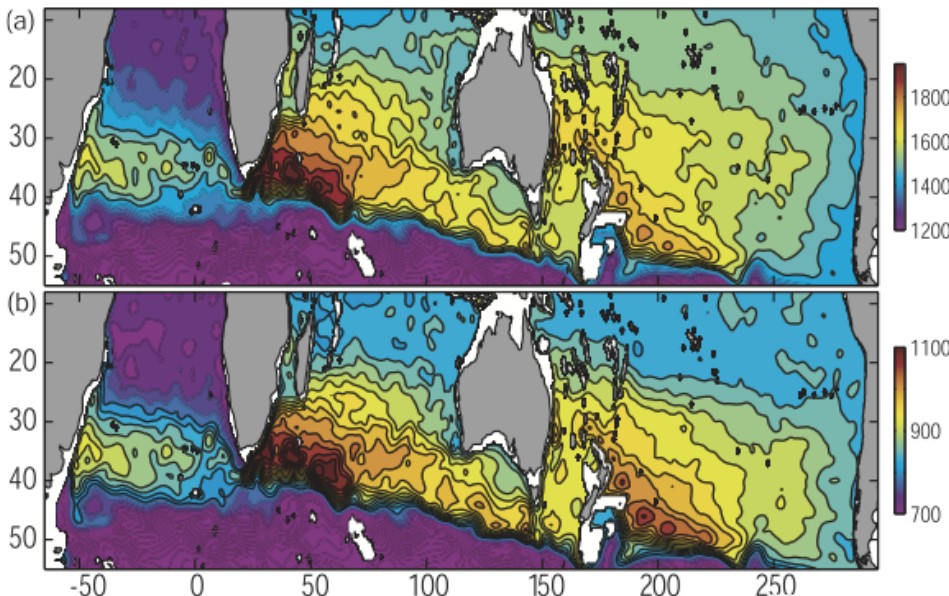

**Figure 16: The interbasin supergyre system for the Pacific and Indian Oceans as shown by the depth-integrated steric height (a) $P_{0/2000}$, and (b) $P_{400/2000}$, derived from the CARS climatological temperature and salinity fields. The contour interval in (a) is 50 m$^2$ and in (b) is 25 m$^2$. Taken from Ridgway and Dunn 2007.**

The supergyre is the subtropical gyre of the southern hemisphere. As such, its flow is primarily determined by the westward integration of wind stress curl from the eastern boundaries as determined by Sverdrup dynamics. The latitudinal position of the Subtropical Front at the southern edge of the supergyre is found to be controlled by strong bottom pressure torque due to the interaction between the ACC and the ocean floor topography (De Boer et al., 2013). According to one analysis in SODA (Simple Ocean Data Reanalysis), the water masses in the supergyre became cooler and fresher and shifted southward by about 2.5$^{\circ}$ due to changes in the basin-scale wind forcing during 1958–2007 (Duan et al., 2013). A recent study using altimeter observations shows a clear strengthening of the Southern Hemisphere supergyre in all three oceans since 1993 as indicated in the large trends of sea surface height and their contrast. Argo observations and ECCO assimilations suggest that the strengthening extends to deeper than 2000 m (Qu et al., 2019). The spin-up of the Southern Hemisphere supergyre is attributed to the poleward shift and strengthening of westerly winds that are linked to an increasingly positive southern annular mode (Qu et al., 2019).

**5.4 Roles of salinity in inter-ocean exchange**

Ocean salinity is one of the basic variables that determines the oceanic stratification, sea level change and climate change (e.g., Llovel and Lee, 2015; Kido and Tozuka, 2017; Sprintall et al., 2019). However, the role of salinity in ocean circulation has been largely underestimated until the recent decade when *in situ* observations of subsurface and surface

salinity from Argo and satellite salinity missions became available. These new observations have revolutionized our understanding of the influence of salinity on ocean circulation and dynamics (Vinogradova et al. 2019, and references therein).

Four major processes control the salinity in the Indian Ocean: net air-sea fluxes (evaporation minus precipitation), freshwater inflow from large rivers in the Bay of Bengal, inflow of relatively fresh waters from the Pacific Ocean via the Indonesian Throughflow, and inflow of saltier waters from the Red Sea and the Persian and Arabian Gulfs. These different drivers combine to give the Indian Ocean salinity its unique flavour: a strong east-west gradient in the North Indian Ocean (salty in the Arabian Sea and fresh in the Bay of Bengal) and strong north-south gradients in the South Indian Ocean (fresh in the tropics, and salty in the subtropics) (Fig. 4).

Salinity is a crucial variable to understand Indian Ocean dynamics. For instance, salinity has strong ties with the Indian Ocean Dipole (e.g., Du and Zhang, 2015; Durand et al., 2013; Grunseich et al., 2011; Kido and Tozuka, 2017; Nyadjro and Subrahmanyam, 2014; Zhang et al. 2016; Section 6.2), the EGC (Menezes et al., 2013; Section 4.2.3), LC transport, Ningaloo Niño and marine heatwaves off western Australia (e.g., Feng et al., 2015a), and the El Niño/La Niña climate mode (e.g., Hu and Sprintall, 2016; Zhang et al., 2016). Salinity plays an essential role in the dynamics of the seasonal Wyrtki Jets in the equatorial zone (e.g., Masson et al., 2003), extra-equatorial Rossby waves (Heffner et al., 2008; Menezes et al., 2014b; Vargas-Hernandez et al., 2015; Banks et al., 2016), Madden-Julian and Intraseasonal Oscillations (e.g., Grunseinch et al., 2013; Guan et al., 2014; Subrahmanyam et al., 2018), barrier-layer dynamics (e.g., Drushka et al., 2014; Felton et al., 2014), and the North Indian Ocean (e.g., D'Addezio et al., 2015, Fournier et al., 2017; Mahadevan et al., 2016; Nyadjro et al., 2011, 2012, 2013; Wilson and Riser, 2016; Spiro Jaeger and Mahadevan, 2018).

Salinity variability within the Indonesian Seas has been shown to control the transport of the ITF. Andersson and Stigebrandt (2005) proposed that a downstream buoyancy pool in the outflowing ITF region acts to regulate the ITF transport. Gordon et al. (2003, 2012) pointed out that low salinity surface water from the South China Sea is drawn into the Java Sea. Combined with the monsoonal precipitation over the Maritime Continent and seasonal monsoon winds, this freshwater plug contributes to the seasonal fluctuation of the Makassar Strait Throughflow transport and inhibits the inflow of tropical Pacific surface water from the Mindanao Current (e.g., Gordon et al., 2012; Lee et al., 2019). Recently, Hu and Sprintall (2016) found that about 36% of the interannua---l ITF transport is attributable to the salinity effect associated with freshwater input anomalies due to the ENSO cycle. Jyoti et al. (2019) further examined this salinity effect and found that the unprecedented sea-level rise in the southern Indian Ocean since the beginning of the 21st Century is attributed to the accelerated heat and freshwater intrusion by the ITF. A significant strengthening of the ITF transport in the 2000s has given rise to a subsequent warming and freshening of the eastern Indian Ocean (e.g., Hu and Sprintall, 2017a, 2017b,

Section 6.1). The southeast Indian Ocean is one of the few places in the global ocean where the halosteric component of
sea level rise is as large as the thermosteric component (Llovel and Lee, 2015).

## 6 Modes of Interannual Climate Variability in the Indian Ocean

### 6.1 ENSO teleconnection and the Indian Ocean Basin mode

ENSO influences the Indian Ocean circulation through the Pacific-to-Indian Ocean oceanic waveguide and atmospheric
teleconnections. Through the atmospheric bridge, El Niño conditions in the Pacific induce an anticyclonic wind anomaly
pattern in the southeast Indian Ocean (Xie et al., 2002), whereas La Niña induces a cyclonic wind anomaly pattern (Feng
et al., 2013). The ENSO teleconnection also drives SST variability over the western Indian Ocean during ENSO
development. The tropical Indian Ocean experiences prolonged warming (cooling) that peaks in the following boreal
spring and persists into boreal summer, after the decay of El Niño (La Niña) events, the so-called Indian Ocean Basin
(IOB) mode (Yang et al., 2007). The westward propagating Rossby waves induced by ENSO may also help sustain the
warming (cooling) of the tropical Indian Ocean (Xie et al., 2002), fueled by regional air-sea coupling (Du et al., 2009).
The IOB warming has a capacitor effect for El Niño to influence boreal summer climate, such as for the Indian monsoon
(Zhou et al., 2019), and remote impacts in the northwest Pacific (Xie et al., 2009, 2016), including China and Japan (Hu
et al., 2019). Details of the Indo-Western North Pacific capacitor effect are summarized in Xie et al. (2016) and Kosaka et
al. (2021). The relationship between ENSO and IOB varies on decadal time scales (e.g., Xie et al., 2010; Chowdary et al.,
2012) and under global warming scenarios. The IOB warming tends to persist longer after El Niño events according to
CMIP5 model simulations (Zheng et al., 2013).
The ITF variability lags ENSO by 8-9-months, found in ocean model results (England and Huang, 2005) and derived from
the geostrophic transport across an Australia-Indonesia XBT section (Liu et al., 2015). The variability of the ITF transport
drives sea level and upper ocean heat content anomalies in the southeast Indian Ocean. Through the waveguide, ENSO
has a direct influence on the strength of the Leeuwin Current (Section 4.2.4), with a stronger poleward volume and heat
transport during a La Niña event (Feng et al., 2008). A stronger Leeuwin Current during La Niña events leads to greater
baroclinic instability of the current and enhanced generation of eddies that leads to interannual variability of the eddy
kinetic energy in the southeast Indian Ocean (Feng et al., 2005; Zheng et al., 2018). The increase of the ITF transport and
enhancement of rainfall in the Indonesian Seas during strong La Niña events can drive up to 0.2-0.3 psu freshening
anomalies in the upper southeast Indian Ocean (Phillips et al., 2005; Feng et al., 2015a; Hu and Sprintall, 2017a; Section
5.1.2), which may have a compound effect in accelerating the Leeuwin Current (Feng et al., 2015a). Both ENSO and the
IOD (see Section 6.2) influence the ITF and thus the exchange of heat from the Pacific into the Indian Ocean, but in
concurrent IOD and ENSO events it appears that the influence from the IOD dominates (Sprintall and Revelard, 2014).
Due to the opposing effects of the winds and dissipation, ENSO induced sea level and upper ocean heat content anomalies
in the southeast Indian Ocean do not propagate far into the western Indian Ocean; instead, wind anomalies generate sea
level and heat content anomalies of opposite signs in the western Indian Ocean through Rossby wave propagations
(Masumoto and Meyers, 1998; Xie et al., 2002; Zhuang et al., 2013; Ma et al., 2019; Volkov et al., 2020; Nagura and
McPhaden, 2021). Thus, the joint forcing of the oceanic waveguide and atmospheric teleconnection results in variations
of meridional overturning circulation and heat transport in the Indian Ocean on a multi-year time scale, in phase with the
ITF variability (Ma et al., 2019).

## 6.2 The Indian Ocean Dipole

There is increasing evidence that positive IOD events are more frequent and intense during the 20th century (e.g., Abram
et al., 2008; Cai et al., 2013; Abram et al., 2020a,b; and references therein). A rare occurrence of three consecutive positive
IOD events took place in 2006-2008 (Cai et al., 2009b). The skewness towards more positive and fewer negative IOD
events (Cai et al., 2009a) is due potentially to an anthropogenically-driven shoaling thermocline in the eastern Indian
Ocean (Cai et al., 2008). The three consecutive positive IOD events rarely occurred in Coupled Model Intercomparison
phase 5 (CMIP5) models and the more recent frequent occurrence was consistent with regional Indo-Pacific Walker
circulation trends (Cai et al., 2009c,d). An anthropogenic contribution was proposed since positive IOD events became
more frequent over the period 1950–1999 in the CMIP5 models. Projected mean-state changes in the Indian Ocean with
stronger easterly winds and a shoaling thermocline in the southeast Indian Ocean during austral spring favour positive
IOD development, with a reduction in skewness between positive and negative IOD events likely (Cai et al., 2013; Figure
17), and a three-fold increase in frequency of extreme positive IOD events by 2100 compared to the previous century (Cai
et al., 2014a). However, model biases in Indian Ocean mean-state and IOD variability challenge these projected changes:
models with excessive IOD amplitude bias tend to project a strong IOD-like warming pattern and increase in extreme
pIOD occurrences, consistent with an enhanced Bjerknes feedback, and hence the projected IOD changes could represent
spurious artefacts of model biases (Li et al., 2016). Yet, paleoclimate evidence supports trends observed in recent decades:
based on a millennial IOD reconstruction from corals, extreme positive IOD events, as were observed in 1997 and 2019,
were historically rare (Abram et al., 2020b). In the reconstruction, only ten extreme positive IOD events occurred and yet
four events occurred in the last 60 years (Abram et al., 2020b). The increase in event frequency and intensity highlights
the need to improve preparedness in regions affected by IOD events to minimize future climate risks posed by them.

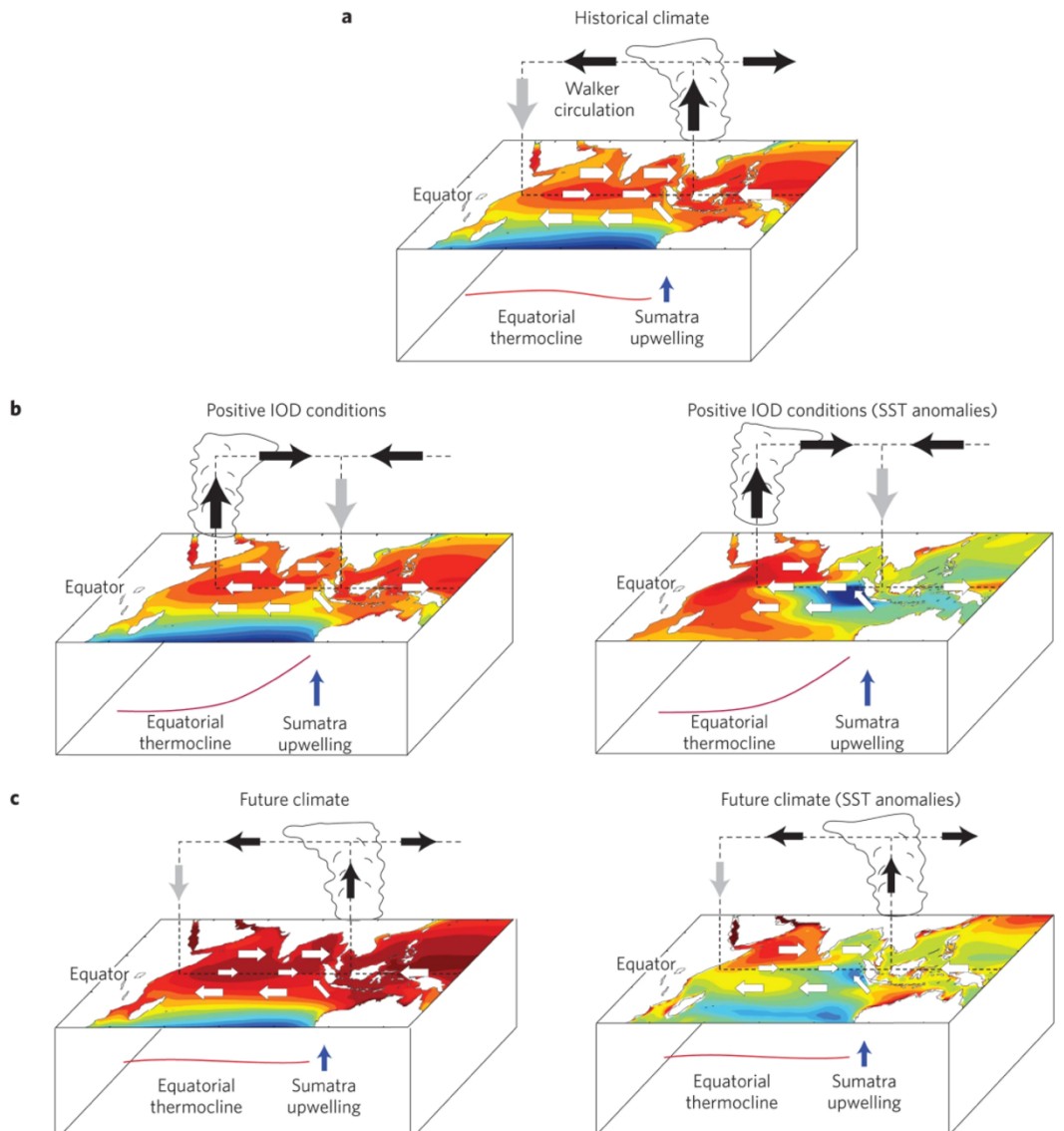


**Figure 17: Historical austral spring mean climate and positive IOD conditions for the twentieth century, and future austral spring mean climate. a, Historical mean climate, indicating SSTs, surface winds, the associated atmospheric Walker circulation, the mean position of convection and the thermocline. In the western Indian Ocean, the descending branch is broad and not well-defined, as indicated by a grey arrow. b, Typical conditions during a positive IOD event. c, Projected future mean climate based on a CMIP5 multi-model ensemble average. Diagrams with total SST fields are shown on the left; diagrams with SST anomalies referenced to the 1961–1999 mean for b,**

**and referenced to the basin mean for c, are shown on the right. Reprinted from Cai et al. (2013) with permission**

While model simulations and paleo proxy records suggest changes in the frequency and magnitude of IOD events in a
warming climate, there is less observational evidence from other sources. Given the short observational record in the
Indian Ocean, the role of decadal to multi-decadal variability across the broader Indo-Pacific region has recently emerged
as a compounding factor: the number and frequency of IOD events have been observed to vary on decadal timescales.
Decadal variations in SST featuring an IOD-like out-of-phase pattern between the western and eastern tropical Indian
Ocean have been linked to the PDO (Krishnamurthy and Krishnamurthy, 2016) or IPO (Dong et al., 2016). A combination
of processes transmits the signal from the Pacific to the Indian Ocean through both the atmospheric and oceanic bridges,
leading to variations in the subsurface temperature structure in the Indian Ocean (Zhou et al., 2017; Jin et al., 2018a).
Decadal modulations of the background state of the eastern Indian Ocean thermocline depth can thus pre-condition the
Indian Ocean to more or less IOD events (Annamalai et al., 2005). Consequently, positive IOD events were unusually
common in the 1960s and 1990s with a relatively shallow eastern Indian Ocean thermocline, while the deeper thermocline
in the 1970s and 1980s was associated with frequent negative IOD and rare positive IOD events (Ummenhofer et al.,
2017). The Indian Ocean stands out as a region with high skill in decadal predictions (Guemas et al., 2013) and improved
understanding of decadal modulation of IOD events can aid in decadal prediction efforts for the Indian Ocean region.
The relationship between ENSO and the IOD has been subject to ongoing debate. Recent research has shown that around
two-thirds of IOD variability arises as a remote response to ENSO (Stuecker et al., 2017; Yang et al., 2015), with the
remaining variability being independent of ENSO. Stuecker et al. (2017) argue that the ENSO-driven IOD can be seen as
a combination of remotely driven wind and heat flux anomalies modulated by seasonally-varying Bjerknes feedback in
the Indian Ocean. Further, they suggest that the ENSO-independent IOD events arise out of white noise atmospheric
forcing coupled to these feedbacks (Stuecker et al., 2017). Variability internal to the Indian Ocean basin and unrelated to
ENSO, arising from ocean-atmosphere feedback processes, does however modulate the evolution of IOD events and can
lead to early termination of IOD events; as a result, including internal variability improves the predictability of the IOD
(Yang et al., 2015). IOD variability internal to the Indian Ocean resembles recharge oscillator dynamics for ENSO, but
equatorial heat content is less effective as a precursor for the IOD than for ENSO because of the strong impact of remote
forcing from the Pacific on the IOD. Internal Indian Ocean dynamics however may contribute to the biennial nature of
the IOD through the cycling of Kelvin/Rossby wave energy across the basin (McPhaden and Nagura, 2014). The
relationship between ENSO and the IOD is not only one-way: IOD events have also been shown to influence the
development of ENSO in the following year (Izumo et al., 2010; Wang et al., 2019; Cai et al., 2019; and references therein).
Different types of IOD events have been described, each with distinct evolution and regional impacts (Du et al., 2013;
Endo and Tozuka, 2016). Du et al. (2013) distinguished three types of IOD events according to the timing of their peak

amplitude and overall duration: 'unseasonable' events that develop and mature mostly within June-August (JJA), 'normal' events that develop and mature mostly within September-November (SON), and 'prolonged' events that develop in JJA and mature in SON, with the latter two described as the canonical IOD events (Du et al., 2013). The unseasonable IOD events have only been observed since the mid-1970s and have been suggested to be a response to the rapidly warming Indian Ocean SST and a weakened Walker circulation during austral winter (Du et al., 2013). The seasonal evolution and type of ENSO also seems to play a role in determining the IOD evolution and type, with atmospheric influences transmitted through variations in the Walker Circulation and oceanic ones through anomalous oceanic Rossby waves affecting timing and evolution of IOD events, especially during their developing phase (Guo et al., 2015; Zhang et al. 2015; Fan et al., 2017). However, Sun et al. (2015) suggested more IOD events independent of ENSO since the 1980s, along with higher correlations between the IOD and Indian summer monsoon activity, likely due to mean-state change in the tropical Indian Ocean due to weaker equatorial westerlies. The relationship between ENSO and the IOD has weakened in recent decades, linked to changes in the ENSO-induced rainfall anomalies over the Maritime Continent (Han et al., 2017).

Recent advances in understanding variability and change in IOD characteristics have implications for the relationships between SST and regional rainfall patterns in Indian Ocean rim countries. For example, different types of IOD events exhibit distinct regional impacts, with only the canonical events associated with enhanced rainfall over East Africa due to the low-level moisture convergence over the region (Endo and Tozuka, 2016). The effect of Indian Ocean SST on East African rainfall is most pronounced during the short rains (September-November), though Williams and Funk (2011) argued that warming Indian Ocean SST in recent decades was also associated with reduced long rains for the March-June season in Ethiopia and Kenya. Changes in the tropical atmospheric circulation across the Indo-Pacific on multi-decadal timescales (Vecchi and Soden, 2007; L'Heureux et al., 2013) have further implications for the relationship between Indian Ocean SST and regional rainfall: When the Pacific Walker cell weakened and the Indian Ocean one strengthened post-1961, the East African short rains became more variable and wetter (Nicholson, 2015). Similarly, Manatsa and Behera (2013) described an epochal strengthening in the relationship between the IOD and East African rainfall post-1961, with 73% of short rain variability in East Africa explained by the IOD, up from 50% in previous decades. After 1997, this increased further to 82%, explaining spatially coherent events across the region and frequent rainfall extremes (Manatsa and Behera, 2013). Recent observed and projected changes in frequency and intensity of IOD events highlight the increasing need for preparedness in vulnerable regions affected by these events. One such event is the recent 2019 positive IOD, the largest IOD on record since the 1960s (Du et al. 2020), which was linked to unusual hydroclimate around the Indian Ocean rim and further afield. It was linked to extreme rainfall and floods in East Africa (e.g., Wainwright et al., 2021), anomalously wet Indian monsoon season (Ratna et al., 2021), abnormally warm conditions in many parts of East Asia (Doi et al., 2020), unusually wet subsequent summer monsoon season in Japan and China due to downwelling Rossby waves that had affected Western Pacific SST (Takaya et al. 2020; Zhou et al., 2021), and was seen as a contributing factor to the severe bushfire season experienced in Australia in 2019/2020 (e.g., Wang and Cai, 2020). The 2019 IOD was unique

in that it developed independently from any El Nino events and resulted from westward propagating Rossby waves in the
southwest tropical Indian Ocean (Du et al., 2020) and/or an interhemispheric pressure gradient over the Maritime continent
(Lu and Ren, 2020).

### 6.2.1 Biogeochemical Variability

IOD events are associated with distinct changes in primary productivity, as measured by chlorophyll. During positive IOD
events, increased chlorophyll indicative of phytoplankton blooms is apparent in the normally oligotrophic eastern Indian
Ocean in fall (Wiggert et al., 2009; Currie et al., 2013). Positive chlorophyll anomalies occur in the southeastern Bay of
Bengal in boreal winter, while negative anomalies are observed over much of the Arabian Sea and southern tip of India.
In a case study of the 2006 positive IOD event, Iskandar et al. (2010) using an eddy-resolving biophysical model found
the offshore chlorophyll signal in the southeastern Indian Ocean to be associated with regions of high eddy kinetic energy
implying that cyclonic eddies injected nutrient-rich water into the upper layer enabling the bloom. Currie et al. (2013)
emphasize the importance of assessing the relative contributions of IOD events and remote impacts from ENSO on primary
productivity in the Indian Ocean through their respective influence on upper-ocean properties for improved understanding
and ultimately predictions of productivity, ecosystems, and fisheries within the basin. Little attention has been paid so far
to resultant effects of these blooms on biogeochemical cycling (Wiggert et al., 2009).

### 6.3 The subtropical Indian Ocean Dipole

The subtropical Indian Ocean Dipole (SIOD) is a climate mode in the southern Indian Ocean, which tends to arise and
peak in the austral summer (Behera and Yamagata, 2001). During the SIOD's positive phase, the climate mode has positive
SST anomalies in the southwestern Indian Ocean and negative SST anomalies in the northeastern region (Behera and
Yamagata, 2001; Suzuki et al., 2004; Hermes and Reason, 2005). During the positive phase, enhanced precipitation occurs
over southern Africa (Behera and Yamagata 2001; Reason 2001, 2002). Recent studies have shown that the SIOD affects
the Indian summer monsoon rainfall (Terray et al., 2003), rainfall over southwestern Australia (England et al., 2006) and
tropical cyclone trajectories in the southern Indian Ocean (Ash and Matyas, 2012).
Initially, SST anomalies associated with the SIOD were considered to be generated directly by latent heat flux anomalies
(Behera and Yamagata, 2001). However, recent studies (Morioka et al. 2010, 2012) based on a mixed layer heat budget
analysis revealed the importance of mixed layer depth anomalies generated by latent heat flux anomalies. Wind anomalies
associated with the anomalous Mascarene High suppress latent heat loss and shoal the mixed layer in the southwestern
part, while latent heat release is enhanced and the mixed layer deepens anomalously in the northeastern part (Morioka et
al. 2010, 2012). With these changes in the upper ocean heat capacity, warming of the surface mixed layer by the
climatological shortwave radiation is enhanced in the southwestern part and becomes less effective in the northeastern
part. As a result, the dipole SST anomalies appear in the southern Indian Ocean.
Because the above mechanism operates more effectively as the thickness of the mixed layer becomes thinner, the return
period of the SIOD is becoming shorter associated with the shoaling trend of the mixed layer (Yamagami and Tozuka,
2015). Whether this mechanism is associated with decadal-to-interdecadal variations and/or global warming awaits further
study. Many coupled models are relatively successful in simulating the SIOD with some biases in the location and structure
of the SST anomaly (Kataoka et al., 2012). However, no study has examined if the SIOD is modulated by climate modes
of variability with decadal-to-interdecadal timescales or changes with global ocean warming.
**6.4 Ningaloo Niño and marine heatwaves in the Indian Ocean**
The Ningaloo Niño (Niña) phenomenon is an interannual climate mode associated with anomalously warm (cold) water
in the eastern Indian Ocean (Feng et al., 2013; see Figure 18). This mode is seasonally phase-locked, with a peak during
austral summer (Kataoka et al., 2014). The mode exerts significant impacts on rainfall over Australia (Kataoka et al., 2014)
and affects marine ecosystems and fisheries (e.g. Pearce et al. 2011). The phenomenon can alter biological productivity,
with negative chlorophyll anomalies during Ningaloo Niño (Narayanasetti et al., 2016). Ningaloo Niños can develop in
response to remote ENSO forcing from the western Pacific transmitted as a coastally trapped wave (Kataoka et al., 2014).
During the La Niña events, high sea level anomalies propagate poleward along the west coast of Australia, intensifying
the Leeuwin Current and causing poleward advection of heat and anomalously warm waters (e.g. Benthuysen et al., 2014;
Section 4.2.4). Poleward transport of tropical, low salinity waters can further enhance the total geostrophic transport of
the Leeuwin Current (Feng et al., 2015a).

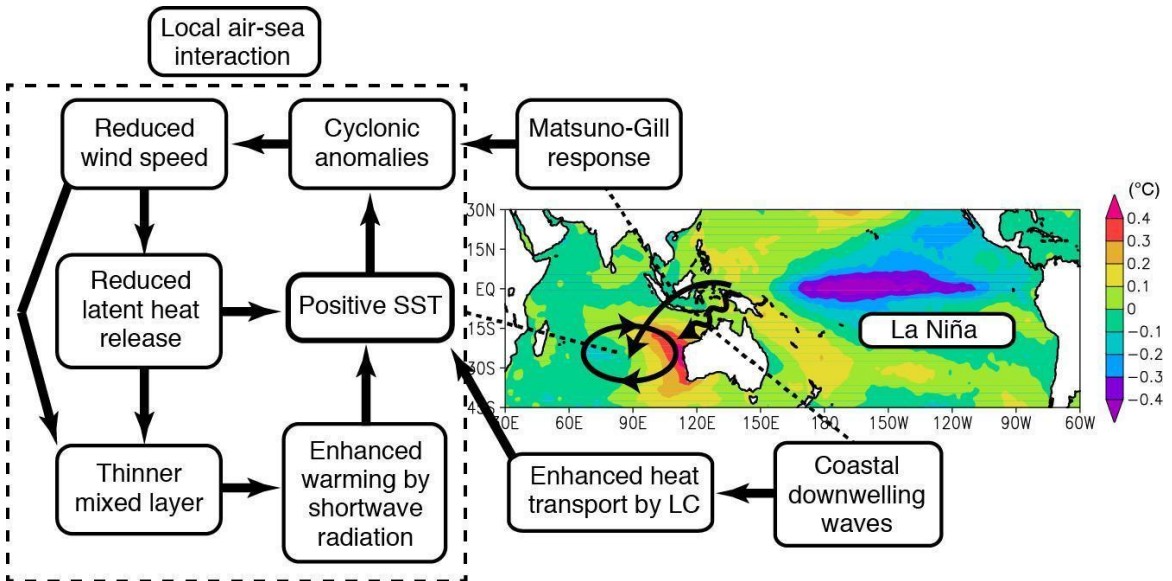


**Figure 18: Schematic diagram illustrating generation mechanisms (i.e. local air-sea interaction, atmospheric teleconnection, and oceanic wave propagation) of the Ningaloo Niño. SST anomalies are regressed against the Ningaloo Niño Index to illustrate typical SST anomalies associated with the phenomenon.**

Atmospheric teleconnection can further enhance the development of Ningaloo Niño. A reduction in southerly winds over the shelf, which would strengthen the Leeuwin Current, can arise through a Gill-type response with low sea level pressure anomalies in the southeast Indian Ocean owing to the Niño3.4 SST anomalies (Feng et al., 2013; Tozuka et al., 2014). Ningaloo Niños can arise from local air-sea interactions off western Australia, through the wind-evaporation-SST feedback during its initial stage (Marshall et al., 2015) and coastal SST-wind-Leeuwin Current (Bjerknes) feedback (Kataoka et al. 2014). In the coastal feedback mechanism, positive SST anomalies lead to northerly alongshore wind anomalies and coastal downwelling anomalies, causing enhancement of the positive SST anomalies (Kataoka et al., 2014). During the Ningaloo Nino's development phase, estimates of air-sea heat flux contributions have been found to be dependent on products and their resolution and bulk flux algorithms (Feng and Shinoda, 2019). Since the late 1990s, Ningaloo Niño events have occurred more frequently (Feng et al., 2015b). This decadal increase is corroborated by coral proxy records of Leeuwin Current strength, with the most extreme SST anomalies associated with Ningaloo Niños occurring since 1980 (Zinke et al., 2014).

More generally, marine heatwaves refer to prolonged, extremely warm water events. Over the past decade, most studies on marine heatwaves in the Indian Ocean have focused on the eastern sector of the Indian Ocean. Major events in the Indian Ocean have been associated with phases of ENSO. Along the west coast of Australia, marine heatwaves have occurred predominantly at subtropical reefs during La Niña events due to increased heat transport (Zhang et al., 2017).

The term "marine heatwave" was first coined owing to a +5°C warm water event in 2011 off Western Australia during a
strong La Niña (Pearce et al., 2011). The 2011 event was associated with the strongest recorded Leeuwin Current transport
anomaly, bringing warm tropical waters south, and was partly due to air-sea heat fluxes (Feng et al., 2013; Benthuysen et
al., 2014).
Across Australia's northwestern shelf, marine heatwaves have been found to occur at tropical coral reefs from El Niño
due to solar radiation and a weakened monsoon (Zhang et al., 2017). During the strong El Niño of 2015-2016, the southeast
tropical Indian Ocean experienced the warmest and longest marine heatwave on record, with weakened monsoon activity
and anomalously high air-sea heat flux into the ocean (Benthuysen et al., 2018). The anomalously warm water conditions
persisted into winter, during one of the strongest negative IOD events (Benthuysen et al., 2018). The 2016 marine heatwave
was associated with coral bleaching spanning Australia's inshore Kimberley region to remote coral reef atolls (Gilmour et
al., 2019). More broadly across the Indian Ocean during 2016, marine heatwaves have been studied in terms of their
ecological impacts, such as coral bleaching in the western Indian Ocean (e.g. Gudka et al., 2018), the Maldives (e.g.
Ibrahim et al., 2017) and consequences for fishes in the Chagos Archipelago (Taylor et al., 2019).
Trends in marine heatwave metrics indicate widespread regions across the Indian Ocean where events have increased in
frequency, based on SST from 1982-2016, especially in the central and southwestern sectors (Oliver et al., 2018). Over
the same time period, the duration and intensity of marine heatwaves have increased in the Indian Ocean and globally
(Oliver et al. 2018, Marin et al. 2021). Primary climate modes of variability correlated with an increased occurrence of
marine heatwaves include the following: (1) the positive phase of the Dipole Mode Index for the northwestern sector, the
tropical sector, and south to the Seychelles Islands, (2) the positive phase of the Niño3.4 index for the south-central sector,
and (3) the negative phase of the El Niño Modoki index, which measures the strength of the Central Pacific ENSO, for the
eastern Indian Ocean (Holbrook et al., 2019). While the marine heatwaves in the eastern Indian Ocean have been well
documented, there have been fewer studies into the physical mechanisms causing marine heatwaves across the basin and
other regions and less confidence, for example in the Bay of Bengal, in the local processes causing reported events on a
range of time scales (Holbrook et al. 2019). There are indications that increased extremes in El Niño (Cai et al., 2014b)
and La Niña events (Cai et al., 2015) due to mean ocean warming trends increase the likelihood of marine heatwave
occurrence in the southeast Indian Ocean (Zhang et al., 2017).
**6.5 Monsoon variability and links to the Indian Ocean**
Several monsoon systems surround the Indian Ocean, notably the South Asian monsoon, the East Asian monsoon and
the Australian monsoon. These monsoon systems are remotely influenced by global coupled modes of variability such as
ENSO, which is often associated with dry conditions in the South Asian monsoon (e.g., Rasmusson and Carpenter,
1983; Ropelewski and Halpert, 1987) and Australian monsoon (e.g., Risbey et al., 2009; Jourdain et al., 2013), although

the relationship with the Indian monsoon has recently weakened (e.g., Kumar et al., 1999). In the Indian Ocean, the IOD has a strong influence on the Asian monsoon systems, but is weak during the Australian monsoon period. The IOD tends to oppose the ENSO teleconnection to the South Asian monsoon by enhancing monsoon rainfall (e.g., Ashok et al., 2004; Chowdary et al., 2015; Krishnaswamy et al., 2015; Pokhrel et al., 2012). However, the exact combination of SST patterns between the Indian Ocean and the Pacific is crucial for determining the rainfall response in the Asian monsoons (e.g., Lau and Wu, 2001; Ratna et al., 2020; Yuan and Yang, 2012), and the relative strengths of the teleconnections have varied over time (Krishnaswamy et al., 2015). Furthermore, there is evidence that the Indian Ocean forcing of the South Asian monsoon may be primarily driven by ENSO, with pure IOD events only weakly influencing monsoon rainfall (Cretat et al., 2017).

The monsoon systems around the Indian Ocean tend to vary in phase and are also linked to the western North Pacific Monsoon (e.g., Gu et al., 2010). There is a biennial oscillation in the strength of the monsoon systems, with a strong Asian monsoon preceding a negative IOD and coinciding with cold eastern Pacific SSTs, followed by a strong Australian monsoon and subsequently by a reversal in the SST patterns (Loschnigg et al., 2003; Meehl & Arblaster, 2011). Thus, each monsoon system interacts with the ocean dynamics and thermodynamics and with the other monsoon systems through a complex set of teleconnections.

At a regional scale, upwelling in the Arabian Sea reduces rainfall along the western Ghats of India during the monsoon due to a reduction in evaporation and water vapour transport (Izumo et al., 2008). Moisture fluxes across the Arabian Sea are crucial to accurate simulation of the Indian Monsoon, yet many models fail to accurately capture these (Levine and Turner, 2012). In the Bay of Bengal, the shallow surface mixed layer, supported by the vertical salinity gradient, leads to rapid variations in SST (e.g., Sengupta and Ravichandran, 2001; Vecchi and Harrison, 2002) that interact with intraseasonal oscillations (Gao et al., 2019) in the atmosphere and thus with the active/break cycles on the monsoon (e.g., Lucas et al., 2014). This strong and rapid variability in upper ocean conditions in the Bay of Bengal, and the potential feedbacks on the monsoon, motivated multiple observational research programmes with field campaigns in the Bay of Bengal, as discussed in the next section.

## 7. Multiscale upper ocean processes in the Bay of Bengal

Reflective of its name, the Bay of Bengal is in many ways analogous to a large-scale estuary with seasonally reversing winds and boundary currents that facilitate the transport, stirring, and mixing of water masses. To the north, the Ganga-Brahmaputra-Meghna watershed delivers on average 1300 km$^3$ in annual runoff of freshwater with a seasonal peak in discharge from July to September (Sengupta et al., 2006). During the southwest monsoon (boreal summer), the Summer Monsoon Current (Fig. 10) flows eastward advecting high salinity waters from the Arabian Sea into the southern Bay of

Bengal, balancing the Bay's net outflow of freshwater. Instabilities and eddies result in mesoscale stirring of these different water types and create a strongly filamented and complex near-surface thermohaline structure. Lateral and vertical gradients in stratification are further modified by submesoscale processes, instabilities, and mixing. The resultant shallow stratification allows for rapid coupling with the atmosphere. Collectively, these conditions present a natural laboratory to study multi-scale mixing processes and their link to air-sea interaction. This section discusses new understanding of physical processes in the Bay from the large-scale to sub-mesoscale and finally at the smallest mixing scales.

Recent focus on the Bay of Bengal's upper ocean structure has been prompted by the need to understand atmosphere and ocean coupling with the aim of ultimately informing monsoon forecasting efforts at the intraseasonal timescale and shorter. Several bi-lateral international collaborations (Lucas et al., 2014; Wijesekera et al., 2016; Mahadevan et al., 2016; Vinaychandran et al., 2018; Gordon et al., 2019, 2020) have collectively supported multiple field campaigns, beginning in 2013 and concluding in 2019, using a combination of shipboard, moored, and autonomous platforms. These atmospheric and oceanic measurements have provided new insights into the BoB's structure and the processes that regulate that structure, particularly at fine lateral scales (<5 km).

Results from these combined efforts span from large-scales, e.g., the quantification of coastal transport along the Sri Lankan coast (Lee et al., 2016) and the mesoscale stirring of freshwater (Sree Lekha et al., 2018), to intermediate scales, e.g., high-resolution (order 100 m) frontal surveys that hint at the roles of submesoscale (Ramachandran et al., 2018) and non-hydrostatic processes in setting stratification (Sarkar et al., 2016), to small-scales with direct measurements of microstructure yielding new insights into the BoB's mixing regimes (Jinadasa et al., 2016; Thakur et al., 2019; Cherian et al. 2020).

## 7.1 The Bay's Forcing and Upper Ocean Structure

At the largest scales, the Bay is forced by air-sea fluxes of buoyancy and momentum, which are strongly modulated by the monsoon and vice versa. Precipitation and multiple river systems, including the Ganga-Brahmaputra-Meghna system, contribute to freshwater input that creates a barrier layer in the surface Bay of Bengal, which is strongest in the northern Bay weakening toward the south. The Bay's stratification, in particular its barrier layer, is unique in how it impacts the evolution of seasonal SST, in turn setting the lower boundary condition for the development of the monsoon (Li et al., 2017). For this reason, recent emphasis has been placed on understanding processes that determine the Bay's upper ocean salinity and temperature structure.

The monsoon cycle of surface forcing plays a first-order role in controlling the Bay's upper ocean temperature structure. Direct flux measurements are a critical component in our ability to accurately capture/represent and predict the magnitude and variability of monsoon air-sea coupling. Recent studies have shown that of the air-sea heat flux terms, shortwave

radiation and latent heat flux are the largest drivers of variability to the total heat tendency. These variables are also those which reanalysis products struggle most to accurately represent, showing biases up to 75 W/m$^2$ (Sanchez-Franks et al., 2018). High-quality air-sea surface flux measurements over the BoB historically have been limited to the few sites maintained by the RAMA array (McPhaden et al., 2009). However, regional measurement efforts have expanded and baseline surface measurements are now collected and sustained through India's National Institute of Ocean Technology's met-ocean buoy program (Venkatesan et al., 2018), as well as the recent transition of an 18°N air-sea flux buoy from Woods Hole Oceanographic Institution to Indian National Centre for Ocean Information Services (Weller et al., 2016).

Precipitation and riverine discharge along the Bay's margins respectively contribute roughly 60% and 40% of the 0.14 Sv net freshwater delivered to the Bay (Sengupta et al. 2006; Wilson and Riser, 2016). Precipitation peaks in early summer (June) with a value near 0.4 m month$^{-1}$, while discharge peaks slightly later in summer (August) with a value near 0.3 m month$^{-1}$. Evaporative loss (included in the net 0.14 Sv) is relatively steady throughout the year at 0.1 m month$^{-1}$ (Wilson and Riser, 2016). Estimates of river discharge from gauged sources are known to have uncertainties (underestimates) related to unmonitored tributaries and streams. For large deltas, altimeter-based elevations offer a means of extrapolating gauge data over space and time. Papa et al. (2010, 2012) applied such an approach to the Ganga-Brahmaputra River system for the period 1998-2011. This time series allows for assessment of interannual variability over time ranges not spanned by gauged efforts. Papa et al. (2012) note a 12,500 m$^3$/s standard deviation in interannual variability in the Ganga-Brahmaputra discharge. Importantly, such data sets are also easily accessible by the general public, facilitating progress and understanding by the scientific community.

The Bay's upper ocean temperature and salinity structure is an integrated representation of the above summarized sources/sinks of heat and freshwater, combined with the physical processes that redistribute these quantities. The thermohaline structure of the Bay is remarkable in several regards—for shallow mixed layer depths (< 5 m, Sengupta and Ravichandran, 1998), for inversions of temperature (Shroyer et al., 2016, 2019; Thadathil et al. 2016), for large-scale coherent layering that spans 100 kms (Shroyer et al., 2019), an active mesoscale field and the strong influence of river discharge over the interior basin. The Bay's salinity stratification is a critical, if not dominant, contributor to the upper ocean density stratification. It supports the formation of barrier layers that are frequently observed to be warmer than the mixed layer thereby providing a substantial subsurface heat reservoir with the potential to modify air-sea interaction (Girishkumar et al., 2011; Shroyer et al., 2016). For example, in conditions supportive of formation of a diurnal warm layer (low winds, strong insolation), subsurface turbulent fluxes can act to modulate the diurnal SST cycle by transporting (typically) warm barrier layer waters into the mixed layer at night while still cooling the base of the diurnal warm layer (DWL) during the day (Shroyer et al., 2016). A similar phenomenon, albeit on a much different scale, results with passage of cyclones, which often show a salty wake even in the absence of a cool wake which is common for cyclones elsewhere (for e.g. Chaudhuri et al. 2019, Qiu et al. 2019). Below, we review recent progress on understanding of processes that determine the Bay's upper ocean thermohaline structure.

## 7.2 Lateral Processes

### 7.2.1 Stirring from the Margins

The Bay of Bengal has an active mesoscale eddy field that stirs diverse source waters into the interior of the Bay of Bengal. The origins of these source waters are the Arabian Sea waters to the west, the Ganga-Brahmaputra-Meghna at the northern tip, Andaman Sea waters to the east, and Equatorial waters to the south. This stirring effectively contributes to a quasi-stationary balance of the fresher waters from the north and the high salinity waters from the west and south over time. Lateral advection is a fundamental contributor to the formation of the barrier layer (George et al., 2019) and the freshwater budget of the Bay (e.g. Sree Lekha et al., 2018). In the northern Bay, the dispersal of water from the periphery into the interior depends critically on mesoscale stirring and the time varying Ekman transport, as indicated from mooring (Sree Lekha et al., 2018) and ship-based surveys (Shroyer et al., 2019), and constrained by modelling results (Sree Lekha et al., 2018). Here, the advection of freshwater by the mesoscale stirring also plays an important role in determining SST over the northern BoB (Buckley et al. 2020), as these waters are typically associated with relatively shallow mixed layers. In the southern Bay, measurements have suggested the competing influences of mixing and advection of salty Arabian Sea water in the erosion and reformation of the barrier layer during the southwest monsoon (George et al., 2019; Vinayachandran et al., 2018). In particular, George et al. (2019) show that maintenance of the barrier layer and the associated maximum depth of mixing was critically dependent on horizontal advection through its impact on stratification. Surface freshwater input also has an impact on barrier layer evolution; several freshening events were captured at various stages of their seasonal evolution in the southern Bay of Bengal in recent observations (Vinayachandran et al., 2018). These events play a significant role in the formation of a thick barrier layer, showing that during the southwest monsoon the shoaling of the mixed layer in the southern BoB has a similar magnitude and behaviour to that in the northern BoB (Vinaychandran et al., 2018).

### 7.2.2 Inter-basin exchange

Inter-basin exchange is critical to the Bay's salinity budget; since the Bay receives net freshwater input, this freshwater must be balanced by salty water imported from either the Arabian Sea or the western equatorial Indian Ocean (Jensen et al., 2001; Sanchez-Franks et al., 2019), and turbulent transport of salt into the fresh water layer is necessary to maintain the BoB's long-term salinity balance. Observations show that intrusion of high salinity water from the Arabian Sea enters the BoB between 80°-90°E during the southwest monsoon, (e.g. Murty et al, 1992; Vinayachandran et al., 2013) and has been found in several models (e.g. Vinayachandran et al., 1999; Han and McCreary, 2001 and Jensen, 2001). More recent observational and modeling studies show that both lateral and vertical transfer of heat and salt occur at multiple space-time scales. Seasonal currents play an important role in transporting heat and salt in and out of the BoB, but the role of mesoscale eddies on lateral transports is not well known.

Using unique year-long mixing measurements detailed in Section 7.3, Cherian et al. (2020) tentatively estimated a turbulent salt flux of 1.5e-6 psu ms$^{-1}$ out of Arabian Sea water averaged between 85°E and 88.5°E at 8°N through the 34.75 psu isohaline between August and January. Over those 6 months, this flux would increase the salinity of a 75m layer of water by 0.3 psu, though much of this would be cancelled out by surface fluxes. The magnitude and timing of this salt flux roughly match that necessary to restore the Bay's near-surface salinity after the large freshwater input in August as estimated by a few modelling studies (Akhil et al., 2014; Benshila et al., 2014; Wilson and Riser, 2016). This is the first direct measurement of turbulence that supports the hypothesis of intrusion of high salinity water from the Arabian Sea during the southwest monsoon (Vinayachandran et al., 2013).

### 7.2.2.1 Andaman Sea Exchange

The Irrawady river drains into the Andaman Sea, a marginal sea at the eastern edge of the Bay. Export from the Andaman is then another source of freshwater for the Bay, particularly at intermediate densities (22-25 kg m$^{-3}$). A striking example of the interaction between strong surface forcing and an anticyclonic eddy can be found in the fortuitous crossing of an intrathermocline eddy (ITE) in 2013 as reported by Gordon *et al.* (2017). The water mass characteristics clearly identify ITE waters from the Andaman Sea; and, analysis of ancillary Argo data suggest a similar water type often penetrates westward into the Bay extending from the three passages connecting the two basins . While at the time of transit the observed ITE had a very weak surface expression, a week prior to encountering the ITE a clear sea surface high (>10 cm) is evident in AVISO SSHA. Tropical cyclone Lehar passed near the location of this sea surface high in the interim, and the working conjecture is that the winds associated with Lehar were sufficient to modify a typical mode-1 anticyclone into the observed ITE.

### 7.2.2.2 Arabian Sea Exchange

Near-surface exchange from the Arabian Sea into the Bay of Bengal is influenced by the Sri Lanka Dome (SLD), an upwelling thermal dome that recurs seasonally within the SMC in the wind shadow of Sri Lanka (Vinayachandran and Yamagata 1998, de Vos et al. 2014, Burns et al. 2017). The SLD has long been recognized as a prominent circulation feature in the southwestern bay during the summer monsoon; and it has been noted as a region of enhanced productivity (Vinayachandran et al., 2004, de Vos et al. 2014), cool SST (Burns et al. 2017), and consequently depressed convection (Figure 15). The SLD displays pronounced interannual variability (Cullen and Shroyer 2019). In some years the SLD has a strong surface manifestation (amplitude of the low ~30 cm) that persists well beyond the southwest monsoon; in other years the SLD has a weak expression that is intermittent and short-lived (~1-2 months). The SLD is not fixed in location despite its strong association with the wind stress curl. Its position varies from year-to-year as well as over the course of one season. Variations in its location and strength may influence the properties of waters entrained and upwelled within the SLD.

At intermediate depths (<~200 m), the signature of the neighboring Arabian Sea is notable across much of the basin
(Gordon et al., 2016). During summer, Arabian Sea High Salinity Water (ASHSW; density near 22-24 kg m$^{-3}$) is
carried/advected into the Bay of Bengal as a 'high salinity core' via the Southwest Monsoon Current (SMC, Webber et
al., 2018; Sanchez-Franks et al., 2019) and then spread north along the bay's central spine (Hormann et al., 2019). During
this journey, salt is mixed upward into the near-surface fresh layer (Cherian et al 2020; Section 7.3). A nearly two-year
long moored current record in the southern BoB captured seasonally varying large eddies generated by the SMC and
Northeast Monsoon Current. These eddies included a cyclonic eddy, the SLD, and an anticyclonic eddy south of the SLD
(Wijesekera et al. 2016c). These observations revealed that the average transport over a nearly two year period into the
BoB was about 2 Sv (1 Sv = 10$^6$ m$^3$ s$^{-1}$) but likely exceeded 15 Sv during summer of 2014, which is consistent with the
transport associated with the SMC (e.g., Schott et al. 2009; Webber et al. 2018). The observations further indicate that the
water exchange away from coastal boundaries, in the interior of the BoB, may be largely influenced by the location and
strength of the two eddies that modify the path of the SMC.  The strength and location of the SMC itself is dependent on
a combination of local and remote forcing (Webber et al., 2018).
As discussed above several hypotheses have been suggested for cyclonic eddy (SLD) and anticyclonic eddy formation in
the southern BoB. It has been suggested that the cyclonic wind stress-curl over southwestern BoB generates the SLD
(McCreary at al., 1996; Vinayachandran and Yamagata 1998; Schott et al., 2001; Cullen and Shroyer 2019). Based on
numerical simulations, de Vos et al., (2014) argued that the separation of SMC from the (southern) boundary of Sri Lanka
may lead to SLD, where a cyclonic vorticity is generated by lateral frictional effects. A mechanism for the anticyclonic
eddy formation has been proposed by Vinayachandran and Yamagata (1998), where the interaction of the SMC with
Rossby waves arriving from the eastern boundary leads to the anticyclonic eddy. Pirro et al (2020a) proposed a new
hypothesis wherein the anticyclonic eddy is generated by a topographically trapped Rossby wave response of the SMC to
perturbations by the Sri Lankan coast. They reported that observations of the size, location and origins of the SLD were
broadly consistent with their hypothesis, based on a laboratory experiment designed to mimic natural flow in the BoB by
creating an eastward jet (SMC) on a simulated $\beta$ plane.
High-resolution sampling of the interior BoB has provided a more detailed look at the lateral extent of typical 'patches' of
Arabian Sea water, which tend to remain well-defined over scales of 10-50 km, suggesting the importance of eddy activity
in exchange (Shroyer et al., 2019). While many studies have traced origins of ASHSW from the eastern Arabian Sea,
entering the Bay of Bengal directly via the southwest monsoon current (e.g., Jensen et al, 2016);  a recent study suggests
an equatorial pathway may also be relevant (Sanchez-Franks et al., 2019; Section 7.2.3).  Highly salty and highly
oxygenated waters from the Persian Gulf and the Red Sea have also been noted in the southern regions of the Bay of
Bengal (Jain et al., 2017).  These waters are injected into the Bay of Bengal via current systems (equatorial and the
southwest monsoon current) with important repercussions for the oxygen concentrations of the Bay of Bengal oxygen
minimum zone (Sheehan et al., 2020).
Velocity and hydrographic profiles from a shipboard survey in December 2013 combined with drifter observations,
satellite altimetry, global ocean nowcast/forecast products, and coupled model simulations were used to examine the
circulation in the southern Bay of Bengal during the Northeast monsoon (Wijesekera et al. 2015). The observations
captured the southward flowing East India Coastal Current (EICC, e.g., Shetye et al. 1994) off southeast India and east of
Sri Lanka. The EICC was approximately 100 km wide, with speeds exceeding 1 m s$^{-1}$ in the upper 75 m. East of the EICC,
a subsurface-intensified 300-km-wide, northward current was observed, with maximum speeds as high as 1 m s$^{-1}$ between
50 m and 75 m. The EICC transported low-salinity water out of the bay and the subsurface northward flow carried high-
salinity water into the bay during typical northeast monsoon conditions (Wijesekera et al. 2015; Jensen et al. 2016).
**7.2.3 Equatorial Connections**
The Equatorial undercurrent (EUC) in the Indian ocean is seasonally variable. The summer–fall EUC tends to occur in the
western basin in most years but exhibits evident interannual variability in the eastern basin (Chen et al. 2015), with
different processes dominating its generation in the western and eastern basins. In the eastern basin reflected Rossby waves
from the eastern boundary play a crucial role in the EUC, whereas directly forced Kelvin and Rossby waves control the
EUC in the western basin.
Equatorial Kelvin waves, commonly interpreted as Wyrtki (1973) jets, propagate eastward along the equator during
April/May and September/October. Upon reflection from the IO eastern boundaries, energy of Wyrtki jets is reflected
back in part as long Rossby waves that disperse slowly during the following two months and reach the central-eastern BoB
during July-August (Han et al., 1999, 2001; Han, 2005; Nagura and McPhaden, 2010a). The remaining energy is
partitioned into two coastally-trapped Kelvin waves traveling poleward (Moore, 1968), which excite long Rossby waves
propagating westward. Therefore it is suggested that planetary waves driven by remote forcing from the interior IO
contribute significantly to the formation, strength and intensity of the BoB circulation (Vinayachandran et al. 1998; Nagura
and McPhaden, 2010b; Chen, 2015).  A subset of these planetary waves are the mainstay of intraseasonal oscillations
(ISOs), a sub-seasonal phenomenon of period less than 120 days. The genesis of oceanic ISOs has been attributed to
multiple mechanisms: external forcing (e.g., atmospheric ISOs and Ekman pumping, e.g. Duncan and Han 2012) and
internal processes (upper ocean processes and instabilities e.g. Zhang et al. 2018).
Observations in the IO have captured a range of variabilities in the 30 – 120 days frequency band (e.g., Girishkumar et al.,
2013), and past research has identified roughly three distinct ISO bands in the context of the thermocline: 30-60 days, 60-
90 days, and 120 days (Han et al., 2001; Girishkumar et al., 2013). Pirro et al. (2020b) discussed interaction between 30-
60 day ISOs and the SMC in the southern BoB using long-term moored observations. They estimated that the background
mean flow acceleration resulting from the meridional divergence of wave momentum flux in the thermocline was about
$10^{-8}$ m s$^{-2}$. As a result, within a wave period, ISOs can enhance the eastward flow in the thermocline by about 25%. The
negative shear production computed for the same period is consistent with this finding suggesting that the mean flow
gained kinetic energy at the expense of the ISO band. The meridional heat-flux divergence was -10$^{-7}$ °C s$^{-1}$ and has a
tendency for cooling the thermocline by about 0.5°C when ISOs are active (Pirro et al., 2020b). Observations have also
captured energetic and consequential 5-20 day convectively coupled Kelvin waves in the atmosphere (Baranowski et al,
2016) that generate oceanic Kelvin waves, affect surface heat fluxes and generate upper ocean turbulence (Pujiana and
McPhaden, 2018).
High salinity waters from the western Arabian Sea and the western Equatorial Indian Ocean can route to the Bay of Bengal
via the Somali Current and the Indian Ocean EUC (Sanchez-Franks et al., 2019). Changes in strength of the Bay of Bengal
high salinity core are linked to the convergence of the East Africa Coastal Current and the wintertime southward-flowing
Somali Current, with anomalously strong equatorial Undercurrent (Fig. 19). Because of the seasonal reversal of currents,
two junctions form naturally, one in the western equatorial Indian Ocean (Somali Current) and another south of India
(monsoon currents), which effectively act as 'railroad switches' rerouting water masses to different basins in the Indian
Ocean depending on the season (Fig. 19, Sanchez-Franks et al., 2019).

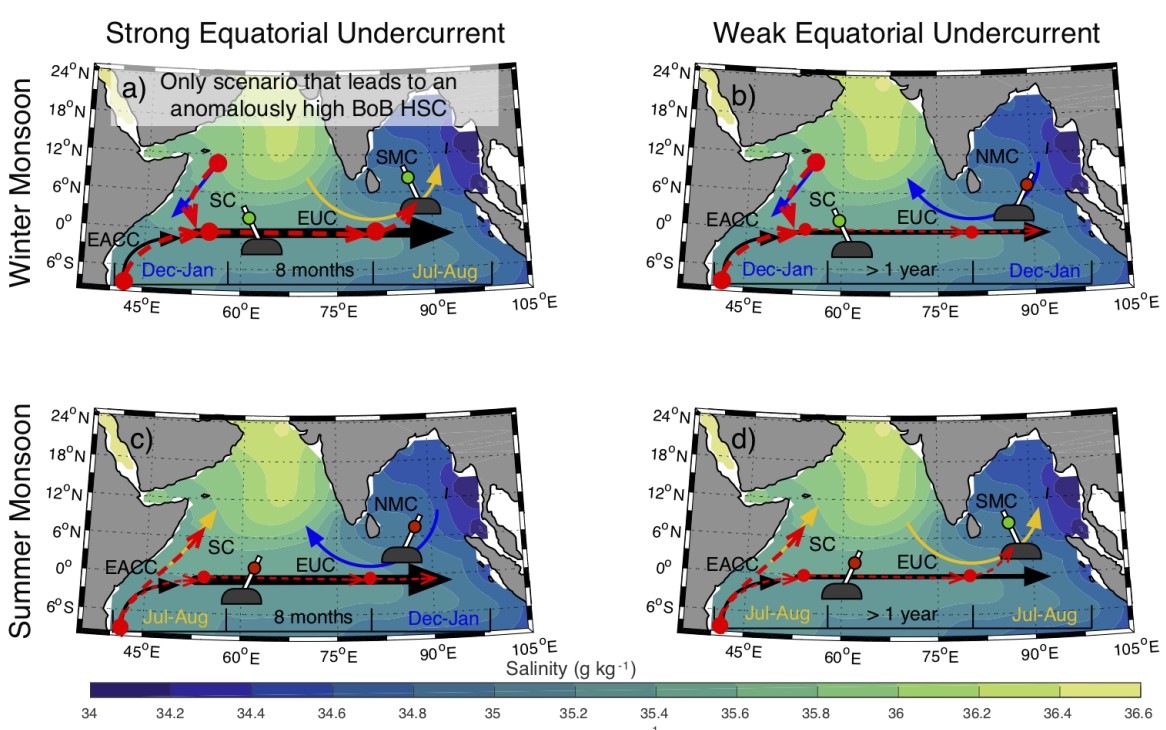


**Figure 19: Seasonal circulation pathways in the northern Indian Ocean, or Railroad Switch schematic, on**
**subsurface (90 m) salinity climatology (psu; shaded) from the Argo optimally interpolated product for the four**
**Equatorial Undercurrent scenarios: (a, b) winter monsoon and strong (weak) Equatorial Undercurrent and (c, d)**

 **summer monsoon and strong (weak) Equatorial Undercurrent. Red dashed arrows indicate high-salinity advection.**

**BoB = Bay of Bengal; HSC = high-salinity core; SMC = Southwest Monsoon Current; SC = Somali Current; EUC**
**= Equatorial Undercurrent; EACC = East African Coastal Current. From Sanchez-Franks et al. (2019).**
**7.3 Vertical Mixing**
Strong stratification in the Bay of Bengal plays a critical role in setting the upper ocean turbulence, notably leading to
relatively weak mixing compared to other regions (e.g. Gregg et al., 2006). However, large-scale inferences suggest that
mixing must play a key role in at least two regards. First, the net surface flux during the southwest monsoon on average is
warming but yet the SST cools (Shenoi et al, 2002). Second, the large-scale salt balance must be closed through upward
mixing of high-salinity water carried into the Bay via the Summer Monsoon Current (Vinayachandran et al., 2013).
Recent year-long direct measurements of mixing in the Bay have helped link the seasonal cycle in mixing to the seasonal
cycle of winds, currents and freshwater. These year-long measurements were recorded by mixing meters called χpods.
χpods consist of two temperature microstructure sensors and a suite of ancillary sensors necessary to infer the rate of
dissipation of temperature variance at 1Hz frequency for up to a year (Moum & Nash, 2009). χpods have been deployed
on moorings in three different regions of the Bay (Figure 20): the air-sea buoy at 18°N, top 65m (Thakur et al., 2019),
RAMA moorings along 90°E (mixing measurements at 15m, 30m and 45m; Warner et al. 2016), and the EBoB array in
the south-central Bay (mixing measurements spanning between 30m and 100m at sites in the region 85°E-88°E, 5°N-8°N,
Cherian et al., 2020). Across the basin, turbulence within and near the base of the mixed layer shows strong seasonality
that parallels the monsoon cycle in winds (Thakur et al., 2019, Warner et al., 2016). In the thermocline of the south-central
Bay (EBoB array), mixing is correlated with packets of downward propagating near-inertial waves implicating wind
forcing. As depicted in Figure 20, both near-surface and thermocline mixing are relatively high during the NE and SW
monsoons (Dec-Feb, May-Sep) and relatively low during the transition (Mar, Apr). Cyclones during the post-monsoon
months of October and November can drive a hundredfold increase in near-surface mixing both locally and throughout
the Bay (Warner et al. 2016). Turbulence profiles collected by a fast thermistor on a CTD rosette during a basin-wide
survey before and after the passage of cyclone Madi (6-12 Dec, 2013) show a basin-wide increase in diffusivity linked to
near-inertial waves forced by the cyclone (Wijesekera et al., 2016b).
Indirect estimates of turbulent diffusivity and turbulent heat fluxes at the base of the mixed layer can be found as the
residual of a mixed layer heat budget whose terms are estimated using a combination of mooring and satellite
measurements. Girishkumar et al. (2020) use this approach to indirectly estimate seasonal median turbulent diffusivities
using decade-long RAMA mooring records at 90°E. They find a robust seasonal cycle of mixing at 8°N, 12°N, and 15°N;
and strong latitudinal variability in turbulence, with larger diffusivities inferred at 8°N relative to 12°N and 15°N in all

seasons. When comparisons are possible, the indirect estimates compare well against the more direct but time-limited

estimates of Warner et al (2016) at 90°E, 12°N.

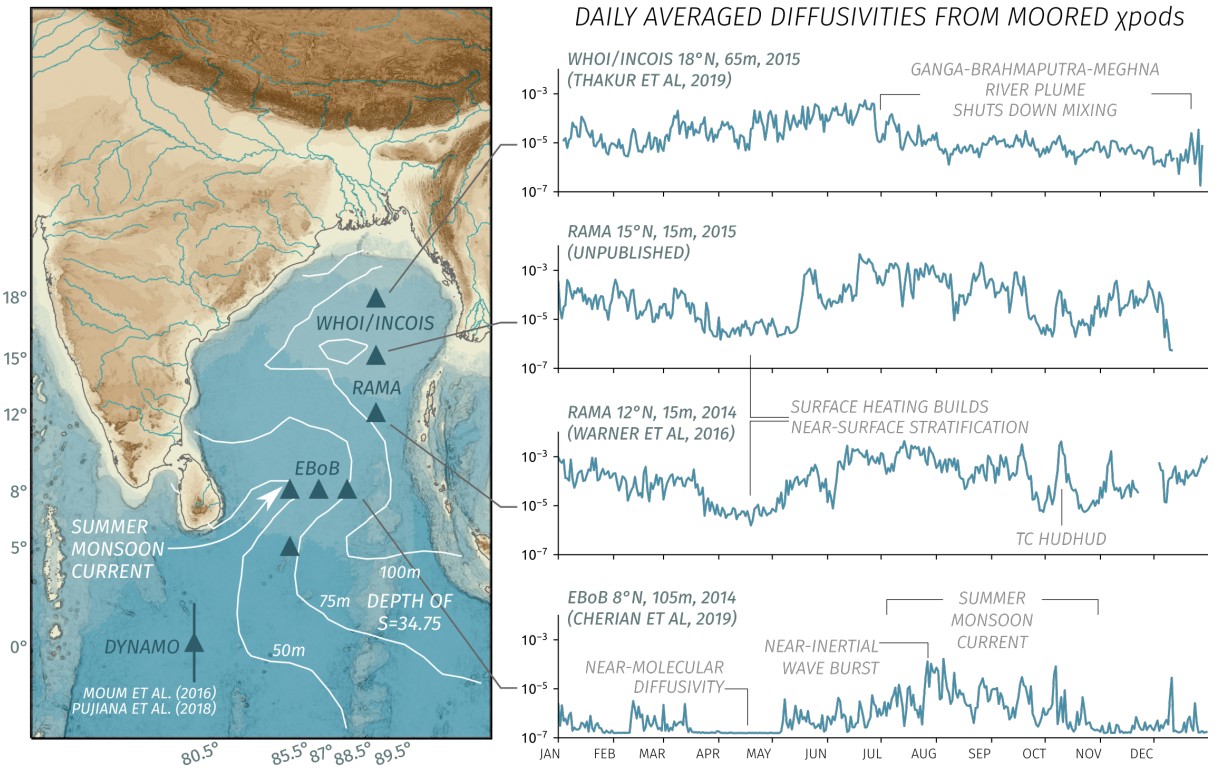

**Figure 20: Annual cycle of daily averaged temperature diffusivities derived from χpod measurements. The data are from two different years, 2014 and 2015, depending on location. Note the similar wind-forced seasonal cycle at 12°N, 15m and 15°N, 15m and the dramatically different seasonal cycle at 8°N, 105m (reflecting near-inertial wave activity) and at 18°N, 65m reflecting freshwater influence.**

The influence of freshwater is a critical caveat to the above generalizations: the arrival in August of the Ganga-Brahmaputra-Meghna freshwater plume at 18°N has been observed to suppress turbulence (diffusivity $K_T < 10^{-5}$ m² s$^{-1}$) for multiple months (Aug-Nov) at depths of approximately 50-65 m (Figure 20). This buoyant lens limited the vertical extent of the influence of Tropical Cyclone Komen as compared to a previous (weaker) storm (Chaudhuri et al 2019, Thakur et al 2019). Similar observations of extremely weak turbulence below strong, salinity-stratified surface layers have been reported throughout the Bay using data from a variety of platforms: ship-based microstructure (Jinadasa et al, 2016) profiling floats with a temperature microstructure sensor (Shroyer et al, 2016) and glider-based microstructure measurements (St. Laurent and Merrifield, 2017). Lucas et al (2016) find that near-inertial shear was elevated at the base of the mixed layer but not elevated at the base of the barrier layer — direct evidence that salinity stratification can insulate

deeper depths from the effects of near-surface forcing (downward propagating near-inertial waves in this case). Li et al. (2017) use a combination of observations and modelling results to demonstrate that barrier layers in the Bay of Bengal influence the amplitude of intraseasonal oscillations in SST and precipitation. However, a recent coarse resolution coupled modelling study suggests that freshwater has little influence on SST or rainfall, since the SST tendency caused by a reduction in mixing is offset by changes in surface heat fluxes (Krishnamohan et al., 2019)

Surface freshwater advection can create subsurface reservoirs of heat and salt that can be accessed when the winds are strong enough, such as during cyclones that regularly form in the Bay during October and November. In one dramatic example Qiu et al (2019) report up to 5 psu increases in SSS and only a smaller 0.5°C decrease in SST following the passage of Cyclone Phailin (2013). In this case, mooring records indicate that mixing was limited to the isothermal layer (Chaudhuri et al. 2019). Subsurface warm layers (i.e. temperature inversions stabilized by strong salinity stratification) are also observed, representing a reservoir of heat that can be accessed if a storm excites enough turbulence, as appears to have happened during the passage of Cyclone Hudhud (Warner et al, 2016). The influence of stratification in limiting the extent of vertical mixing and creating subsurface warm layers mean that cyclone-induced cooling is generally either weak or negligible in the Bay, unlike in other ocean basins (Sengupta et al, 2008). Subsurface warm layers influence SST on longer timescales too: Girishkumar et al (2013) find that the wintertime SST at 8°N, 90°E is quite sensitive to the thickness of the barrier layer, and to the presence of temperature inversions (subsurface warm layers) in the barrier layer on intraseasonal and interannual timescales.

Long periods of near-molecular diffusivities (weeks to a month) were also inferred at multiple χpods along 8°N between 50 m and 100 m during transition months of March and April. Here freshwater insulation does not appear to be the major factor. Instead the period of weak turbulence may be linked to low levels of near-inertial energy (a consequence of weak wind forcing in March and April) and the absence of strong mean oceanic flows during these transition months (Cherian et al 2020). Relatively weak diffusivities are also present in the LADCP fine structure estimate of depth-integrated (thermocline to bottom) turbulent kinetic energy dissipation ε (Kunze et al, 2006) and the Argo fine structure-based 250-500 m diffusivity estimates of Whalen et al. (2012). The extended presence of such weak turbulence suggests that the Bay's internal wave field is weaker than might be expected from the Garrett-Munk internal wave spectrum at least during some months of the year. Another (related) question is the issue of representation of such weak background mixing in climate models and whether that matters to known biases in such models.

Published efforts so far have been directed towards understanding the modulation of turbulence by larger-scale variations in the wind, currents and freshwater. Questions remain as to the impact of small-scale mixing on the large-scale long-term T-S structure in the Bay as well as the influence of subsurface mixing and the ensuing modification of SST on coupled ocean-atmosphere phenomena such as the MJO and the MISO (Section 3.2)

**7.4 Where vertical and lateral processes meet: The Role of Submesoscale**

Freshwater inflow from the Ganga-Brahmaputra-Meghna (GBM) and the Irrawady river in the Bay of Bengal is stirred by the mesoscale eddies into sharp frontal gradients (in salinity and in density) at O(1-10km) scales with shallow vertical extent. These fronts are acted upon by winds seasonally, setting up complex sub-mesoscale structures with salinity differences O(1 psu) over 1-10 km, developing bore-like features with O(0.5 psu) difference over a few meters horizontally (Nash et al 2016; Figure 21). Wavenumber spectra of temperature at O(1-10km) scale show a −2 slope in many regions of the Bay (Mackinnon et al 2016), a signature of frontogenesis in the Bay at these scales. The BoB is thus replete with fronts which evidently slump at sub-mesoscales due to both symmetric and baroclinic instabilities (Ramachandran et al. 2018), and show higher stratification near fronts (Sree Lekha 2019).

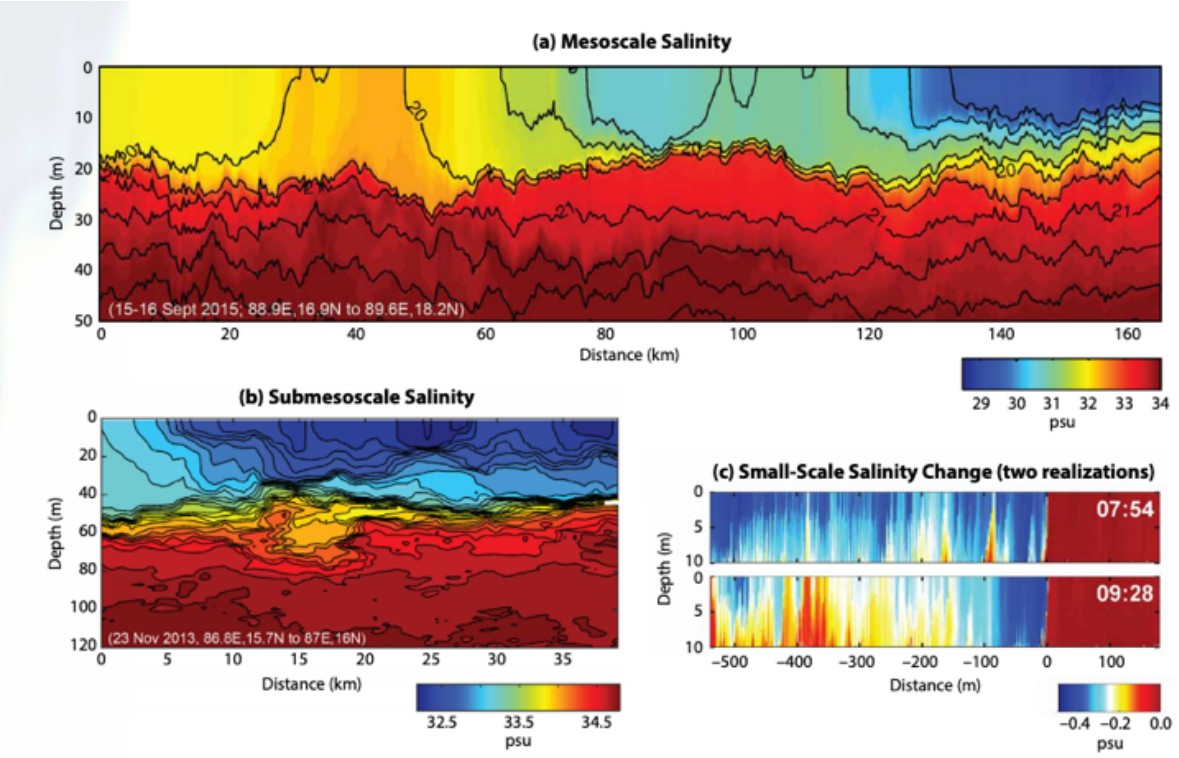

**Figure 21: Observed salinity gradients at mesoscale, sub-mesoscales and small horizontal scales from in the Bay of Bengal (Nash et al. 2016).**

The fronts and filaments at O(1-10km), which are dominated by salinity gradients and weakly compensated, have strong implications for setting up the density stratification in the top 50-100m in the BoB (Section 4.4.1). The stratification in this depth range often has multi-layered structure with stratification varying at O(1-10km) scales (Lucas et al 2016), showing evidence that the stratification in the Bay cannot be explained simply in terms of vertical processes, and horizontal

submesoscale processes are intimately coupled with the vertical processes at these scales. Ramachandran et al. (2018)
show that a mesoscale strained region with strong fronts (O($1kg/m^3$ over 40km)) and weak down front wind shows multiple
dynamical signatures of sub-mesoscale instabilities. Ageostrophic secondary circulations arising near the fronts and the
accompanied sheared advection plays an important role in setting the stratification (Pham and Sarkar 2019). Both
observations and process modeling show O(1-10km) patches of low potential vorticity consisting of subducted warm water
patches due to a combination of baroclinic and forced symmetric instabilities, creating barrier layers whose thickness
varies laterally at sub-mesoscales (Ramachandran and Tandon, 2020 JGR-in review).
During winter, the temperature gradients in the horizontal compensate for the salinity gradients to reduce the density
gradient, and the sub-mesoscale processes in BoB lead to a unique situation. Jaeger & Mahadevan (2018) show that surface
cooling fluxes combined with submesoscale instabilities of the haline fronts during wintertime leads to shallower mixed
layers on the less saline (cooler) side. Therefore, cold SSTs in wintertime in the Bay mark surface trapped waters (Fig.
22), whereas in other regions of the world ocean, cold filaments mark upwelling of nutrient-rich waters. Further, since the
shallow fresher mixed layers lead to larger drops in temperature, this develops the correlation between SST and SSS at
O(1-10km) scales.
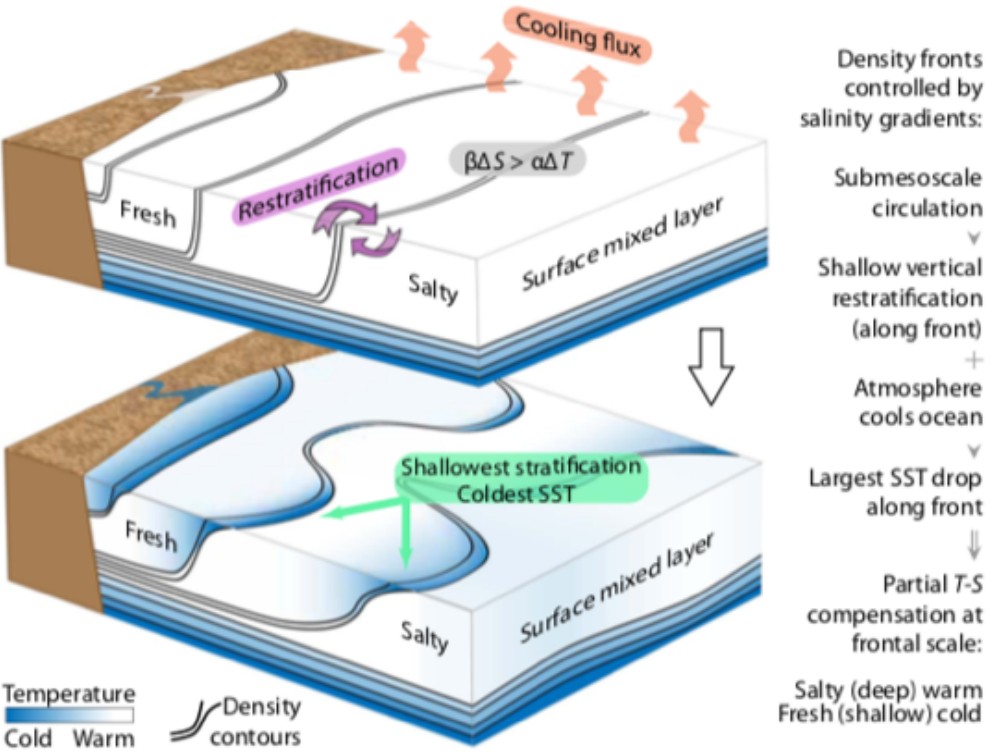

**Figure 22: Interaction of submesoscale salinity gradients with atmospheric cooling leads to shallow cold regions (From Jaeger and Mahadevan, Science Advances 2018)**

## 7.5 Putting the Pieces Together

### 7.5.1 Coupled ocean-atmosphere phenomena

Due to the presence of a barrier layer over much of the Bay of Bengal, entrainment and upwelling of waters from the thermocline are inhibited, and the evolution of SST is largely driven by net air-sea heat flux variability (Duncan and Han, 2009). However, the dependency of SST on surface fluxes is controlled by subsurface processes such as formation of barrier layers, entrainment warming and cooling of the mixed layer, penetrative solar radiation and zonal advection (Thangaprakash et al., 2016). Advection is important in influencing the SST as lateral variations in the mixed layer depth alone can result in variations in air-sea fluxes of roughly 20 $Wm^{-2}$ over distances of kilometers (Adams et al., 2019). This magnitude is similar to uncertainty in air-sea flux products (Weller et al. 2016) thus implying that variations in sub-mesoscales are important for heat balance in the northern BoB. The coupling of the ocean-atmosphere over BoB at large scales implicates the air-sea interaction and the mixed layer heat budget in the BoB (Rahaman et al. 2019), although at oceanic mesoscale and finer scales in the horizontal and at sub-seasonal timescales this coupling is a topic of active research.

### 7.5.2 Implications for biogeochemistry in the Bay

Eddies in the central BoB arise not by the baroclinic instability of boundary currents but rather due to planetary wave dynamics off the equator that triggers coastal Kelvin waves around the Bay. The Kelvin waves then trigger south-westward propagating Rossby waves, which result in large mesoscale structures in the Bay (Cheng et al. 2018). The Andaman and Nicobar Islands are also shown to be very important for the generation of these eddies; without these islands the number of eddies would have reduced to almost half in the western bay of Bengal (Mukherjee et al., 2019). These eddies provide much of the horizontal stretching and stirring of the tracers, including those relevant to the ecosystems

Eddies have tremendous potential to influence ocean biogeochemistry by providing "new" nutrients to the ocean's euphotic layer (Stramma et al., 2013). However, we do not fully understand the spatial distribution of nutrients within the eddy surface area – e.g., there is a debate whether nutrients upwell at the core and downwell at the edge of the eddy, or vice versa. Further, such discrepancy also continues in the type of eddies – i.e., whether upwelling occurs in cyclonic and downwelling occurs in anticyclonic eddies and vice versa (Mahadevan, 2014; Mahadevan et al., 2012; Martin and Richards, 2001). But there is a consensus that eddies do impact biogeochemistry (McGillicuddy et al., 2007).

There have been only a handful of studies on the role of eddies in biological productivity in this region (Kumar et al., 2007; Singh et al., 2015). Kumar et al. (2007) observed an increase in surface nutrients in the Bay through eddies during both fall-2002 and spring-2003 followed by higher biomass. Despite being highly eutrophic, biological activity did not increase following cyclonic eddies during the summer-2003 in the northern Bay (Muraleedharan et al., 2007). But primary production switched from 'regenerated' to 'new' production during summer-2003. In a [15]N based new production estimate to assess the role of cyclonic eddies in enhancing primary production, Singh et al. (2015) carried out measurements of primary production at four stations in the Bay of Bengal (around a cyclonic eddy close to 17.8°N, 87.5°E) during winter 2007. The measurements sampled one cyclonic eddy during the campaign. The highest surface productivity (2.71 μM C d$^{-1}$) and chlorophyll a (0.18 μg L$^{-1}$) were observed within the eddy due to intrusion of nutrients from subsurface waters. Given new nitrogen input via vertical mixing, river discharge or aerosol deposition, the additional primary production due to this new nutrient input and its contribution to the total production increased from 40% to 70%. Eddies could be a reason for the otherwise unexplained high new production rates in the Bay of Bengal (Singh and Ramesh, 2015). Eddies also seem to have a potential for transferring a high fraction of fixed carbon to the deep. A couple of recent studies have highlighted the role of mesoscale eddies in changing the elemental proportions of carbon:nitrogen:phosphorus in the organic and nutrient pools in the euphotic layer of the Bay (Sahoo et al., 2020, 2021).

## 8. Summary and open questions

This paper summarises a suite of new studies in the Indian Ocean that have been made possible through national, bilateral, and international programmes, including the IIOE-2. An increase in high quality observations (both increased spatial resolution and the acquisition of longer time series) has led to a substantial increase in our understanding of processes and interactions. These in-situ observations, in combination with remote sensing, detailed syntheses and modeling have increased our knowledge of the surface circulation and its complex implications for biological production, along with an increased understanding of air-sea interaction in the Indian ocean.

There are, however, a number of outstanding questions that require prioritised efforts. Compared to the Atlantic and Pacific, where the important boundary currents are now being monitored with a suite of gliders with repeated and sustained sections (Todd et al. 2019), the boundary currents and their variability in the Indian Ocean remain poorly constrained. Given the anomalous warming of the Indian Ocean, the frequency of heatwaves, and the population supported by the Indian Ocean and Monsoons, the air-sea fluxes and the coupled atmosphere-ocean exchange in this ocean remain poorly understood at many scales. Understanding of the intermediate, deep and abyssal layer circulation and the vertical overturning cells that connect these layers in the Indian Ocean is lacking.

There are still many gaps in current understanding of Indian Ocean biogeochemical cycles, which we have presented here in the context of the physical processes that affect them. Although the characterization of the temporal and spatial variability in chlorophyll concentration and primary production has greatly improved as a result of recent in situ measurements and satellite remote sensing, there are still many areas where there is little or no information about how this relates to changes in planktonic food web structure and particulate organic matter export to the deep ocean. Although nutrient limitation patterns were not discussed in this review, it should be pointed out that the importance of nitrogen verses iron and silica limitation in the Arabian Sea and elsewhere in the Indian Ocean is still a subject of debate - more nutrient and trace metal measurements are needed along with nutrient limitation bioassays throughout the Indian Ocean.

The number of nitrogen fixation rate measurements in the Indian Ocean has increased significantly over the last decade, but the importance of this process as a source of new nitrogen to the surface ocean has been quantified in only a few regions (e.g., off northwest Australia) and its contribution to bloom formation (e.g., the Madagascar Bloom) is still uncertain. From a spatial standpoint, the quantification of biogeochemical variability in the northern Indian Ocean (Arabian Sea and Bay of Bengal) has benefited, in particular, from numerous shipboard measurements, moorings and biogeochemical Argo float deployments in the last decade. Many questions still remain, for example, related to the influence of freshwater inputs on biogeochemical cycles in the Bay of Bengal. Remarkably, the biogeochemical and ecological impacts of the Indonesian Throughflow have been examined in only a handful of studies. Similarly, there are very few studies that focus on the biogeochemical and ecological impacts of the Seychelles-Chagos Thermocline Ridge (SCTR). The ITF and the SCTR are unique features of the Indian Ocean, yet the understanding of their biogeochemical and ecological impacts is rudimentary at best. Finally, the quantification of biogeochemical variability in the Leeuwin and Agulhas Currents and adjacent waters has also benefited from recent measurements, though it is important to point out that the biogeochemical impacts of boundary currents in the Indian Ocean are still poorly understood compared to the Atlantic and Pacific.

There are still large uncertainties in air-sea fluxes. Even in the regional basin of the Bay of Bengal where there have been focused international efforts, the river discharge and rain need to be better represented in models, as do the processes that set the shallow salinity stratification. These have important feedbacks on the SST which impacts atmospheric convection with a global reach. At longer time scales, the salinity feedbacks to climate at interannual to decadal timescales need to be investigated in further detail. The decadal variability of the Indian Ocean Dipole and its link to the Pacific decadal variability also needs to be better understood, particularly given events like the record breaking 2019 positive IOD that developed independently from ENSO conditions. Marine heatwaves are an increasing threat to marine ecosystems fuelled by increasing mean temperatures in the ocean and atmosphere. There are still large gaps in our understanding of the Indian Ocean dynamics that lead to these extremes, and consequently in our ability to predict the onset, intensity and frequency of extreme weather such as rainfall, flooding and heatwaves, associated with anomalously strong climatic mode events that have major socioeconomic impacts.

Modeling and observational efforts have both pointed to the increased role of air-sea coupling at higher frequencies to
improve the predictions of sub-seasonal Monsoon forecasts. Observations and models indicate that MISOs may be slowing
down because of the warming in the Indian Ocean (e.g. Sabeerali et al. 2013), which needs to be understood better for
providing reliable monsoon predictions and projections in this climate vulnerable region.
On the influence of small-scale mixing, increased measurements of ocean mixing both along the equator and new long-
term measurements in the Bay of Bengal, have shown intensively enhanced mixing during the passage of eddies and during
cyclones. However, there are still significant uncertainties in subsurface ocean mixing in setting the large-scale balance in
the Indian ocean.
It has been proposed that the hiatus in warming of the surface atmosphere may have ceased as the Pacific Ocean enters an
El Nino like state (Cha et al. 2018). However, the secular trends in the Pacific Ocean trade winds are expected to continue
to affect the Indo-Pacific Ocean heat content through the Indonesian Throughflow (Maher et al. 2018). The Indian Ocean
thus remains a critical component of the Earth's global response to the continued anthropogenic forcing and the ocean's
role as a clearing house for distributing heat to modulate global warming.
**Code Availability**
No original data analyses were undertaken as part of this review paper.
**Data Availability**
No original data analyses were undertaken as part of this review paper. All data presented in this manuscript have been
previously published and are available from sources identified in the original manuscripts.
**Author Contributions**
HEP and AT designed the review, wrote the introductory and concluding parts and sections in their areas of expertise. HP
and AT reviewed the contributions of the authors and made editorial adjustments. RH wrote the sections on
biogeochemical variability in Section 4. All co-authors contributed to the writing of sections relevant to their areas of
expertise and response to reviewer questions. All authors contributed to refining the manuscript for submission. RF, CU,
JB, BW, AS-F, JH and RM contributed editorial advice.

## Competing interests

The authors declare that they have no conflict of interest.

## Acknowledgements

The authors acknowledge the sustained efforts of researchers and funding agencies in observing and modelling the oceanic and atmospheric processes that control climate variability in the Indian Ocean region. These contributions during the International Indian Ocean Expeditions (I and II) and in the intervening years through national and international programs, such as CLIVAR and GOOS, are fundamental to improving our knowledge of these systems and increasing our skill at forecasting variability and extreme events. We thank the IIOE-2 leadership team (https://iioe-2.incois.gov.in/) for their unwavering efforts to share new discoveries and promote understanding of the importance of the Indian Ocean to the climate system and Earth's inhabitants. We are very grateful to Michael McPhaden, Lisa Beal and an anonymous reviewer for their encouraging and constructive comments that have led to a more comprehensive and balanced synthesis of recent advances. HEP acknowledges support from the Earth Systems and Climate Change Hub and Climate Systems Science Hub of the Australian Government's National Environmental Science Programme and the ARC Centre of Excellence for Climate Extremes. AT acknowledges the US Office of Naval Research. This is INCOIS contribution number 437.

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
