# Peer review of "Progress in understanding of Indian Ocean circulation, variability, air-sea exchange and impacts on"

_Ocean Science, 2021_

## Referee Comment (RC1)

Review of "Progress in understanding of Indian Ocean circulation, variability, air-sea exchange and impacts on biogeochemistry" by Phillips et al.

This is an ambitious effort to summarize scientific advances since the last comprehensive reviews of Indian Ocean dynamics and their role in the climate system by Schott and McCreary (2001) and Schott et al (2009). The paper contains an impressive amount of information compiled by an impressive list of authors. The writing is generally clear and the range of topics comprehensive. I consider myself an Indian Ocean expert and still learned a lot by reading it.

I list specific comments for the authors to consider below. Some are related to missing ideas, references that should be cited, or organizational issues. The biggest concern I have though is the length of the paper and the effort to include a comprehensive review of ocean dynamics with biogeochemistry. I appreciate that the authors are attempting an interdisciplinary synthesis, but in my opinion it doesn't work. The problem is that the paper is very long (80 pages of text). The sections on biogeochemical processes break the flow of ideas on ocean dynamics, are not complete, and don't do justice to range of important topics on biogeochemistry. For example, there is almost no meaningful discussion of OMZs and little discussion of how vertical mixing (section 7.3) affects biogeochemistry. You could add more on these and other BGC topics (like the ocean carbon cycle), but that is not the right solution with the paper so long already. My suggestion would be to focus this paper on just ocean dynamics to keep it to a manageable length. Then write a second companion paper using the material in this paper as a start on how ocean circulation and mixing processes affect Indian Ocean biogeochemistry, and what the implications are for ecosystems and fisheries. Some topics on biogeochemistry may still fit in a dynamics paper, like how primary productivity affects penetrative radiation in the mixed layer, but they will be limited and focused.

Specific comments follow in the order in which the occur.

*Abstract and Introduction. Calling out only IIOE-2 is inappropriate. IIOE-2 started in 2015. Much of the progress reviewed in this paper between the Schott and McCreary and Schott et al reviews has been made in programs that predated IIOE-2. Mentioning IIOE-2 to the exclusion of other national, bilateral and international (e.g. CLIVAR, GOOS) programs leaves the impression that it is only IIOE-2 that is responsible for all the wonderful science reported in the paper. This can be easily remedied by removing mention of IIOE-2 from the abstract and being more inclusive of other programs in the introduction. I flag this issue in the final section as well.

*Line 82. Call out the Ningaloo Nino here. This was a major new discovery in the past 10 years that gave rise to the term "marine heat wave" as noted on lines 1593-95.

*Lines 152ff an Lines 194ff. The shallow meridional overturning circulation in the Indian Ocean consists to two distinct cells, the subtropical cell (STC) and the cross-equatorial cell (CEC) (Lee, 2004). The ascending branches of these cells connect to different upwelling zones (e.g. SCTR region in the case of the STC, Somali in the case of the

CEC). These cells and need to be discussed in more detail rather than simply grouped together as part of the "meridional overturning circulation."

*Line 251 (Section 3.1.1/Fig.6) Indo-Pacific warming trends are warping the life cycle of the MJO, which is spending less time over the Indian Ocean, more time over the Pacific (Roxy et al, 2019). This warped life cycle is projecting onto mean rainfall trends in various parts of the globe and should be mentioned.

*Lines 277ff. Kelvin waves propagate energy not only eastward but downward into the interior ocean; they also propagate into the Indonesian seas where they affect the ITF (Pujiana and McPhaden, 2020). Intraseasonal winds excite Rossby waves directly, but Rossby wave are also excited by reflection of wind-forced Kelvin waves at the eastern boundary (Nagura and McPhaden, 2012; Pujiana and McPhaden, 2020). These points should be mentioned.

*Line 326. Cite Nagura and McPhaden (2014) here. Also, as they point out, it is not just intraseasonal time scale variability that rectifies into mean flows along the equator, but also lower frequencies as well (especially semi-annual time scales). This point should be expanded upon in short subsection at the start of Section 4.1 (see below).

*Line 399-400 Reference Chelton et al (2001) here

*Line 407. "…tropical Indian Ocean" rather than "tropics" for clarity.

*Section 4.1/4.2 Reference Nagura and McPhaden (2018) who used Argo and CTD today to map out the circulation and water masses in density classes associated with the shallow overturning circulation, with emphasis on the southern hemisphere.

*Line 467-70. Imprecise language: "conveys"?? Is this advection, waves, other?

*Line 826. Section 4.1
The mean circulation along the equator is unique compared to the Pacific and Atlantic and should be highlighted in a subsection at the start of 4.1. Mean westerly winds are downwelling favorable: surface convergence and thermocline divergence that are part of that downwelling circulation have been described in Wang and McPhaden (2017) from Argo and RAMA data (mean downwelling is mentioned in lines 898-900 but would be better to include in a subsection that included other notable interbasin differences in mean equatorial circulation). Also, in the Pacific and Atlantic, easterly winds produce an eastward mean undercurrent in the thermocline but in the Indian Ocean westerly winds do not produce a mean westward undercurrent. The reason is that nonlinear momentum advection drives mean eastward currents in the thermocline that flow up the zonal pressure gradient (Nagura and McPhaden, 2014). The near surface meridional mean flow is southward across the equator in the interior ocean in the surface branch of the cross-equatorial cell (Lee, 2004) consistent with Sverdrup dynamics (Wang and McPhaden, 2017). Also, Horii et al (2013) and Wang and McPhaden (2017) present the first observational evidence for the "equatorial roll", unique to the Indian Ocean and first

identified in models (Wacogne and Pacanowski, 1996) as reviewed in Schott et al (2009).

*I am surprised that "meridional circulation" (section 4.3.2) does not highlight the seasonal cross-equatorial flow of mass and heat. These variations need to be discussed. In particular, the seasonal cross-equatorial mass flux is oppositely directed along the western boundary and interior (Beal et al., 2013).  Flow in the interior is directed from the summer to the winter hemisphere (Horii et al., 2013; Wang and McPhaden, 2017) consistent with monsoon wind forced Ekman and Sverdrup dynamics as proposed in the model study of Miyama et al. (2003). Heat transports associated with these cross equatorial flows help to moderate seasonal climate of the region.

*Section 4.3.2 Meridional Circulation is more than just high frequency biweekly mixed Rossby gravity waves and similar high frequency phenomena.  It includes the meridional overturning circulation, which includes the cross equatorial cell and subtropical cell, their means, seasonal, interannual and longer term variations. This section should be retitled as something like 5-30 day ocean waves and instabilities. Note also that the recent studies on the topic of mixed Rossby gravity waves by Arzeno et al (2020) and Pujiana and McPhaden (2021) should be referenced.

In this section you could also include a discussion of convectively coupled atmospheric Kelvin waves (Baranowsky et al, 2016) and how they force ocean Kelvin waves, affect surface heat fluxes, and generate upper ocean turbulence (Pujiana and McPhaden, 2018).

*There are two sections 4.3.4, which is an error in labelling.
Section 4.3.4 (the first one) The undercurrents also undergo significant inter annual variations related to the IOD.  These variations are important in the mass and heat balance on IOD time scales, with significant impacts on upwelling and SST (Zhang et al., 2014; Nyadjro and McPhaden, 2014)

*Lines 852-53 and 909-10: Reference Nyadjro and McPhaden (2014)

*Line 1279.  For the strong negative IOD event in 2016, the Indian Ocean influence overwhelmed that of the Pacific leading to record low ITF volume transports because of the reduction in the interbasin pressure gradient (Pujiana et al., 2019).

*Line 1281. Reference Pujiana and McPhaden (2020)

*Line 1289-1295. Dong and McPhaden (2016) should be referenced here especially in relation to the recent hiatus in global warming and the interhemispheric contrasts in the Indian Ocean related to Pacific forcing of ITF mass transports

*Line 1433-45 repeats some of the earlier discussion of the ITF

*Line 1446-49.  Two recent studies on this topic should be referenced: Volkov et al (2020) and Nagura and McPhaden (2021)

*Line 1496. IOD variability internal to the Indian Ocean resembles recharge oscillator dynamics for ENSO, but equatorial heat content is less effective as a precursor for the IOD than for ENSO because of the strong impact of remote forcing from the Pacific on the IOD. Internal Indian Ocean dynamics however may contribute to the biennial nature of the IOD through the cycling of Kelvin/Rossby wave energy across the basin (McPhaden and Nagura, 2014).

*Section 7.2.3 repeats some of the same material on Wyrtki jets and ISOs presented previously

*Line 1805.  Observations have also have captured energetic and consequential 5-20 day convectively coupled Kelvin waves in the atmosphere (Baranowsky et al, 2016) that generate oceanic Kelvin waves, affect surface heat fluxes and generate upper ocean turbulence (Pujiana and McPhaden, 2018).

*Line 1531-41.  Not a BGC subsection like for other topics?

*Section 7.3 This section is presumably only about the Bay of Bengal, but mixing and the role of inertial waves in the SCTR should also be discussed (e.g. Cuypers et al, 2013; Sabu et al., 2021) in the paper.

*Line 1834-35.  Even longer records (10 years) have been used to infer via inverse methods the seasonal cycle of mixing, Kt and barrier layer effects and how they vary spatially in the Bay of Bengal (Girishkumar et al, 2020).

*Line 1873-75.  The influence of barrier layer induced subsurface warm layers on SST is not limited to just cyclone events (e.g. Girishkumar et al, 2013)

*I originally thought Section 7.5 was supposed to be for the entire paper, but then realized that is just about the Bay of Bengal. Section 7 has more subsections than any other part of the paper which was part of my confusion.  After almost 80 pages though, I thought other pieces needed to be put together, like how oceanic variability affects monsoon rainfall (see below) and how biogeochemistry affects ecosystems and fisheries.

*Section 8.  The big question is how does the Indian Ocean affect the monsoons and on what time/space scales? This is not addressed in a coherent way in the paper. I would have expected Izumo et al (2008) and articles like it on this topic to be discussed somewhere since it is such an important question.

*Line 1964. IIOE-2 contributed to the progress reported in this review, but only beginning in 2015.  The way this sentence is worded does not do justice to all the other programs involved.

**References**

All references beginning with Z are missing
England et al 2014 reference is missing

**Additional References**

Arzeno, I. B., S. N. Giddings, G. Pawlak, and R. Pinkel, 2020: Generation of Quasi-Biweekly Yanai Waves in the Equatorial Indian Ocean. *Geophys Res Lett*, 47, e2020GL088915. https://doi.org/10.1029/2020GL088915

Baranowski, D. B., M. K. Flatau, P. J. Flatau, and A. J. Matthews (2016), Impact of atmospheric convectively coupled equatorial kelvin waves on upper ocean variability, Journal of Geophysical Research-Atmospheres, 121(5), 2045–2059, doi:10.1002/2015jd024150.

Beal, L. M., Hormann, V., Lumpkin, R., & Foltz, G. R. (2013). The Response of the Surface Circulation of the Arabian Sea to Monsoonal Forcing, Journal of Physical Oceanography, 43(9), 2008-2022. https://journals.ametsoc.org/view/journals/phoc/43/9/jpo-d-13-033.1.xml

Chelton, D.B., S.K. Esbensen, M.G. Schlax, N. Thum, M.H. Freilich, F.J. Wentz, C.L. Gentemann, M.J. McPhaden, and P.S. Schopf, 2001: Observations of coupling between surface wind stress and sea surface temperature in the eastern tropical Pacific. J. Climate, 14, 1479–1498.

Cuypers, Y., X. Le Vaillant, P. Bouruet-Aubertot, J. Vialard and M. J. McPhaden, 2013: Tropical storm-induced near-inertial internal waves during the Cirene experiment: energy fluxes and impact on vertical mixing. *J. Geophys. Res., 118,* 358-380, doi: 10.1029/2012JC007881.

Dong, L. and M.J. McPhaden, 2016: Interhemispheric SST gradient trends in the Indian Ocean prior to and during the recent global warming hiatus. *J. Climate*, *29*, 9077-9095.

Girishkumar, M. S., M. Ravichandran and M. J. McPhaden, 2013: Temperature inversions and their influence on the mixed layer heat budget during the winters of 2006-07 and 2007-08 in the Bay of Bengal. *J. Geophys. Res.,118*, doi:10.1002/jgrc.20192.

Girishkumar, M.S., K. Ashin, M.J. McPhaden, B. Balaji, and B. Praveenkumar, 2020: Estimation of vertical heat diffusivity at the base of the mixed layer in the Bay of Bengal. *J. Geophys. Res.*, *125*, e2019JC015402. http://dx.doi.org/10.1029/2019JC015402.

Horii, T., K. Mizuno, M. Nagura, T. Miyama, and K. Ando (2013), Seasonal and interannual variation in the cross-equatorial meridional currents observed in the eastern Indian Ocean, J. Geophys. Res., 118, 6658–6671, doi:10.1002/2013JC009291.

Izumo, T., C. de Boyer Montegut, J.J. Luo, S.K. Behera, S. Masson, and T. Yamagata, 2008: The role of the western Arabian Sea upwelling in Indian monsoon rainfall variability. J. Clim., 21, 5603–5623, doi:10.1175/2008JCLI2158.1.
McPhaden, M. J. and M. Nagura, 2014: Indian Ocean Dipole interpreted in terms of Recharge Oscillator theory. *Clim. Dyn., 42*, 1569–1586. doi 10.1007/s00382-013-1765-1.

Miyama, T., J. P. McCreary, T. G. Jensen, J. Loschnigg, S. Godfrey, and A. Ishida (2003), Structure and dynamics of the Indian-Ocean crossequatorial cell, Deep Sea Res., Part I, 50(12), 2023–2047.

Nagura, M., and M. J. McPhaden, 2012: The dynamics of wind-driven intraseasonal variability in the equatorial Indian Ocean. *J. Geophys. Res., 115,* C07009, doi:10.1029/2011JC007405.

Nagura, M. and M. J. McPhaden, 2014: Zonal momentum budget along the equator in the Indian Ocean from a high resolution ocean general circulation model. *J. Geophys. Res.*, *119*, 4444-4461, doi:10.1002/2014JC009895.

Nagura, M. and M.J. McPhaden, 2018: The Shallow Overturning Circulation in the Indian Ocean, *J. Phys. Oceanogr., 48*, 413-434.

Nagura, M. and M. J. McPhaden, 2021: Interannual variability in sea surface height at southern mid-latitudes of the Indian. J. Phys. Oceanogr., https://doi.org/10.1175/JPO-D-20-0279.1.

Nyadjro, E. and M. J. McPhaden, 2014: Variability of zonal currents in the eastern equatorial Indian Ocean on seasonal to interannual time scales. *J. Geophys. Res.*, *119*, 7969-7986, doi:10.1002/2014JC010380.

Pujiana, K. and M.J. McPhaden, 2018: Ocean's response to the convectively coupled Kelvin waves in the eastern equatorial Indian Ocean. *J. Geophys. Res., 123*, 5727-5741. https://doi.org/10.1029/2018JC013858.

Pujiana, K. and M.J. McPhaden, 2020: Intraseasonal Kelvin waves in the equatorial Indian Ocean and their propagation into the Indonesian Seas. *J. Geophys. Res, 25*. https://doi.org/10.1029/2019JC015839.

Pujiana, K. and M. J. McPhaden, 2021: Biweekly mixed Rossby-Gravity waves in the equatorial Indian Ocean.  J. Geophys. Res.,  https://doi.org/10.1029/2020JC016840.

Pujiana, K., M.J. McPhaden, A.L. Gordon, and A.M. Napitu, 2019: Unprecedented response of Indonesian throughflow to anomalous Indo-Pacific climatic forcing in 2016. *J. Geophys. Res*., 124, 3737-3754. https://doi.org/10.1029/2018JC014574.

Roxy, M.K., P. Dasgupta, M.J. McPhaden, T. Suematsu, C. Zhang, and D. Kim, 2019: Twofold expansion of the Indo-Pacific warm pool warps the MJO life cycle. *Nature, 575*, 647-651. https://doi.org/10.1038/s41586-019-1764-4.

Sabu, P., M.P. Subeesh, J.V. George et al., 2021: Enhanced subsurface mixing due to near-inertial waves: observation from Seychelles-Chagos Thermocline Ridge. Ocean Dynamics 71, 391–409. https://doi.org/10.1007/s10236-020-01430-z
Volkov, D.L., S.-K. Lee, A.L. Gordon, and M. Rudko, 2020: Unprecedented reduction and quick recovery of the South Indian Ocean heat content and sea level in 2014-2018. Sci. Adv., 6, eabc1151

Wacongne, S., and R. C. Pacanowski (1996), Seasonal heat transport in a primitive equation model of the tropical Indian Ocean, J. Phys. Oceanogr., 26, 2666–2699.

Wang, Y. and M.J. McPhaden, 2017: Seasonal Cycle of Cross-Equatorial Flow in the Central Indian Ocean. *J. Geophys. Res*., *122*, doi:10.1002/2016JC012537.

Zhang, D., M. J. McPhaden, and T. Lee, 2014: Observed Interannual Variability of Zonal Currents in the Equatorial Indian Ocean Thermocline and Their Relation to Indian Ocean Dipole. *Geophys. Res. Lett.*, 41, 7933-7941, doi: 10.1002/2014GL061449.

---

## Author Comment (AC1)

We received two comprehensive and constructive reviews and were grateful for the advice and additional references provided by the reviewers. Both reviewers commented on the structure of the paper and on the uneven representation of some of the topics. We have reworked the manuscript to address both of these comments. We feel that the manuscript is substantially improved and hope that the reviewers agree. The revised manuscript is 8 pages longer than the original but the bulk of this is in the additional references and supporting text recommended by the reviewers.

Reviewer 1 suggested that much of the biogeochemical variability could be removed and converted into a second BGC-focussed paper. We had extensive discussions within the author group and decided that a BGC-only paper would be hard to write without extensive physical knowledge, so either there would be a lot of repetition, or the second BGC paper would not stand on its own. We have decided to keep the BGC discussion, with some reorganisation and reduction in length.

In the following, we address the individual comments of both reviewers. Their original comments are included in black text and our response is in blue text.

**Reviewer 1**

Review of "Progress in understanding of Indian Ocean circulation, variability, air-sea exchange and impacts on biogeochemistry" by Phillips et al.

This is an ambitious effort to summarize scientific advances since the last comprehensive reviews of Indian Ocean dynamics and their role in the climate system by Schott and McCreary (2001) and Schott et al (2009). The paper contains an impressive amount of information compiled by an impressive list of authors. The writing is generally clear and the range of topics comprehensive. I consider myself an Indian Ocean expert and still learned a lot by reading it.

We thank the reviewer (Dr. Michael McPhaden) for his thorough review and his insightful comments. We very much appreciate the additional references and notes he has provided to fill some gaps in our original submission. As detailed below, we have taken nearly all of his comments into account.

I list specific comments for the authors to consider below. Some are related to missing ideas, references that should be cited, or organizational issues. The biggest concern I have though is the length of the paper and the effort to include a comprehensive review of ocean dynamics with biogeochemistry. I appreciate that the authors are attempting an interdisciplinary synthesis, but in my opinion it doesn't work. The problem is that the paper is very long (80 pages of text). The sections on biogeochemical processes break the flow of ideas on ocean dynamics, are not complete, and don't do justice to range of important topics on biogeochemistry. For example, there is almost no meaningful discussion of OMZs and little discussion of how vertical mixing (section 7.3) affects biogeochemistry. You could add more on these and other BGC topics (like the ocean carbon cycle), but that is not the right solution with the paper so long already. My suggestion would be to focus this paper on just ocean dynamics to keep it to a manageable length. Then write a second companion paper using the material in this paper as a start on how ocean circulation and mixing processes affect Indian Ocean biogeochemistry, and what the implications are for ecosystems and fisheries. Some

topics on biogeochemistry may still fit in a dynamics paper, like how primary productivity affects penetrative radiation in the mixed layer, but they will be limited and focused.

To respond to Dr. McPhaden's concern, both about the length and the organization of the paper, we had extensive discussions within the group. The authors (including both the authors who are not the BGC experts, and those who specialize in it) thought that the aspects covering the BGC are a very special part of this paper. Initially the leading/convening authors (Phillips and Tandon) liked the idea of a second paper; however, after discussions with the BGC contributors it seemed that a BGC alone paper would be hard to write without extensive physical knowledge, so either there would be a lot of repetition, or the second BGC paper would not stand on its own. After a fair amount of deliberation, we have decided to keep the BGC discussion, with some reduction in length.

We also thought carefully about Dr. McPhaden's comment on the organization and length. As a result, we rearranged Section 4 so that the BGC variability part in Section 4 comes at the end of each level 2 section. E.g.

4.2 Southern Indian Ocean

4.2.1 South Equatorial Current
4.2.2 Western Boundary
4.2.3 Interior Flows
4.2.4 Eastern Boundary
4.2.5 Biogeochemical variability

There was some inconsistency with the section level for the BGC text before, so this is now improved. We think that now with the BGC grouped together, there is more opportunity to reduce repetition and tighten up the text. To note the lack of OMZ discussion in the paper we have directed the reader to Rixen et al. (2020) and McCreary et al. (2013) reviews of OMZ. Although we don't have a section dedicated to vertical mixing, it is part of the discussion throughout the manuscript. For example, Section 3.2.1 (Influence of the MJO; relative importance of surface heat fluxes and subsurface ocean processes for the evolution of SST) and Section 4.3.6 (biogeochemical variability in the equatorial region).

Specific comments follow in the order in which the occur.

*Abstract and Introduction. Calling out only IIOE-2 is inappropriate. IIOE-2 started in 2015. Much of the progress reviewed in this paper between the Schott and McCreary and Schott et al reviews has been made in programs that predated IIOE-2. Mentioning IIOE-2 to the exclusion of other national, bilateral and international (e.g. CLIVAR, GOOS) programs leaves the impression that it is only IIOE-2 that is responsible for all the wonderful science reported in the paper. This can be easily remedied by removing mention of IIOE-2 from the abstract and being more inclusive of other programs in the introduction. I flag this issue in the final section as well.

This review paper was born out of IIOE-2 efforts and is specifically part of an IIOE-2 special edition. However, we unintentionally conflated the achievements of all of the other programs while only explicitly mentioning IIOE-2. You are quite right that IIOE-2

is a newcomer to the delivery of Indian Ocean science, and has played a coordinating role, rather than funding science. We have tried to provide a more balanced perspective in the revised manuscript and have adopted your suggestions for revision of the abstract and introduction.

Specifically, mention of IIOE-2 has been removed from the abstract and replaced by 'Coordinated international focus on the Indian Ocean..' (Line 42).

In the Introduction, we revised the text to be more inclusive of other programs (Lines 58-69).

*Line 82. Call out the Ningaloo Nino here. This was a major new discovery in the past 10 years that gave rise to the term "marine heat wave" as noted on lines 1593-95.

We have added mention of the Ningaloo Niño in the Abstract (Line 47) and expand on this topic in Section 7.

*Lines 152ff an Lines 194ff. The shallow meridional overturning circulation in the Indian Ocean consists to two distinct cells, the subtropical cell (STC) and the cross-equatorial cell (CEC) (Lee, 2004). The ascending branches of these cells connect to different upwelling zones (e.g. SCTR region in the case of the STC, Somali in the case of the CEC). These cells and need to be discussed in more detail rather than simply grouped together as part of the "meridional overturning circulation."

We have included the shallow cells in the first mention of poleward heat flow (Line 173-175) and have added the points you raise in the description of the upper ocean overturning (Line 181) and in Section 3.1, including the references to Schott et al., 2002; Schott et al., 2009; Lee 2004). We have also added a new subsection about the CEC (Section 4.3.5).

*Line 251 (Section 3.1.1/Fig.6) Indo-Pacific warming trends are warping the life cycle of the MJO, which is spending less time over the Indian Ocean, more time over the Pacific (Roxy et al, 2019). This warped life cycle is projecting onto mean rainfall trends in various parts of the globe and should be mentioned.

This is an important point that we now include in Section 3.2.1 (Line 285-287).

*Lines 277ff. Kelvin waves propagate energy not only eastward but downward into the interior ocean; they also propagate into the Indonesian seas where they affect the ITF (Pujiana and McPhaden, 2020). Intraseasonal winds excite Rossby waves directly, but Rossby wave are also excited by reflection of wind-forced Kelvin waves at the eastern boundary (Nagura and McPhaden, 2012; Pujiana and McPhaden, 2020). These points should be mentioned.

We thank the reviewer for these suggestions, which have now all been included in the revised manuscript (Lines 301-304, 309-310)

*Line 326. Cite Nagura and McPhaden (2014) here. Also, as they point out, it is not just intraseasonal time scale variability that rectifies into mean flows along the equator, but also lower frequencies as well (especially semi-annual time scales). This point should be expanded upon in short subsection at the start of Section 4.1 (see below).

We have added this citation at Lines 363-364.

*Line 399-400 Reference Chelton et al (2001) here

We removed the link to tropical instability waves in this section and so did not need to add this reference.

*Line 407. "…tropical Indian Ocean" rather than "tropics" for clarity.

Done (line number 443)

*Section 4.1/4.2 Reference Nagura and McPhaden (2018) who used Argo and CTD today to map out the circulation and water masses in density classes associated with the shallow overturning circulation, with emphasis on the southern hemisphere.

We have added this information at the end of Section 4.1 (line 486).

*Line 467-70. Imprecise language: "conveys"?? Is this advection, waves, other?

The expanded description in this part now provides more detail that distinguishes between advection and waves (Lines 514-518).

*Line 826. Section 4.1
The mean circulation along the equator is unique compared to the Pacific and Atlantic and should be highlighted in a subsection at the start of 4.1. Mean westerly winds are downwelling favorable: surface convergence and thermocline divergence that are part of that downwelling circulation have been described in Wang and McPhaden (2017) from Argo and RAMA data (mean downwelling is mentioned in lines 898-900 but would be better to include in a subsection that included other notable interbasin differences in mean equatorial circulation). Also, in the Pacific and Atlantic, easterly winds produce an eastward mean undercurrent in the thermocline but in the Indian Ocean westerly winds do not produce a mean westward undercurrent. The reason is that nonlinear momentum advection drives mean eastward currents in the thermocline that flow up the zonal pressure gradient (Nagura and McPhaden, 2014). The near surface meridional mean flow is southward across the equator in the interior ocean in the surface branch of the cross-equatorial cell (Lee, 2004) consistent with Sverdrup dynamics (Wang and McPhaden, 2017). Also, Horii et al (2013) and Wang and McPhaden (2017) present the first observational evidence for the "equatorial roll", unique to the Indian Ocean and first identified in models (Wacogne and Pacanowski, 1996) as reviewed in Schott et al (2009).

Thanks for providing these important points to include and additional references. We have included these in Section 4.3. Rather than having a separate section called "Interbasin Differences", we decided to keep the separation of the unique Indian Ocean features into separate subsections and we added a new subsection on the CEC and included some information on the equatorial roll. Section 4.3 subsections are now:
4.3.1 Wyrtki Jets
4.3.2 5-30 Day Ocean Waves
4.3.3 Equatorial Upwelling and Downwelling
4.3.4 Equatorial Undercurrents

Although some headings are similar to the original submission, the content of each is now expanded. In addition to the references you have provided, we also cite Miyama, T., McCreary, J.P., Jensen, T.G., Loschnigg, J.L., Godfrey, S., and Ishida, A.: Structure and dynamics of the Indian-Ocean cross-equatorial cell, Deep Sea Res. II, 50, 2023–2047, https://doi.org/10.1016/S0967-0645(03)00044-4, 2003.

*I am surprised that "meridional circulation" (section 4.3.2) does not highlight the seasonal cross-equatorial flow of mass and heat. These variations need to be discussed. In particular, the seasonal cross-equatorial mass flux is oppositely directed along the western boundary and interior (Beal et al., 2013). Flow in the interior is directed from the summer to the winter hemisphere (Horii et al., 2013; Wang and McPhaden, 2017) consistent with monsoon wind forced Ekman and Sverdrup dynamics as proposed in the model study of Miyama et al. (2003). Heat transports associated with these cross equatorial flows help to moderate seasonal climate of the region.

We have now included this information in Section 4.3.5 Cross-Equatorial Cell.

*Section 4.3.2 Meridional Circulation is more than just high frequency biweekly mixed Rossby gravity waves and similar high frequency phenomena. It includes the meridional overturning circulation, which includes the cross equatorial cell and subtropical cell, their means, seasonal, interannual and longer term variations. This section should be retitled as something like 5-30 day ocean waves and instabilities. Note also that the recent studies on the topic of mixed Rossby gravity waves by Arzeno et al (2020) and Pujiana and McPhaden (2021) should be referenced.

We changed the name of Section 4.3.2 as you suggested and have added the two additional references (Line 895).

In this section you could also include a discussion of convectively coupled atmospheric Kelvin waves (Baranowsky et al, 2016) and how they force ocean Kelvin waves, affect surface heat fluxes, and generate upper ocean turbulence (Pujiana and McPhaden, 2018).

We have adopted your suggestions and added a brief description of CCKW, their link to the MJO as well as the oceanic response to them (Lines 916-924).

*There are two sections 4.3.4, which is an error in labelling.

This is corrected now. Thankyou.

Section 4.3.4 (the first one) The undercurrents also undergo significant inter annual variations related to the IOD. These variations are important in the mass and heat balance on IOD time scales, with significant impacts on upwelling and SST (Zhang et al., 2014; Nyadjro and McPhaden, 2014)

We have added this point to the end of Section 4.3.4 (lines 1888-1891).

*Lines 852-53 and 909-10: Reference Nyadjro and McPhaden (2014)

This reference is now added (line 883 and 957).

*Line 1279. For the strong negative IOD event in 2016, the Indian Ocean influence overwhelmed that of the Pacific leading to record low ITF volume transports because of the reduction in the interbasin pressure gradient (Pujiana et al., 2019).

This information has been added (lines 1321-1323).

*Line 1281. Reference Pujiana and McPhaden (2020)

This reference has been added (line 1326).

*Line 1289-1295. Dong and McPhaden (2016) should be referenced here especially in relation to the recent hiatus in global warming and the interhemispheric contrasts in the Indian Ocean related to Pacific forcing of ITF mass transports

This point has been added (line 1337).

*Line 1433-45 repeats some of the earlier discussion of the ITF

We have deleted the first sentence of this paragraph, which repeats the relationship between ITF transport and ENSO phase described earlier.

*Line 1446-49. Two recent studies on this topic should be referenced: Volkov et al (2020) and Nagura and McPhaden (2021)

These references have been added (line 1492).

*Line 1496. IOD variability internal to the Indian Ocean resembles recharge oscillator dynamics for ENSO, but equatorial heat content is less effective as a precursor for the IOD than for ENSO because of the strong impact of remote forcing from the Pacific on the IOD. Internal Indian Ocean dynamics however may contribute to the biennial nature of the IOD through the cycling of Kelvin/Rossby wave energy across the basin (McPhaden and Nagura, 2014).

Text to this effect has been added in Section 6.2 (lines 1547-1550).

*Section 7.2.3 repeats some of the same material on Wyrtki jets and ISOs presented previously

We didn't find any repetition between Section 7.2.3 on Wyrtki Jets and/or ISOs with previous sections (e.g. 4.3.1). We preferred to keep the discussion of the BoB case study complete, so have not removed any material on Wyrtki jets and ISOs from section 7.2.3.

*Line 1805. Observations have also have captured energetic and consequential 5-20 day convectively coupled Kelvin waves in the atmosphere (Baranowski et al, 2016) that generate oceanic Kelvin waves, affect surface heat fluxes and generate upper ocean turbulence (Pujiana and McPhaden, 2018).

We have added this information to Section 7.2.3 (lines 1904-1907).

*Line 1531-41. Not a BGC subsection like for other topics?

We have added a subheading for this BGC discussion (Line 1586). We initially left it out because it was only in the IOD section that there were BGC points to make. We agree with you that it's better to declare this text as BGC Variability.

*Section 7.3 This section is presumably only about the Bay of Bengal, but mixing and the role of inertial waves in the SCTR should also be discussed (e.g. Cuypers et al, 2013; Sabu et al., 2021) in the paper.

This has been addressed in section 3.2.1, (lines 331-333)

*Line 1834-35. Even longer records (10 years) have been used to infer via inverse methods the seasonal cycle of mixing, Kt and barrier layer effects and how they vary spatially in the Bay of Bengal (Girishkumar et al, 2020).

This is now included in Section 7.3 (lines 1945-1951).

*Line 1873-75. The influence of barrier layer induced subsurface warm layers on SST is not limited to just cyclone events (e.g. Girishkumar et al, 2013)

We have added a sentence to indicate the influence on longer timescales and cite this reference (lines 1979-1982).

*I originally thought Section 7.5 was supposed to be for the entire paper, but then realized that is just about the Bay of Bengal. Section 7 has more subsections than any other part of the paper which was part of my confusion. After almost 80 pages though, I thought other pieces needed to be put together, like how oceanic variability affects monsoon rainfall (see below) and how biogeochemistry affects ecosystems and fisheries.

*Section 8. The big question is how does the Indian Ocean affect the monsoons and on what time/space scales? This is not addressed in a coherent way in the paper. I would have expected Izumo et al (2008) and articles like it on this topic to be discussed somewhere since it is such an important question.

We have added a new section (Line 1677) with the following text to address this question.

Section 6.5 Monsoon variability and links to the Indian Ocean
Several monsoon systems surround the Indian Ocean, notably the South Asian monsoon, the East Asian monsoon and the Australian monsoon. These monsoon systems are remotely influenced by global coupled modes of variability such as ENSO, which is often associated with

dry conditions in the South Asian monsoon (e.g., Rasmusson and Carpenter, 1983; Ropelewski and Halpert, 1987) and Australian monsoon (e.g., Risbey et al., 2009; Jourdain et al., 2013), although the relationship with the Indian monsoon has recently weakened (e.g., Kumar et al., 1999). In the Indian Ocean, the IOD has a strong influence on the Asian monsoon systems, but is weak during the Australian monsoon period. The IOD tends to oppose the ENSO teleconnection to the South Asian monsoon by enhancing monsoon rainfall (e.g., Ashok et al., 2004; Chowdary et al., 2015; Krishnaswamy et al., 2015; Pokhrel et al., 2012). However, the exact combination of SST patterns between the Indian Ocean and the Pacific is crucial for determining the rainfall response in the Asian monsoons (e.g., Lau and Wu, 2001; Ratna et al., 2020; Yuan and Yang, 2012), and the relative strengths of the teleconnections have varied over time (Krishnaswamy et al., 2015). Furthermore, there is evidence that the Indian Ocean forcing of the South Asian monsoon may be primarily driven by ENSO, with pure IOD events only weakly influencing monsoon rainfall (Cretat et al., 2017).

The monsoon systems around the Indian Ocean tend to vary in phase and are also linked to the western North Pacific Monsoon (e.g., Gu et al., 2010). There is a biennial oscillation in the strength of the monsoon systems, with a strong Asian monsoon preceding a negative IOD and coinciding with cold eastern Pacific SSTs, followed by a strong Australian monsoon and subsequently by a reversal in the SST patterns (Loschnigg et al., 2003; Meehl & Arblaster, 2011). Thus, each monsoon system interacts with the ocean dynamics and thermodynamics and with the other monsoon systems through a complex set of teleconnections.

At a regional scale, upwelling in the Arabian Sea reduces rainfall along the western Ghats of India during the monsoon due to a reduction in evaporation and water vapour transport (Izumo et al., 2008). Moisture fluxes across the Arabian Sea are crucial to accurate simulation of the Indian Monsoon, yet many models fail to accurately capture these (Levine and Turner, 2012). In the Bay of Bengal, the shallow surface mixed layer, supported by the vertical salinity gradient, leads to rapid variations in SST (e.g., Sengupta and Ravichandran, 2001; Vecchi and Harrison, 2002) that interact with intraseasonal oscillations (Gao et al., 2019) in the atmosphere and thus with the active/break cycles on the monsoon (e.g., Lucas et al., 2014). This strong and rapid variability in upper ocean conditions in the Bay of Bengal, and the potential feedbacks on the monsoon, motivated multiple observational research programmes with field campaigns in the Bay of Bengal, as discussed in the next section.

*Line 1964. IIOE-2 contributed to the progress reported in this review, but only beginning in 2015. The way this sentence is worded does not do justice to all the other programs involved.

We have reworded this sentence to be more inclusive, as we did for the Abstract and Introduction (Line 2070).

**References**

All references beginning with Z are missing
England et al 2014 reference is missing

Thankyou for checking. We have added England et al. (2014) and reinstated the missing Z references. We appreciate that you provided the full citations below.

**Additional References**

Arzeno, I. B., S. N. Giddings, G. Pawlak, and R. Pinkel, 2020: Generation of Quasi Biweekly Yanai Waves in the Equatorial Indian Ocean. *Geophys Res Lett*, 47, e2020GL088915. https://doi.org/10.1029/2020GL088915

Baranowski, D. B., M. K. Flatau, P. J. Flatau, and A. J. Matthews (2016), Impact of atmospheric convectively coupled equatorial kelvin waves on upper ocean variability, Journal of Geophysical Research-Atmospheres, 121(5), 2045–2059, doi:10.1002/2015jd024150.

Beal, L. M., Hormann, V., Lumpkin, R., & Foltz, G. R. (2013). The Response of the Surface Circulation of the Arabian Sea to Monsoonal Forcing, Journal of Physical Oceanography, 43(9), 2008-2022. https://journals.ametsoc.org/view/journals/phoc/43/9/jpo-d-13-033.1.xml

Chelton, D.B., S.K. Esbensen, M.G. Schlax, N. Thum, M.H. Freilich, F.J. Wentz, C.L. Gentemann, M.J. McPhaden, and P.S. Schopf, 2001: Observations of coupling between surface wind stress and sea surface temperature in the eastern tropical Pacific. J. Climate, 14, 1479–1498.

Cuypers, Y., X. Le Vaillant, P. Bouruet-Aubertot, J. Vialard and M. J. McPhaden, 2013: Tropical storm-induced near-inertial internal waves during the Cirene experiment: energy fluxes and impact on vertical mixing. *J. Geophys. Res., 118,* 358-380, doi: 10.1029/2012JC007881.

Dong, L. and M.J. McPhaden, 2016: Interhemispheric SST gradient trends in the Indian Ocean prior to and during the recent global warming hiatus. *J. Climate*, *29*, 9077-9095.

Girishkumar, M. S., M. Ravichandran and M. J. McPhaden, 2013: Temperature inversions and their influence on the mixed layer heat budget during the winters of 2006-07 and 2007-08 in the Bay of Bengal. *J. Geophys. Res.,118*, doi:10.1002/jgrc.20192.

Girishkumar, M.S., K. Ashin, M.J. McPhaden, B. Balaji, and B. Praveenkumar, 2020: Estimation of vertical heat diffusivity at the base of the mixed layer in the Bay of Bengal. *J. Geophys. Res*., *125*, e2019JC015402. http://dx.doi.org/10.1029/2019JC015402.

Horii, T., K. Mizuno, M. Nagura, T. Miyama, and K. Ando (2013), Seasonal and interannual variation in the cross-equatorial meridional currents observed in the eastern Indian Ocean, J. Geophys. Res., 118, 6658–6671, doi:10.1002/2013JC009291.

Izumo, T., C. de Boyer Montegut, J.J. Luo, S.K. Behera, S. Masson, and T. Yamagata, 2008: The role of the western Arabian Sea upwelling in Indian monsoon rainfall variability. J. Clim., 21, 5603–5623, doi:10.1175/2008JCLI2158.1.

McPhaden, M. J. and M. Nagura, 2014: Indian Ocean Dipole interpreted in terms of Recharge Oscillator theory. *Clim. Dyn., 42*, 1569–1586. doi 10.1007/s00382-013-1765- 1.

Miyama, T., J. P. McCreary, T. G. Jensen, J. Loschnigg, S. Godfrey, and A. Ishida (2003), Structure and dynamics of the Indian-Ocean crossequatorial cell, Deep Sea Res., Part I, 50(12), 2023–2047.

Nagura, M., and M. J. McPhaden, 2012: The dynamics of wind-driven intraseasonal variability in the equatorial Indian Ocean. *J. Geophys. Res., 115,* C07009, doi:10.1029/2011JC007405.

Nagura, M. and M. J. McPhaden, 2014: Zonal momentum budget along the equator in the Indian Ocean from a high resolution ocean general circulation model. *J. Geophys. Res., 119*, 4444-4461, doi:10.1002/2014JC009895.

Nagura, M. and M.J. McPhaden, 2018: The Shallow Overturning Circulation in the Indian Ocean, *J. Phys. Oceanogr., 48*, 413-434.

Nagura, M. and M. J. McPhaden, 2021: Interannual variability in sea surface height at southern mid-latitudes of the Indian. J. Phys. Oceanogr., https://doi.org/10.1175/JPO-D 20-0279.1.

Nyadjro, E. and M. J. McPhaden, 2014: Variability of zonal currents in the eastern equatorial Indian Ocean on seasonal to interannual time scales. *J. Geophys. Res., 119*, 7969-7986, doi:10.1002/2014JC010380.

Pujiana, K. and M.J. McPhaden, 2018: Ocean's response to the convectively coupled Kelvin waves in the eastern equatorial Indian Ocean. *J. Geophys. Res., 123*, 5727-5741. https://doi.org/10.1029/2018JC013858.

Pujiana, K. and M.J. McPhaden, 2020: Intraseasonal Kelvin waves in the equatorial Indian Ocean and their propagation into the Indonesian Seas. *J. Geophys. Res, 25.* https://doi.org/10.1029/2019JC015839.

Pujiana, K. and M. J. McPhaden, 2021: Biweekly mixed Rossby-Gravity waves in the equatorial Indian Ocean. J. Geophys. Res., https://doi.org/10.1029/2020JC016840.

Pujiana, K., M.J. McPhaden, A.L. Gordon, and A.M. Napitu, 2019: Unprecedented response of Indonesian throughflow to anomalous Indo-Pacific climatic forcing in 2016. *J. Geophys. Res.*, 124, 3737-3754. https://doi.org/10.1029/2018JC014574.

Roxy, M.K., P. Dasgupta, M.J. McPhaden, T. Suematsu, C. Zhang, and D. Kim, 2019: Twofold expansion of the Indo-Pacific warm pool warps the MJO life cycle. *Nature, 575*, 647-651. https://doi.org/10.1038/s41586-019-1764-4.

Sabu, P., M.P. Subeesh, J.V. George et al., 2021: Enhanced subsurface mixing due to near-inertial waves: observation from Seychelles-Chagos Thermocline Ridge. Ocean Dynamics 71, 391–409. https://doi.org/10.1007/s10236-020-01430-z

Volkov, D.L., S.-K. Lee, A.L. Gordon, and M. Rudko, 2020: Unprecedented reduction and quick recovery of the South Indian Ocean heat content and sea level in 2014-2018. Sci. Adv., 6, eabc1151

Wacongne, S., and R. C. Pacanowski (1996), Seasonal heat transport in a primitive equation model of the tropical Indian Ocean, J. Phys. Oceanogr., 26, 2666–2699.

Wang, Y. and M.J. McPhaden, 2017: Seasonal Cycle of Cross-Equatorial Flow in the Central Indian Ocean. *J. Geophys. Res*., *122*, doi:10.1002/2016JC012537.

Zhang, D., M. J. McPhaden, and T. Lee, 2014: Observed Interannual Variability of Zonal Currents in the Equatorial Indian Ocean Thermocline and Their Relation to Indian Ocean Dipole. *Geophys. Res. Lett.*, 41, 7933-7941, doi: 10.1002/2014GL061449.

**Reviewer 2**

Anonymous Referee #2

Referee comment on "Progress in understanding of Indian Ocean circulation, variability, air-sea exchange and impacts on biogeochemistry" by Helen E. Phillips et al., Ocean Sci. Discuss., https://doi.org/10.5194/os-2021-1-RC2, 2021

This is a timely review that updates popular earlier reviews by Schott and McCreary (2001) and Schott et al. (2009). The difference is that a large group of authors wrote this review. As such, the coverage of various topics is at places uneven and the quality of the writing is variable. Sections 4.2.3 and 4.2.4 are good in coverage and readability. The discussion of biogeochemical variability helps bridge to communities beyond physical oceanography.

We agree that there was an imbalance and variable presentation across the sections. We have restructured some of the material, notably the BGC discussion in Section 4. We have also worked through the manuscript to improve the flow and balance of material. We hope that the reviewer will find the improvements satisfactory.

Major comments

The paper can benefit from better integration of different parts, possibly written different authors. A more balanced presentation among topics covered also helps, especially on climate. Here are some examples.

1. Abstract promises to bring together three major areas of research: the Indian Ocean circulation patterns, air-sea interactions, and climate variability. Reading the paper, I felt that that the emphasis was clearly on the ocean circulation while the latter topics did not receive as much attention. The Indian-western Pacific ocean capacitor (IPOC) perhaps represents an important advance in coupled ocean-atmosphere dynamics in the region but was hurried through in a single paragraph of 12 lines, including climate change, although it is a cross-basin mode (Xie et al. 2016, Adv Atmos Sci) with well-documented impacts on monsoon rainfall from India (Z. Zhou et al. 2019, GRL) to China and Japan (K. Hu et al. 2019, JC). In comparison, the text is 80 pages long, the Indian Ocean dipole (IOD) received a coverage of 4 pages and Ningaloo Niño 2 pages. This seems a lack of representation of IPOC research in the author team but a balanced approach is needed.

Thankyou for your comments. An extensive discussion of the IPOC is beyond the scope of the paper, since all reviewers commented on the overall length of the paper already. However, reference to IPOC and the suggested references are now added in Section 6. Information on IOD and Ningaloo Nino is presented directly in Sections 6.2 and 6.4, respectively. However, references to these phenomena are threaded through the whole manuscript in discussions of variability and impacts. We have worked to provide a better balance across sections and hope you will find the revision more acceptable. We added a new section: Section 6.5 Monsoon variability and links to the Indian Ocean (Line 1677-) as this was not well drawn out in the original submission. The summary section has also been revised to better bring together the three major areas covered in the manuscript.

2. Abstract promises to discuss the role of the Indian Ocean in climate change. L80-81: "This coupling (not sure what's this coupling) … causing a hiatus in the warming of Earth's surface atmosphere (Section 6)." I did not see further discussion of this topics in section 6, which is entitled "Modes of Interannual Climate Variability".

We have clarified the sentence that refers to coupling. It now reads *"This coupling between the ocean and atmosphere in the Indian and Pacific Oceans shifted the balance of global warming, accelerating ocean warming and causing a hiatus in the warming of Earth's surface atmosphere (Section 6)."* (line 91).

We have added a paragraph in Section 2 that discusses the role of the Indian Ocean in climate change (lines 153-161) and link back to this in Section 8 (lines 2123-2127)

L135-136: "The Indian Ocean accounts for 50-70% of the total anthropogenic warming in the global upper (700 m) ocean." This seems an overstatement. Do we even know how much "the anthropogenic warming" there is in the ocean? Does this merely refer to the trends over a certain time period?

We thank the reviewer for pointing this out. With regard to the amount of warming, this is supported by Lee et al. (2015). The term "anthropogenic warming" as presented in the Lee et al. (2015) study is in relation to the global heat budget, which has been shown to be on an upwards trend since the 1970s, in association with an increase in downward longwave radiation. Increases in downward longwave radiation are thought to be a result of the increase in GHG in the atmosphere, which is anthropogenic. We recognise, however, that the term 'anthropogenic warming' can be contentious and, in this instance, not necessarily clear cut, so we have altered the statement to take this into account. We now say "The Indian Ocean accounts for 50-70% of the total ocean heat uptake in the global upper (700 m) ocean over the last decade, associated with anthropogenic warming  (Lee et al., 2015)." (Lines 146-147).

Additional comments

Figure 1 is hard to read. It's probably from a ppt slide, which seems to include many layers of animation.

We have obtained a higher resolution version of this figure from Lisa Beal and now include this in the paper.

L296. "the SST variability is predominantly generated by variability in surface heat fluxes in the Seychelles-Chagos Thermocline Ridge." Shouldn't it first mention that intraseasonal SST

variability is large in this region (e.g., Saji et al. 2006, GRL)? I understand the need to emphasize recent results but a big picture needs to be presented first.

This has been amended in Section 3.2.1 and the Saji et al. 2006 reference has been added (lines 327-328).

L398-400. Is it really relevant to discuss Pacific TIWs?

We have removed the sentence about TIWs to keep the paragraph focused on the Indian Ocean.

L520-521. "The Agulhas variability is linked upstream to the IOD, SIOD and ENSO…" Is there a literature to back this up?

This is now clarified in Section 4.2.2 and references added (line 537).

L830-832. The semi-annual cycle in the zonal wind over the equatorial Indian Ocean is well known observationally but was never explained physically. Ogata and Xie (2011, JC) showed that it's due to the meridional advection of easterly momentum by the cross equatorial monsoon winds.

Thankyou for providing this information. We have included it in Section 4.3.1 (lines 860-861).

Does Figure 13 need to show two operational products?

We agree that only one of the products is necessary and have modified the figure to show only the COAMPS product.

L1460-1461. The projection was challenged by G. Li et a. (2016, JC), a paper entitled "A robust but spurious pattern of climate change in model projections over the tropical Indian Ocean."

We have now added text to this effect in Section 6.2 and highlight this caveat given model biases (lines 1508-1511).

Section 8 could be stronger. What happened during the 12-month period of Sept 2019-August 2020 showed that we don't know the Indian Ocean very well as a driver of major climate anomalies in the rim countries. The period opened with a record IOD (Doi et al. 2020; Du et al. 2020, GRL) and ended with historic heavy rainfall in China and Japan that have been attributed to a strong IPOC event (Takaya et al. 2020, GRL; Zhou et al. 2021 PNAS), all without major ENSO signals in the Pacific. I don't think we know well how this chain of events happened in the Indian Ocean but clearly we should.

We thank the reviewer for this suggestion. We now provide a brief discussion of the 2019 IOD in section 6.2 (lines 1581-1585) and in section 8 (Lines 2109-2111).

Several papers were cited in the text but not in References.

We regret that a number of papers were omitted from the reference list. We thank the reviewer for picking these up and have ensured they are now in the reference list.

L368. Zhou et al. (2017a, b)

Zhou, L., R. Murtugudde, D. Chen and Y. Tang, 2017a: A Central Indian Ocean Mode and Heavy Precipitation during Indian Summer Monsoon. J. Clim., DOI: 10.1175/JCLI-D-16-0347.1

Zhou, L., R. Murtugudde, D. Chen, and Y. Tang, 2017b: Seasonal and Interannual Variabilities of the Central Indian Ocean. J. Clim. DOI: 10.1175/JCLI-D-16-0616.1.

L391. Xi et al., 2015. Is it a typo of Xie et al. (2006, JC) on orographic effects on monsoon rainfall?

Xi, J., L. Zhou, R. Murtugudde, and L. Jiang, 2015: Impacts of intraseasonal SST anomalies on precipitation during Indian summer monsoon. J. Clim., 28, 4561-4575, doi: http://dx.doi.org/10.1175/JCLI-D-14-00096.1.

L1352. De Boer et al., 2013.

De Boer, A. M., Graham, R. M., Thomas, M. D., and Kohfeld, K. E. (2013), The control of the Southern Hemisphere Westerlies on the position of the Subtropical Front, J. Geophys. Res. Oceans, 118, 5669– 5675, doi:10.1002/jgrc.20407.

L1429. Xie et al. (2009)

Xie, S., Hu, K., Hafner, J., Tokinaga, H., Du, Y., Huang, G., and Sampe, T. (2009). Indian Ocean Capacitor Effect on Indo–Western Pacific Climate during the Summer following El Niño. Journal of Climate 22, 3, 730-747, https://doi.org/10.1175/2008JCLI2544.1

L1432. Zheng et al. (2013)

Zheng, X.-T., Xie, S.-P., Du, Y., Liu, L., Huang, G., and Liu, Q. Y.: Indian Ocean Dipole response to global warming in the CMIP5 multimodel ensemble. J. Climate, 26, 6067–6080, 2013.

---

## Author Response (AR2)

We thank Dr Michael McPhaden and Reviewer 2 for their time and thoughtful reviews of our revised manuscript. The manuscript is very much improved as a result of their efforts.

We have responded to each comment below. The reviewers' original text is in black and our response is in blue. Where line numbers are mentioned these refer to lines in the revised manuscript, which are different from the track change version submitted.

**Referee #1: Michael McPhaden, michael.j.mcphaden@noaa.gov**

The authors have done an excellent job to address my concerns on the original submission of this paper. The revision is much improved both in terms of content and organization. I recommend publication subject to a few minor adjustments as described below. I do not need to see the final revised version of the paper.

General: Since the paper is so long, it might be helpful to include a table of contents at the front end with the various sections and subsections listed. That way the reader can orient with a big picture overview and, if they are interested in only a subset of topics, can navigate directly to the relevant sections.
We found it useful to have a table of contents during the writing of the manuscript and appreciate your suggestion that it might also be useful for the reader. We have included this now.

Specifics:

Line 65. "topography". Do you mean "geography"?
We used the word topography because that is used in the IIOE-2 science plan in the list of questions being addressed in Science Theme 4. There is no need for us to follow the exact wording and we did not do so for the third science question. Geography would fit equally well, or better, and we have changed to this (Line 143).

Line 298. Kessler et al, 1995 and McPhaden 1999 proposed this connection before Bergman, so they should be cited as well.
Thank you. We have added these references.

Line 362-64. The juxtaposition of semi-annual with intraseasonal variability here is a little abrupt. Maybe move the reference to semi-annual variability to the section on the Wyrtki jets or provide a smoother connection at this point.
We agree that this mention of semi-annual variability is a little abrupt here. We removed it from the MJO section but found that it did not fit well in the Wyrtki Jet section either and so have deleted this text.

Section 3.2.2. Please point out that time scales for the MISO as for the MJO. The MJO is classically characterized by 30-60 day periods but MISOs can also occur on 10-20 day time scales (Goswami et al, 2016) and there are studies that have identified a 3-7 day time scale MISOs (e.g. Roman-Stork et al, 2020).
We have added this information about MISO timescales to the start of Section 3.2.2 and have included the suggested references.

Section 4.4.2.1. A new paper has appeared online by Zang et al (2021) on the Somali Current/Undercurrent that might be good to cite in this section.

This is a great addition. We have added a paragraph on this paper to the end of Section 4.4.2.1 (Line 1250).

Lines 2081-82. There has been progress on understanding circulation at intermediate and deeper depths, at least in the equatorial band. Huang et al (2018) summarize Indian Ocean studies of deep ocean propagating features and present some new results. You could possibly fold a short discussion of this topic into section 4.3.3 if you felt is would be of benefit. But it's not necessary given how much territory you've covered already in the paper.

Thank you for pointing out this reference, which we have now added. We have referred to it in a sentence at the end of Section 4.1, the overview of the circulation section (Line 565). Given the length of the paper already, we decided not to add more on this topic.

Line 2113-14: It might be helpful to mention marine heatwaves and their ecosystem impacts here.

Thank you for this suggestion. We have added a sentence about this at line 2208.

A final suggestion (though I'm guessing the authors have already considering this)--some kind of acknowledgement or maybe even a dedication to the recently deceased co-author Satya Prakash.

We have been considering how best to acknowledge Satya's contribution. We have been touched by the many fond messages from his colleagues and have included our message in a dedication just before the abstract of the paper.

References
These references were very helpful and are now included in the manuscript.

Goswami, B.N., S.A. Rao, D. Sengupta, and S. Chakravorty. 2016. Monsoons to mixing in the Bay of Bengal: Multiscale air-sea interactions and monsoon predictability. Oceanography 29(2):18–27, http://dx.doi.org/10.5670/oceanog.2016.35.

Huang, K., McPhaden, M. J., Wang, D., Wang, W., Xie, Q., Chen, J., et al. (2018). Vertical propagation of middepth zonal currents associated with surface wind forcing in the equatorial Indian Ocean. Journal of Geophysical Research: Oceans, 123, 7290–7307. https://doi.org/ 10.1029/2018JC013977

Kessler, W.S., M.J. McPhaden, and K.M. Weickmann, 1995: Forcing of intraseasonal Kelvin Waves in the equatorial Pacific. J. Geophys. Res., 100, 10,613–10,631.

McPhaden, M.J., 1999: Genesis and evolution of the 1997-98 El Niño. Science, 283, 950-954.

Roman-Stork, H. L., Subrahmanyam, B., & Trott, C. B. (2020). Monitoring intraseasonal oscillations in the Indian Ocean using satellite observations. Journal of

Geophysical Research: Oceans, 125, e2019JC015891.
https://doi.org/10.1029/2019JC015891

Zang, N., Sprintall, J., Ienny, R., Wang, F., Seasonality of the Somali Current/Undercurrent
System, Deep-Sea Research Part II, https://doi.org/10.1016/j.dsr2.2021.104953.

**Anonymous Referee #2**

I appreciate the authors' efforts to respond to the reviews. The resultant revision is
improved. Some of issues remain as highlighted below. I hope they can consider these
comments in finalizing the paper.

L144-147. I agree that the Indian Ocean is important for climate but its role should not
be overstated. The just-released IPCC AR6 does not support the claims here, for
example.

Box 9.2, Figure 1 (AR6). The Indian Ocean is small in area. I don't see how to justify the
claim "the western Indian Ocean… has been the largest contributor to the overall global
SST trend."

"The Indian Ocean accounts for 50-70% of the total ocean heat uptake in the global
upper (700 m) ocean over the last decade, associated with anthropogenic warming."
Figure 9.6 of AR6. does not support the statement for "anthropogenic warming." If the
50-70% figure is valid during the "last decade" (specify please), is it meaningful in terms
of the climatic effect?

Thank you for your attention to this point. We have revised the statement about rapid
warming in the Indian Ocean and now refer to Chapter 9 of the IPCC AR6. The text
starting at line 221 now reads
*The tropical Indian Ocean sea surface temperature (SST) has warmed faster over the
period since 1950 than either the tropical Pacific or Atlantic (Han et al., 2014, Fox-
Kemper et al., 2021), with implications for primary productivity (Roxy et al., 2014, 2016).*

L1475. The strong IOB-ENSO correlation is not limited to the post-1975 epoch but was
high "during the decades in the late nineteenth–early twentieth century" as well
(Chowdary et al. 2012, J Clim).

This sentence has been modified and the Chowdary et al (2012) reference has been
included (Line 1563).

L1582. The 2019 IOD "caused extreme rainfall and floods over Japan and China" not
directly but by exciting downwelling Rossby waves in the South Indian Ocean that
triggered the strong IPOC of 2020. See the discussion in the cited references (Takaya et
al. 2020; Zhou et al. 2021). This is an opportunity to integrate with section 6.1.

This sentence has been expanded and climate impacts associated with the 2019 IOD event put into a broader context. The modified text begins at Line 1670.

Saji et al. (1999) was cited in text but not listed in References.

There was a formatting error and this reference was hidden in the reference list before. It now shows clearly in the references.